# On conditional diffusion models for PDE simulations

**Aliaksandra Shysheya**[*]
University of Cambridge
as2975@cam.ac.uk

**Cristiana Diaconu**[*]
University of Cambridge
cdd43@cam.ac.uk

**Federico Bergamin**[*]
Technical University of Denmark
fedbe@dtu.dk

**Paris Perdikaris**
Microsoft Research AI4Science
paperdikaris@microsoft.com

**José Miguel Hernández-Lobato**
University of Cambridge
jmh233@cam.ac.uk

**Richard E. Turner**
University of Cambridge
Microsoft Research AI4Science
The Alan Turing Institute
ret23@cam.ac.uk

**Emile Mathieu**
University of Cambridge
ebm32@cam.ac.uk

## Abstract

Modelling partial differential equations (PDEs) is of crucial importance in science and engineering, and it includes tasks ranging from forecasting to inverse problems, such as data assimilation. However, most previous numerical and machine learning approaches that target forecasting cannot be applied out-of-the-box for data assimilation. Recently, diffusion models have emerged as a powerful tool for conditional generation, being able to flexibly incorporate observations without retraining. In this work, we perform a comparative study of score-based diffusion models for forecasting and assimilation of sparse observations. In particular, we focus on diffusion models that are either trained in a conditional manner, or conditioned after unconditional training. We address the shortcomings of existing models by proposing 1) an autoregressive sampling approach, that significantly improves performance in forecasting, 2) a new training strategy for conditional score-based models that achieves stable performance over a range of history lengths, and 3) a hybrid model which employs flexible pre-training conditioning on initial conditions and flexible post-training conditioning to handle data assimilation. We empirically show that these modifications are crucial for successfully tackling the combination of forecasting and data assimilation, a task commonly encountered in real-world scenarios.

## 1 Introduction

Partial differential equations (PDEs) are ubiquitous as they are a powerful mathematical framework for modelling physical phenomena, ranging from the motion of fluids to acoustics and thermodynamics. Applications are as varied as weather forecasting [4], train aerodynamics profiling [70], and concert hall design [76]. PDEs can either be solved through numerical methods like finite element methods [43], or, alternatively, there has been a rising interest in leveraging deep learning to learn neural approximations of PDE solutions [56]. These approaches have shown promise for generating accurate solutions in fields such as fluid dynamics and weather prediction [37, 52, 54].

Some of the most common tasks within PDE modelling are 1) *forecasting*, where the goal is to generate accurate and long rollouts based on some initial observations, and 2) *inverse problems*,

---

[*]Equal contribution.

38th Conference on Neural Information Processing Systems (NeurIPS 2024).

Table 1: Score-based methods considered in this work. Each method can be classified in terms of the score network (joint or conditional), the rollout strategy at sampling time, and the conditioning mechanism at inference time.

| MODEL | SCORE | ROLLOUT | CONDITIONING |
|---|---|---|---|
| Joint AAO [60] | $\mathbf{s}_\theta(t, x_{1:L}(t))$ | AAO | Guidance |
| Joint AR (ours) | $\mathbf{s}_\theta(t, x_{1:L}(t))$ | AR | Guidance |
| Universal amortised (ours) | $\mathbf{s}_\theta(t, x_{1:L}(t), y)$ | AR | Architecture/Guidance |

which aim to reconstruct a certain aspect of the PDE (i.e. coefficient, initial condition, full trajectory, etc.) given some partial observations of the solution to the PDE. Data assimilation (DA) refers to a particular class of inverse problems [62], with a goal to predict or refine a trajectory given some partial and potentially noisy observations. While in general these tasks are addressed separately, there are cases in which a model that could jointly tackle both tasks would be beneficial. For example, in weather prediction, a common task is to forecast future states, as well as update them after more observations from weather stations and satellites are available [29]. Currently, traditional numerical weather prediction systems tackle this in a two-stage process (a forecast model followed by a data assimilation system), both with a similarly high computational cost. A simpler option would be to have a model that is flexible enough to both produce accurate forecasts as well as condition on any incoming noisy observations.

In this work, we investigate how a probabilistic treatment of PDE dynamics [9, 81] can achieve this goal, with a focus on score-based diffusion models [26, 64, 68] as they have shown remarkable performance in other challenging applications, ranging from image and video [27, 58] to protein generation [72, 78, 79]. We perform a comparative study between the diffusion model approaches in Tab. 1, and evaluate their performance on forecasting and DA in PDE modelling. The unconditionally-trained model that is conditioned post-training through reconstruction guidance [65] is referred to as the *joint model*, while the *amortised model* directly fits a conditional score during training.

The main contribution of this work is that we propose extensions to the joint and amortised models to make them successfully applicable to tasks that combine forecasting and DA:

**Joint model.** We propose an autoregressive (AR) sampling strategy to overcome the limitations of the sampling approach proposed in [60], which samples full PDE trajectories all-at-once (AAO). We provide theoretical justification for why AR outperforms AAO, and empirically show, through extensive experiments, that AR outperforms or performs on par with AAO on DA, and drastically improves upon AAO in forecasting, where the latter fails.

**Amortised model.** We introduce a novel training procedure for amortised models for PDEs, that, unlike previous work [42], allows their performance to remain stable over a variety of history lengths. Moreover, we propose a hybrid model that combines amortisation with reconstruction guidance that is especially well-suited to mixed tasks, that involve both a forecasting and DA component.

Finally, this work provides, to the best of our knowledge, the first quantitative comparison between joint and amortised models for PDE modelling, with all other factors such as training dataset and model architecture carefully controlled.

## 2 Background

**Continuous-time diffusion models.** In this section we recall the main concepts underlying continuous diffusion models, and refer to Song et al. [68] for a thorough presentation. We consider a forward noising process $(x(t))_{t \geq 0}$ associated with the following stochastic differential equation (SDE)

$$\mathrm{d}x(t) = -\tfrac{1}{2}x(t)\mathrm{d}t + \mathrm{d}w(t), \ x(0) \sim p_0, \tag{1}$$

with $w(t)$ an isotropic Wiener process, and $p_0$ the data distribution. The process defined by (1) is the well-known Ornstein–Uhlenbeck process, which geometrically converges to the standard Gaussian distribution $\mathcal{N}(0, \mathrm{I})$. Additionally, for any $t \geq 0$, the noising kernel of (1) admits the following form $p_{t|0}(x(t)|x(0)) = \mathcal{N}(x(t)|\mu_t x(0), \sigma_t^2 \mathrm{I})$, with $\mu_t = e^{-t/2}$ and $\sigma_t^2 = 1 - e^{-t}$. Importantly, under mild assumptions on $p_0$, the time-reversal process $(\overleftarrow{x}(t))_{t \geq 0}$ also satisfies an SDE [10, 24] which is

given by

$$d\overleftarrow{x}(t) = \left\{-\tfrac{1}{2}\overleftarrow{x}(t) - \nabla \log p_t(\overleftarrow{x}(t))\right\}dt + dw(t), \tag{2}$$

where $p_t$ denotes the density of $x(t)$. In practice, the score $\nabla \log p_t$ is unavailable, and is approximated by a neural network $\mathbf{s}_\theta(t, \cdot) \approx \nabla \log p_t$, referred to as the *score network*. The parameters $\theta$ are learnt by minimising the denoising score matching (DSM) loss [30, 68, 73]

$$\mathcal{L}(\theta) = \mathbb{E}[\|\mathbf{s}_\theta(t, x(t)) - \nabla \log p_t(x(t)|x(0))\|^2], \tag{3}$$

where the expectation is taken over the joint distribution of $t \sim \mathcal{U}([0, 1])$, $x(0) \sim p_0$ and $x(t) \sim p_{t|0}(\cdot|x(0))$ from the noising kernel. Sampling involves discretising and solving (2). Optionally, to avoid errors accumulating along the denoising discretisation, each denoising step (predictor) can be followed by a small number of Langevin Monte Carlo (corrector) steps [67].

**Conditioning with diffusion models.** The methodology introduced above allows for (approximately) sampling from $x(0) \sim p_0$. However, in many settings, one is actually interested in sampling from the conditional $p(x(0)|y)$ given some observations $y \in \mathcal{Y}$ with likelihood $p(y|\mathcal{A}(x))$ and $\mathcal{A} : \mathcal{X} \to \mathcal{Y}$ being a *measurement* operator. The conditional is given by Bayes' rule $p(x(0)|y) = p(y|x(0))p(x(0))/p(y)$, yet it is not available in closed form. An alternative is to simulate the *conditional* denoising process [13, 15, 68]

$$d\overleftarrow{x}(t) = \left\{-\tfrac{1}{2}\overleftarrow{x}(t) - \nabla \log p_t(\overleftarrow{x}(t)|y)\right\}dt + dw(t). \tag{4}$$

In the following, we describe the two main ways to sample from this process.

**Amortising over observations.** The simplest approach directly learns the conditional score $\nabla \log p_t(\overleftarrow{x}(t)|y)$ [26, 68]. This can be achieved by additionally feeding $y$ to the score network $\mathbf{s}_\theta$, i.e. $\mathbf{s}_\theta(t, x(t), y)$, and optimising its parameters by minimising a loss akin to (3).

**Reconstruction guidance.** Alternatively, one can leverage Bayes' rule, which allows to express the conditional score w.r.t. $x(t)$ as the sum of the following two gradients

$$\nabla \log p(x(t)|y) = \underbrace{\nabla \log p(x(t))}_{\text{prior}} + \underbrace{\nabla \log p(y|x(t))}_{\text{guidance}}, \tag{5}$$

where $\nabla \log p(x(t))$ can be approximated by a score network $\mathbf{s}_\theta(t, x(t)) \approx \nabla \log p(x(t))$ trained on the prior data—with no information about the observation $y$. The conditioning comes into play via the $\nabla \log p(y|x(t))$ term which is referred to as *reconstruction guidance* [11, 13, 27, 48, 65, 66, 80]. Assuming a Gaussian likelihood $p(y|x(0)) = \mathcal{N}(y|\mathrm{A}x(0), \sigma_y^2 \mathrm{I})$ with linear measurement $\mathcal{A}(x) = \mathrm{A}x$, we have:

$$p(y|x(t)) = \int p(y|x(0))p(x(0)|x(t))\,dx(0) \approx \int \mathcal{N}(y|\mathrm{A}x(0), \sigma^2)q(x(0)|x(t))\,dx(0)$$
$$= \mathcal{N}\left(y|\mathrm{A}\hat{x}(x(t)), (\sigma_y^2 + r_t^2)\mathrm{I}\right) \tag{6}$$

where $p(x(0)|x(t))$ is not available in closed form and is approximated by a Gaussian $q(x(0)|x(t)) \triangleq \mathcal{N}(x(0)|\hat{x}(x(t)), r_t^2 \mathrm{I})$ with mean given by Tweedie's formula [18, 57] $\hat{x}(x(t)) \triangleq \mathbb{E}[x(0)|x(t)] = (x(t) + \sigma_t^2 \nabla \log p_t(x(t)))/\mu_t$ and variance $\mathrm{Var}[x(0)|x(t)] \approx r_t^2$, where $r_t$ is the guidance schedule and is chosen as a monotonically increasing function [19, 53, 60, 65].

## 3 PDE surrogates for forecasting and data assimilation

In this section, we discuss the design space of diffusion models, outlined in Tab. 1, for tackling forecasting and DA in the context of PDE modelling.

**Problem setting.** We denote by $\mathcal{P} : \mathcal{U} \to \mathcal{U}$ a differential operator taking functions $u : \mathbb{R}_+ \times \mathcal{Z} \to \mathcal{X} \in \mathcal{U}$ as input. Along with some initial $u(0, \cdot) = u_0$ and boundary $u(\tau, \partial\mathcal{Z}) = f$ conditions, these define a partial differential equation (PDE) $\partial_\tau u = \mathcal{P}(u; \alpha) = \mathcal{P}\left(\partial_z u, \partial_z^2 u, \ldots; \alpha\right)$ with coefficients $\alpha$ and where $\partial_\tau u$ is a shorthand for the partial derivative $\partial u/\partial\tau$, and $\partial_z u, \partial_z^2 u$ denote the partial

derivatives $\partial u/\partial z, \partial^2 u/\partial z^2$, respectively. We assume access to some accurate numerical solutions from conventional solvers, which are discretised both in space and time. Let's denote such trajectories by $x_{1:L} = (x_1, \ldots, x_L) \in \mathbb{R}^{D \times L} \sim p_0$ with $D$ and $L$ being the size of the discretised space and time domains, respectively.

We are generally interested in the problem of generating realistic PDE trajectories given observations, i.e. sampling from conditionals $p(x_{1:L}|y)$. Specifically, we consider tasks that involve conditioning on initial states, as well as on sparse space-time observations, reflective of some real-life scenarios such as weather prediction. Thus, we focus on two types of problems, and the combination thereof: (a) *forecasting*, which involves predicting future states $x_{H:L}$ given a length-$H$ past history $x_{1:H-1}$, i.e. sampling from $p(x_{H:L}|x_{1:H-1})$; (b) *data assimilation* (DA), which is concerned with inferring the ground state of a dynamical system given some sparse observed states $x_o$, i.e. sampling from $p(x_{1:L}|x_o)$. Then, the goal is to learn a flexible PDE surrogate, i.e. a machine learning model, that, unlike most numerical solvers [43], is able to solve the underlying PDE while accounting for various initial conditions and sparse observations.

## 3.1 Learning the score

Learning a diffusion model over the full joint distribution $p(x_{1:L}(t))$ can become prohibitively expensive for long trajectories, as it requires a score network taking the full sequence as input— $\mathbf{s}_\theta(t, x_{1:L}(t))$, with memory footprint scaling linearly with the length $L$ of the sequence.

One approach to alleviate this, as suggested by Rozet and Louppe [60], is to assume a Markov structure of order $k$ such that the joint over data trajectories can be factorised into a series of conditionals $p(x_{1:L}) = p(x_1)p(x_2|x_1) \ldots p(x_{k+1}|x_{1:k}) \prod_{i=k+2}^{L} p(x_i|x_{i-k:i-1})$. Here we are omitting the time dependency $x_{1:L} = x_{1:L}(0)$ for clarity's sake. The score w.r.t. $x_i$ can be written as

$$\nabla_{x_i} \log p(x_{1:L}) = \nabla_{x_i} \log p(x_i|x_{i-k:i-1}) + \sum_{j=i+1}^{i+k} \nabla_{x_i} \log p(x_j|x_{j-k:j-1})$$

$$= \nabla_{x_i} \log p(x_i, x_{i+1:i+k}|x_{i-k:i-1}) = \nabla_{x_i} \log p(x_{i-k:i+k}) \quad (7)$$

with $k + 1 \le i \le L - k - 1$, whilst formulas for $i \le k$ and $i \ge L - k$ can be found in App. B.1. In the case of modelling the conditional $p(x_{1:L}|y)$, we use the same Markov assumption, such that learning $\nabla_{x_i} \log p(x_{1:L}|y)$ reduces to learning a local score $\nabla_{x_i} \log p(x_{i-k:i+k}|y)$. The rest of Sec. 3.1 describes learning the local score of the joint distribution, but the same reasoning can easily be applied to fitting the local score of the conditional distribution.

When noising the sequence with the SDE (1), there is no guarantee that $x_{1:L}(t)$ will still be $k$-order Markov. However, we still assume that $x_{1:L}(t)$ is Markov but with a larger order $k' > k$, as motivated in Rozet and Louppe [60, Sec 3.1]. To simplify notations we use $k' = k$ in the rest of the paper. Consequently, instead of learning the entire joint score at once $\nabla_{x_{1:L}(t)} \log p(x_{1:L}(t))$, we only need to learn the *local* scores $\mathbf{s}_\theta(t, x_{i-k:i+k}(t)) \approx \nabla_{x_{i-k:i+k}(t)} \log p_t(x_{i-k:i+k}(t))$ with a window size of $2k + 1$. The main benefit is that the network only takes as input sequences of size $2k + 1$ instead of $L$, with $k << L$. The local score network $\mathbf{s}_\theta$ is trained by minimising the following DSM loss $\mathcal{L}(\theta)$:

$$\mathbb{E} \left\| \mathbf{s}_\theta(t, x_{i-k:i+k}(t)) - \nabla \log p_{t|0}(x_{i-k:i+k}(t)|x_{i-k:i+k}) \right\|^2$$

where the expectation is taken over the joint $i \sim \mathcal{U}([k+1, L-k]), t \sim \mathcal{U}([0, 1]), x_{i-k:i+k} \sim p_0$ and $x_{i-k:i+k}(t) \sim p_{t|0}(\cdot|x_{i-k:i+k})$ given by the noising process.

## 3.2 Conditioning

**Reconstruction guidance.** We now describe how to tackle forecasting and DA by sampling from conditionals $p(x_{1:L}|x_o)$—instead of the prior $p(x_{1:L})$—with the trained joint local score $\mathbf{s}_\theta(t, x_{i-k:i+k}(t))$ described in Sec. 3.1. For both tasks, the conditioning information $y = x_o \in \mathbb{R}^O$ is a measurement of a subset of variables in the space-time domain, i.e. $x_o = \mathcal{A}(x) = A \operatorname{vec}(x) + \eta$ with $\operatorname{vec}(x) \in \mathbb{R}^N$ the space-time vectorised trajectory, $\eta \sim \mathcal{N}(0, \sigma_y^2 I)$ and a masking matrix $A = \{0, 1\}^{O \times N}$ where rows indicate observed variables with 1. Plugging this in (6) and computing the score we get the following reconstruction guidance term

$$\nabla \log p(x_o|x(t)) \approx \frac{1}{r_t^2 + \sigma_y^2}(y - A\hat{x}(x(t)))^\top A \frac{\partial \hat{x}(x(t))}{\partial x(t)}. \quad (8)$$

Summing up this guidance with the trained score over the prior as in (5), we get an approximation of the conditional score $\nabla \log p(x_{1:L}(t)|x_o)$ and can thus generate conditional trajectories by simulating the conditional denoising process.

**Conditioning via architecture.** Alternatively, instead of learning a score for the joint distribution and conditioning a posteriori, one can directly learn an approximation to the conditional score. Such a score network $\mathbf{s}_\theta(t, x_{i:i+P}(t)|x_{i-C:i})$ is trained to fit the conditional score given the states $x_{i-C:i}$ by minimising a DSM loss akin to (3). Here, we denote the number of observed states by $C$, while $P$ stands for the length of the predictive horizon. Most of the existing works [35, 75] propose models, which we refer to as *amortised* models, where the number of conditioning frames $C$ is fixed and set before training. Thus, a separate model for each combination of $P$ and $C$ needs to be trained. To overcome this issue, similarly to [28], we propose a *universal amortised* model, which allows to use any $P$ and $C$, such that $P + C = 2k + 1$, during sampling. In particular, during training, instead of determining $C$ beforehand, we sample $C \sim \text{Uniform}(\{0, \ldots, 2k\})$ for each batch and train the model to fit the conditional score $\mathbf{s}_\theta(t, x_{i-C:i+P}(t)|x_{i-C:i})$, thus amortising over $C$ as well.

The conditioning on the previous states is achieved by feeding them to the score network as separate input channels. The conditioning dimension is fixed to the chosen window size of $2k+1$. To indicate whether a particular channel is present we add binary masks of size $2k+1$. Even though we condition on $x_{i-C:i}$, the score for the whole window size $x_{i-C:i+P}(t)$ is predicted. In our experiments, we found that this training scheme works well for our tasks, although another parameterization as in [28], where the score is only predicted for the masked variables, could be used.

**Architecture conditioning and reconstruction guidance.** In the universal amortised model, conditioning on previous states happens naturally via the model architecture. However, this strategy is not practical for conditioning on partial observations (e.g. sparse space-time observations as in DA), as a new model needs to be trained every time new conditioning variables are introduced. To avoid this, we propose to simultaneously use reconstruction guidance (as estimated in (8)) to condition on other variables, as well as keep the conditioning on previous states through model architecture.

### 3.3 Rollout strategies

In this section, we describe sampling approaches that can be used to generate PDE rollouts, depending on the choice of local score and conditioning. We provide in App. K pseudocode for each rollout strategy.

**All-at-once (AAO).** With a trained joint score $\mathbf{s}_\theta(t, x_{i-k:i+k}(t))$ we can generate entire trajectories $x_{1:L}$ in one go, as in Rozet and Louppe [60]. The full score $\mathbf{s}_\theta(t, x_{1:L}(t)) \approx \nabla \log p_t(x_{1:L}(t))$ is reconstructed from the local scores, as shown in App. B.1 and summarised in Rozet and Louppe [60, Alg 2.]. The $i$th component is given by $\mathbf{s}_i = \mathbf{s}_\theta(t, x_{i-k:i+k}(t))[k+1]$ which is the central output of the score network, except for the start and the end of the sequence for which the score is given by $\mathbf{s}_{1:k} = \mathbf{s}_\theta(t, x_{1:2k+1}(t))[: k+1]$ and $\mathbf{s}_{L-k:L} = \mathbf{s}_\theta(t, x_{L-2k:L}(t))[k+1 :]$, respectively. If we observe some values $x_o \sim \mathcal{N}(\cdot|Ax_{1:L}, \sigma_y^2 I)$, the conditional sampling is done by summing the full score $\mathbf{s}_\theta(t, x_{1:L}(t))$ with the guidance term $\nabla \log p(x_o|x_{1:L}(t))$ estimated with (8).

**Autoregressive (AR).** Let us assume that we want to sample $x_{C+1:L}$ given $x_{1:C}$ initial states. An alternative approach to AAO is to factor the forecasting prediction problem as $p(x_{1:L}|x_{1:C}) = \prod_i p(x_i|x_{i-C:i-1})$ with $C \leq k$. As suggested in Ho et al. [27], it then follows that we can sample the full trajectory via ancestor sampling by iteratively sampling each conditional diffusion process. More generally, instead of sampling one state at a time, we can autoregressively generate $P$ states, conditioning on the previous $C$ ones, such that $P + C = 2k + 1$. This procedure can be used both by the joint and the amortised conditional score. DA can similarly be tackled by further conditioning on additional observations appearing within the predictive horizon $P$ at each AR step of the rollout.

**Scalability.** Both sampling approaches involve a different number of neural function evaluations (NFEs). Assuming $p$ predictor steps and $c$ corrector steps to simulate (4), AAO sampling requires $(1 + c)p$ NFEs. In contrast, the AR scheme is more computationally intensive as each autoregressive step also costs $(1 + c)p$ NFEs but with $\left(1 + \frac{L-(2k+1)}{P}\right)$ AR steps. We stress, though, that this is

assuming the full score network evaluation $\mathbf{s}_{1:L} = \mathbf{s}_\theta(t, x_{1:L}(t))$ is parallelised. In practice, due to memory constraints, the full score in AAO would have to be sequentially computed. Assuming only one local score evaluation fits in memory, it would require as many NFEs as AR steps. What's more, as discussed in the next paragraph, AAO in practice typically requires more corrector steps. See App. F.1 for further discussion on scalability.

**Modelling capacity of the joint score.** As highlighted in Tab. 1, when using a joint score, both AAO and AR rollout strategies can be used. We hypothesise there are two main reasons why AR outperforms AAO. First, for the same score network with an input window size of $2k + 1$, the AAO model has a Markov order $k$ whilst the AR model has an effective order of $2k$. Indeed, as shown in (7), a process of Markov order $k$ yields a score for which each component depends on $2k + 1$ inputs. Yet parameterising a score network taking $2k + 1$ inputs, means that the AR model would condition on the previous $2k$ states at each step. This twice greater Markov order implies that the AR approach is able to model a strictly larger class of data processes than the AAO sampling. See App. B.1 for further discussion on this.

Second, with AAO sampling the denoising process must ensure the start of the generated sequence agrees with the end of the sequence. This requires information to be propagated between the two ends of the sequence, but at each step of denoising, the score network has a limited receptive field of $2k + 1$, limiting information propagation. Langevin corrector steps allow extra information flow, but come with additional computational cost. In contrast, in AR sampling, there is no limit in information flowing forward due to the nature of the autoregressive scheme.

## 4 Related work

In the following, we present works on diffusion models for PDE modelling since these are the models we focus on in this paper. We present an extended related work section in App. C, where we broadly discuss ML-based and classical solver-based techniques for tackling PDE modelling.

**Diffusion models for PDEs.** Recently, several works have leveraged diffusion models for solving PDEs, with a particular focus on fluid dynamics. Amortising the score network on the initial state, Kohl et al. [35] introduced an autoregressive scheme, whilst Yang and Sommer [81] suggested to directly predict future states. Recently, Cachay et al. [9] proposed to unify the denoising time and the physical time to improve scalability. Lippe et al. [42] built upon neural operators with an iterative denoising refinement, particularly effective to better capture higher frequencies and enabling long rollouts. In contrast to the above-mentioned work, others have tackled DA and super-resolution tasks. Shu et al. [63] and Jacobsen et al. [32] suggested using the underlying PDE to enforce adherence to physical laws. Huang et al. [29] used an amortised model and inpainting techniques [46] for conditioning at inference time to perform DA on weather data. Rozet and Louppe [60] decomposed the score of long trajectory into a series of local scores over short segments to be able to work with flexible lengths and to dramatically improve scalability w.r.t. memory use. Concurrently to our work, Qu et al. [55] extended this approach to latent diffusion models to perform DA on ERA5 weather data [25]. In this work, we show that such a trained score network can alternatively be sampled autoregressively, which is guaranteed to lead to a higher Markov order, and empirically produce accurate long-range forecasts. Concurrently, Ruhe et al. [61] proposed to use a local score, not justified by any Markov assumption, together with a frame-dependent noising process for video and fluid dynamics generation. Diffusion models for PDE modelling are indeed closely related to other sequence modelling tasks, such as video generation and indeed our universal amortised approach is influenced by Höppe et al. [28], Voleti et al. [75].

## 5 Experimental results

We study the performance of the diffusion-based models proposed in Tab. 1 on three different described in more detail below. The code to reproduce the experiments is publicly available at https://github.com/cambridge-mlg/pdediff.

**Data.** In this work, we consider the 1D Kuramoto-Sivashinsky (KS) and the 2D Kolmogorov flow equations, with the latter being a variant of the incompressible Navier-Stokes flow. KS is a

fourth-order nonlinear 1D PDE describing flame fronts and solidification dynamics. The position of the flame $u$ evolves as $\partial_\tau u + u\partial_x u + \partial_x^2 u + \nu\partial_x^4 u = 0$ where $\nu > 0$ is the viscosity. The equation is solved on a periodic domain with 256 points for the space discretisation and timestep $\Delta\tau = 0.2$. Training trajectories are of length $140\Delta\tau$, while the length of validation and testing ones is set to $640\Delta\tau$. The Kolmogorov flow is a 2D PDE that describes the dynamics of an incompressible fluid. The evolution of the PDE is given by $\partial_\tau \mathbf{u} + \mathbf{u}\cdot\nabla\mathbf{u} - \nu\nabla^2\mathbf{u} + \frac{1}{\rho}\nabla p - f = 0$ and $\nabla\cdot\mathbf{u} = 0$, where $\mathbf{u}$ represents the velocity field, $\nu$ the viscosity, $\rho$ the fluid density, $p$ the pressure, and $f$ is the external forcing. The considered trajectories have 64 states with $64 \times 64$ resolution and $\Delta\tau = 0.2$. The evaluation metrics are measured with respect to the scalar vorticity field $\omega = \partial_x u_y - \partial_y u_x$. We focus on these since their dynamics are challenging, and have been investigated in prior work [17, 42]. We refer to App. D for more details on the data generation. In addition, in App. F.1 we present results for the simpler 1D Burgers' equation.

**Models.**  For forecasting and DA, we train a single local score network on contiguous segments randomly sampled from training trajectories. We use a window 9 model for KS and a window 5 model for Kolmogorov, as these settings give a good trade-off between performance and memory requirements. We show in F.3 that increasing the window size in KS does lead to improved performance in forecasting, but the gains are relatively small. We parameterise the score network $\mathbf{s}_\theta$ with a modern U-Net architecture [22, 59] with residual connections and layer normalization. For sampling, we use the DPM solver [44] with 128 evenly spaced discretisation steps. As a final step, we return the posterior mean over the noise free data via Tweedie's formula. For the guidance schedule we set $r_t^2 = \gamma\sigma_t^2/\mu_t^2$ and tune $\gamma$ via grid search. We refer to App. D.4 for more details.

**Evaluation metrics.**  We compute two per time step metrics—mean squared error $\texttt{MSE}_{1:L} = \mathbb{E}[(x_{1:L} - \hat{x}_{1:L})^2]$ and Pearson correlation $\rho_{1:L} = \frac{\text{Cov}(x_{1:L}, \hat{x}_{1:L})}{\text{Var}(x_{1:L})\text{Var}(\hat{x}_{1:L})}$ between model samples $\hat{x}_{1:L} \sim p_\theta$ and ground truth trajectories $x_{1:L} \sim p_0$. We measure sample accuracy through $\texttt{RMSD} = \sqrt{\frac{1}{L}\sum_{l=1}^{L}\texttt{MSE}_l}$ and high correlation time $t_{\max} = l_{\max} \times \Delta t$ with $l_{\max} = \arg\max_{l\in 1:L}\{\rho_l > 0.8\}$.

### 5.1 Forecasting

As mentioned in Sec. 3, forecasting is a crucial component of the combined task we consider, so we investigate the ability of trained diffusion models to sample long rollouts. The models are evaluated on 128 and 50 test trajectories for KS and Kolmogorov, respectively. To understand the current state of score-based models for PDE forecasting we benchmark them against other ML-based models, including the state-of-the-art PDE-Refiner [42] and a MSE-trained U-Net and Fourier Neural Operator (FNO) [39]. We follow the setup in [42] for the architectures and training settings of the baselines. For the diffusion models and the MSE-trained U-Net baseline, we experiment with both an architecture inspired from [60] and one inspired from [42] and report the best results. The performance of both neural network architectures considered can be found in Tab. 5. More details about the different architectures can be found in App. D.4.

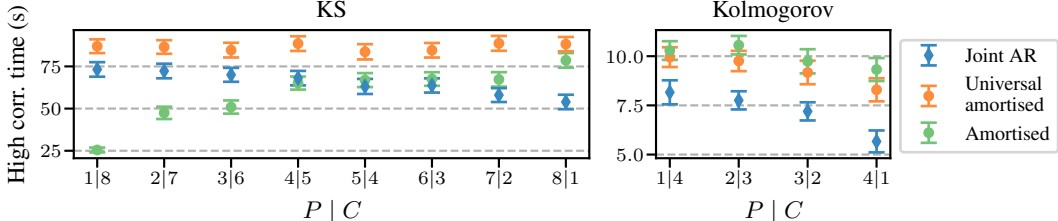

Figure 1: High correlation time (↑) on the KS (left) and Kolmogorov (right) datasets for different $P \mid C$ conditioning scenarios of the joint, amortised and universal amortised models, where $P$ indicates the number of generated states and $C$ the number of states conditioned upon. We show mean $\pm$ 3 standard errors, as computed on the test trajectories.

Fig. 1 shows the performance of the joint, amortised and universal amortised models for different conditioning scenarios $P \mid C$, with $P$ the number of states generated, and $C$ the number of states conditioned upon at each iteration. For the KS dataset, the universal amortised model outperforms the plain amortised one across all $P \mid C$; for Kolmogorov their performance is comparable (i.e. within 3

standard errors). For both datasets, the best (plain and universal) amortised models outperform the best joint AR model, with the exact best high correlation times shown in Fig. 2.

**Stability over history length.**    The most interesting comparison between the models lies in how they leverage past history in the context of forecasting. Previous work notes that longer history generally degrades the performance of amortised models [42]. We find that this behaviour is dataset-dependent—the observation holds for KS, whereas for Kolmogorov the plain amortised models with longer history (i.e. 1 | 4 and 2 | 3) tend to outperform the ones with shorter history (i.e. 3 | 2 and 4 | 1). Due to our novel training procedure for the universal amortised model (i.e. training over a variety of tasks), we obtain stable performance over all conditioning scenarios for the KS dataset. For Kolmogorov, the universal amortised model benefits from longer history, similarly to the plain amortised one. However, note that, unlike the plain amortised model, where a different model needs to be trained for each conditioning scenario, the universal amortised can tackle any scenario, allowing for a trade-off between accuracy and computational speed at sampling time. The joint model generally benefits from longer conditioning trajectories, but is also fairly stable for the majority of $P \mid C$ settings. We show more results in App. F.3.

For a complete comparison between the models in Tab. 1, we also perform AAO sampling for the joint model. However, as shown in App. F.1, the predicted samples rapidly diverge from the ground truth. Thus, the AR method of querying the joint model is crucial for forecasting.

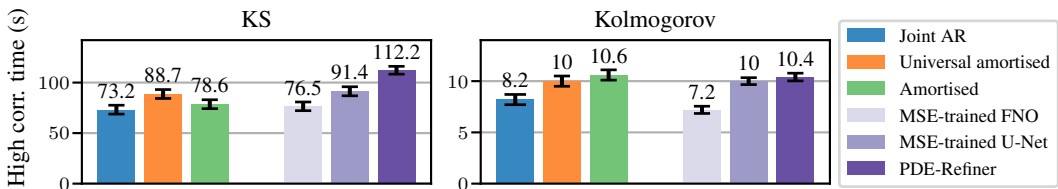

Figure 2: The best joint and amortised models compared against standard ML-based benchmarks for forecasting on the KS (left) and Kolmogorov (right) datasets. High correlation time (↑) is reported, showing the mean $\pm$ 3 standard errors, as computed on the test trajectories. We show in Tab. 5 the configurations for the best models.

**Benchmarks.**    In Fig. 2, we report the comparison against popular methods for forecasting. The universal amortised model performs similarly to the MSE-trained U-Net on both datasets, while the joint model is comparable to the MSE-trained FNO, performing on par with it for KS and slightly outperforming it for Kolmogorov. However, we stress that, unlike these benchmarks, which can only be straight-forwardly applied to forecasting, our models are significantly more flexible—they manage to achieve competitive performance with some of the MSE-trained benchmarks, yet can be applied to a wider variety of tasks (e.g. forecasting combined with DA).

**Additional results.**    For a more thorough understanding of the models' behaviour, we perform further analysis on the frequency spectra of the generated trajectories in App. I. We show that, as noted in previous work [42], the generated KS samples tend to approximate the low-frequency components well, but fail to capture the higher-frequency components, while the Kolmogorov samples generally show good agreement with the ground truth spectrum. We also investigate the long-term behaviour of our proposed models in App. J. Even when generating trajectories that are significantly longer than those used during training, the models are capable of generating physically-plausible states.

## 5.2   Offline data assimilation

In offline DA, we assume we have access to some sparse observations. The experimental setting goes as follows. We first choose a number of observed variables, and then uniformly sample the associated indices. We assume to always observe some full initial states. More details can be found in App. G. We repeat this for different numbers of values observed, varying the sparsity of observations, from approximately a third of all data to almost exclusively conditioning on the initial states. We compute the RMSD of the generated trajectories for six values of the proportion of observed data, evenly spaced on a log scale from $10^{-3}$ to $10^{-0.5}$. We show results for the joint AR model, joint AAO with 0 and 1 corrections (indicated between brackets), and the universal amortised model. To provide

a consistent comparison with the AAO approach from [60], all models use the U-Net architecture inspired by [60]. We tune the guidance strength $\gamma$, and for AR we only report the $P \mid C$ setting that gives the best trade-off between performance and computation time (see Fig. 28 for a comprehensive summary of performance depending on $\gamma$ and $P \mid C$). We show the results on 50 test trajectories for both KS and Kolmogorov. More results can be found in App. G.

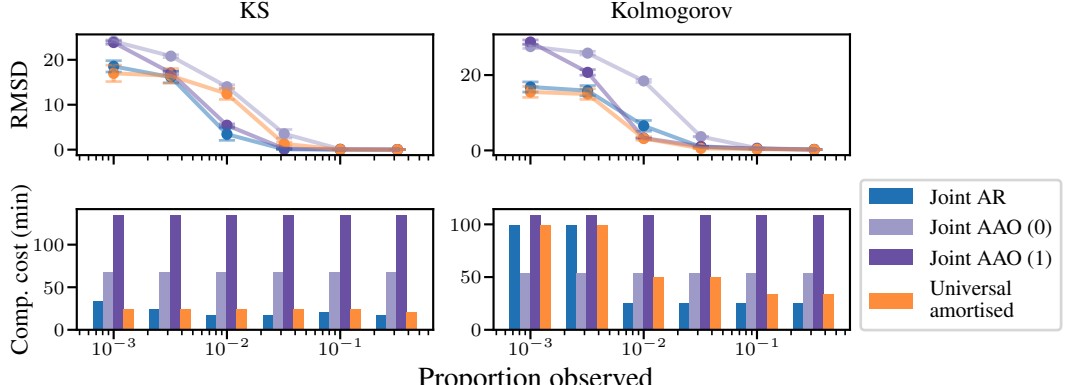

Figure 3: RMSD (mean $\pm$ 3 standard errors) for KS (left) and Kolmogorov (right) on the offline DA setting for varying sparsity levels (top), and the computational cost associated with each setting (bottom). The latter is the same for all sparsity settings for AAO, but differs for AR since it depends on the $P \mid C$ setting that was used. The $c$ in AAO $(c)$ refers to the number of corrector steps used.

**More efficient sampling.** We observe in Fig. 3 that AR sampling outperforms AAO in the sparse observation regimes, and performs on par with it in the dense observation regimes, provided corrections are used. However, Fig. 3 shows that corrections significantly increase the associated computational cost. We believe the difference in performance is mostly due to the limiting information propagation in the AAO scheme, combined with the lower effective Markov order of AAO vs AR ($k$ vs $2k$ for a local score network with a window of $2k + 1$), as discussed in Sec. 3.3. To further investigate this hypothesis, in App. G.2.1 we compare the results for a window 5 ($k = 2$) AR model with a window 9 ($k = 4$) AAO model for KS. These should, in theory, have the same effective Markov order. We observe that for the sparsest observation regime, there is still a gap between AR ($k = 2$) and AAO ($k = 4$), potentially because the high sparsity of observations make the information flow in the AAO scheme inefficient. In fact, this is consistent with the findings from forecasting, where we observe that the AAO strategy is unable to produce coherent trajectories if it does not have access to enough future observations. In contrast, for the other sparsity scenarios the performance of the two schemes becomes similar provided corrections are used. However, this comes with a significant increase in computational cost.

**Reconstruction guidance for universal amortised model.** As outlined in Sec. 3.2, we can combine the universal amortised model with reconstruction guidance for conditioning on sparse observations. Amortised models do not straight-forwardly handle DA — this can be seen in Fig. 29a, where we show that conditioning just through the architecture gives poor results in DA. In KS, the universal amortised model achieves lower RMSD for the very sparse observation scenarios ($10^{-3}$ proportion observed), and for the very dense ones ($10^{0.5}$ proportion observed). However, the joint model outperforms it in the middle ranges. In Kolmogorov, the universal amortised model slightly outperforms the joint for all sparsity settings, with the results being within error bars. These findings indicate that they are both viable choices for offline DA, with the best performing choice depending on the application.

**Interpolation baseline.** In App. G.3 we also provide a quantitative and qualitative comparison to a simple interpolation baseline, proving that the studied models manage to capture the underlying physical behaviour better than such simple baselines.

### 5.3 Online DA: combining forecasting and data assimilation

Inspired by real-life applications such as weather prediction, we compare the models in the online DA setting, which represents the combination of forecasting and DA. We assume that some sparse

and noisy variables are observed once every $s$ states (in blocks of length $s$), and whenever new observations arrive (i.e. at each DA step), we forecast the next $f$ states. At the first step, we perform forecasting assuming we have access to some randomly sampled indices, accounting for $10\%$ of the first $s$-length trajectory. At the second step (after $s$ time-steps), we refine this forecast with the newly observed set of variables. This repeats until we have the forecast for the entire length of the test trajectories. The performance is measured in terms of the mean RMSD between the forecasts and the ground truth (averaged over all the DA steps, each outputting a trajectory of length $f$). We set $s = 20$ and $f = 400$ for KS, and $s = 5$ and $f = 40$ for Kolmogorov, resulting in 13, and 6 DA steps, respectively. The setup is visualised in App. H, and the key findings from the results in Tab. 2 are

- AAO is unsuitable for tasks involving forecasting — As opposed to the offline setting, AAO fails at online DA, leading to a significantly higher RMSD on both datasets.
- The universal amortised model shows the best performance out of all the models, which is in line with the findings from the previous sections — it outperforms joint AR in forecasting and performs on par with it in DA.

Table 2: RMSD ($\downarrow$) (mean $\pm$ 3 standard errors over test trajectories) for online DA for the KS and Kolmogorov datasets. The $c$ in AAO ($c$) refers to the number of corrector steps used.

| DATA | Joint AR | Joint AAO (0) | Joint AAO (1) | Universal amortised |
|------|----------|---------------|---------------|---------------------|
| KS | $11.6 \pm 0.6$ | $25.3 \pm 0.1$ | $26.3 \pm 0.1$ | $7.7 \pm 0.6$ |
| Kolmogorov | $7.8 \pm 0.5$ | $27.4 \pm 0.2$ | $30.1 \pm 0.2$ | $4.9 \pm 0.4$ |

## 6 Discussion

In this work, we explore different ways to condition and sample diffusion models for forecasting and DA. In particular, we empirically demonstrate that diffusion models achieve comparable performance with MSE-trained models in forecasting, yet are significantly more flexible as they can effectively also tackle DA. We theoretically justify and empirically demonstrate the effectiveness of AR sampling for joint models, which is crucial for any task that contains a forecasting component. Moreover, we enhance the flexibility and robustness of amortised models by proposing a new training strategy which also amortises over the history length, and by combining them with reconstruction guidance.

**Limitations and future work.** Although achieving competitive results with MSE-trained baselines in forecasting, the models still lack state-of-the-art performance. Thus, they are most practical in settings where flexible conditioning is needed, rather than a standard forecasting task. Although we have explored different solvers and number of discretisation steps, the autoregressive sampling strategy requires simulating a denoising process at each step, which is computationally intensive. We believe that this can be further improved, perhaps by reusing previous sampled states. Additionally, the guided sampling strategy requires tuning the guidance strength, yet we believe there exist some simple heuristics that are able to decrease the sensitivity of this hyperparameter. Finally, we are interested in investigating how the PDE characteristics (i.e. spatial / time discretisation, data volume, frequency spectrum, etc.) influence the behaviour of the diffusion-based models.

## Acknowledgements

AS, RET and EM are supported by an EPSRC Prosperity Partnership EP/T005386/1 between the EPSRC, Microsoft Research and the University of Cambridge. CD is supported by the Cambridge Trust Scholarship. FB is supported by the Innovation Fund Denmark (0175-00014B) and the Novo Nordisk Foundation through the Center for Basic Machine Learning Research in Life Science (NNF20OC0062606) and NNF20OC0065611. JMHL acknowledges support from a Turing AI Fellowship under grant EP/V023756/1. RET is supported by the EPSRC Probabilistic AI Hub (EP/Y028783/1) and gifts from Google, Amazon, ARM, and Improbable. We thank the anonymous reviewers for their key suggestions and insightful questions that helped improved the quality of the paper.

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

# On conditional diffusion models for PDE simulations (Appendix)

## A  Organisation of the supplementary

In this supplementary, we first motivate in App. B the choice made regarding the (local) score network given the Markov assumption on the sequences. We then provide more reference to related work in App. C, followed by a description of the data generating process in App. D for the datasets used in the experiments in Sec. 5. We also provide further implementation details. After, in App. E, we assess the ability of the all-at-once sampling strategy—which relies on reconstructing the full score from the local ones—to perform accurate forecast, and in particular show the crucial role of correction steps in the discretisation. We then provide more results on forecasting in App. F, including a study of the influence of the guidance strength hyperparameter $\gamma$ and of the conditioning scenario, a comparison with widely used benchmarks in forecasting, such as climatology and persistence, and results for different U-Net architectures. In App. G we show more experimental evidence on the offline DA task, including the influence of the guidance strength and of the conditioning scenarios, as well as perform a comparison between an AR model and an AAO model with twice higher Markov order. We also include a comparison with an interpolation baseline. In App. H we provide further experimental evidence for the online DA task for a range of conditioning scenarios for the joint and amortised models. In App. I we provide frequency spectra of the trajectories generated by our methods and the forecasting baselines, and in App. J we analyse the long-term behaviour of the diffusion models. Finally, we provide pseudo-code in App. K.

## B  Markov score

In this section, we look at the structure of the score induced by a Markov time series. In particular, we lay out different ways to parameterise and train this score with a neural network. In what follows we assume an AR-1 Markov model for the sake of clarity, but the reasoning developed trivially generalises to AR-$k$ Markov models. Assuming an AR-1 Markov model, we have that the joint density factorise as

$$p(x_{1:L}) = p(x_1) \prod_{t=2}^{T} p(x_i|x_{i-1}). \tag{9}$$

### B.1  Window $2k+1$:

Looking at the score of (9) for intermediary states we have

$$
\begin{aligned}
\nabla_{x_i} \log p(x_{1:L}) &= \nabla_{x_i} \log p(x_{i+1}|x_i) + \nabla_{x_i} \log p(x_i|x_{i-1}) \qquad (10) \\
&= \nabla_{x_i} \log p(x_{i+1}, x_i|x_{i-1}) \\
&= \nabla_{x_i} \log p(x_{i+1}, x_i|x_{i-1}) + \underbrace{\nabla_{x_i} \log p(x_{i-1})}_{0} \\
&= \nabla_{x_i} \log p(x_{i+1}, x_i, x_{i-1}) \\
&= \nabla_{x_i} \log p(x_{i-1}, x_i, x_{i+1}). \qquad (11)
\end{aligned}
$$

Similarly, for the first state, we get

$$
\begin{aligned}
\nabla_{x_1} \log p(x_{1:L}) &= \nabla_{x_1} \log p(x_1) + \nabla_{x_1} \log p(x_2|x_1) \\
&= \nabla_{x_1} \log p(x_1, x_2) \\
&= \nabla_{x_1} \log p(x_1, x_2) + \underbrace{\nabla_{x_1} \log p(x_3|x_2)}_{0} \\
&= \nabla_{x_1} \log p(x_1, x_2) + \nabla_{x_1} \log \underbrace{p(x_3|x_2, x_1)}_{\text{since } = p(x_3|x_2)} \\
&= \nabla_{x_1} \log p(x_1, x_2, x_3), \quad (12)
\end{aligned}
$$

and for the last state

$$
\begin{aligned}
\nabla_{x_L} \log p(x_{1:L}) &= \nabla_{x_L} \log p(x_L|x_{L-1}) \quad (13) \\
&= \nabla_{x_L} \log \underbrace{p(x_L|x_{L-1}, x_{L-2})}_{\text{since } = p(x_L|x_{L-1})} \\
&= \nabla_{x_L} \log p(x_L|x_{L-1}, x_{L-2}) + \underbrace{\nabla_{x_L} \log p(x_{L-1}, x_{L-2})}_{0} \\
&= \nabla_{x_L} \log p(x_L, x_{L-1}, x_{L-2}) \\
&= \nabla_{x_L} \log p(x_{L-2}, x_{L-1}, x_L). \quad (14)
\end{aligned}
$$

**Noised sequence.** Denoising diffusion requires learning the score of *noised* states. The sequence $x_{1:L}$ is noised with the process defined in (1) for an amount of time $t$: $x_{1:L}(t)|x_{1:L} \sim p_{t|0}$. With $x_{1:L}$ an AR-$k$ Markov model there is no guarantee that $x_{1:L}(t)$ is still $k$-Markov. Yet, we can assume that $x_{1:L}(t)$ is an AR-$k'$ Markov model where $k' > k$, as in Rozet and Louppe [60]. Although it may seem in contradiction with the previous statement, in what follows we focus on the AR-1 assumption since derivations and the general reasoning is much easier to follow, yet reiterate that this generalises to the order $k$.

**Local score.** Eqs. (11), (12) and (14) suggest training a score network to fit $\mathbf{s}_\theta(t, x_{i-1}(t), x_i(t), x_{i+1}(t)) \approx \nabla \log p_t(x_{i-1}(t), x_i(t), x_{i+1}(t))$ by sampling tuples $x_{i-1}, x_i, x_{i+1} \sim p_0$ and noising them.

**Full score.** The first, intermediary and last elements of the score are then respectively given by

- $\nabla_{x_1(t)} \log p_t(x_{1:L}(t)) \approx \mathbf{s}_\theta(t, x_1(t), x_2(t), x_3(t))[1]$
- $\nabla_{x_i(t)} \log p_t(x_{1:L}(t)) \approx \mathbf{s}_\theta(t, x_{i-1}(t), x_i(t), x_{i+1}(t))[2]$
- $\nabla_{x_L(t)} \log p_t(x_{1:L}(t)) \approx \mathbf{s}_\theta(t, x_{L-2}(t), x_{L-1}(t), x_L(t))[3]$

**Joint sampling.** The full score over the sequence $x_{1:L}$ can be reconstructed via the trained local score network taking segments of length 3 as input. This allows for sampling from the joint. All of the above generalise to AR-$k$ models, where the local score network takes a segment of length $2k + 1$ as input.

**Guided AR.** Alternatively for forecasting, having learnt a local score network $\mathbf{s}_\theta(t, x_{i-1}(t), x_i(t), x_{i+1}(t))$, i.e. modelling joints $p(x_{i-1}, x_i, x_{i+1})$, we can condition on $x_{i-1}, x_i$ to sample $x_{i+1}$ leveraging reconstruction guidance—as described in Sec. 3.3. Effectively, this induces an AR-2—instead of AR-1— Markov model, or in general, an AR-2$k$—instead of $k$—Markov model. As such, this score parameterisation takes more input than strictly necessary to satisfy the $k$ Markov assumption.

**Amortised AR.** Obviously we note that, looking at (10) and (13), one could learn a conditional score network amortised on the previous value such that $\mathbf{s}_\theta(t, x_i(t)|x_{i-1}(0)) \approx \nabla_{x_i(t)} \log p(x_i(t)|x_{i-1}(0))$. This unsurprisingly allows forecasting with autoregressive rollout as in Sec. 3.3. Additionally learning a model over $p(x_1)$ and summing pairwise the scores as in (10) would also allow for joint sampling.

## B.2 Window $k+1$:

In App. B.1, we started with an AR-$k$ model, but ended up with learning a local score taking $2k+1$ inputs. Below we derive an alternative score parameterisation. Still assuming an AR-1 model (9), we have that the score for intermediary states can be expressed as

$$
\begin{aligned}
\nabla_{x_i} \log p(x_{1:L}) &= \nabla_{x_i} \log p(x_{i+1}|x_i) + \nabla_{x_i} \log p(x_i|x_{i-1}) \\
&= \underbrace{\nabla_{x_i} \log p(x_i|x_{i+1}) - \nabla_{x_i} \log p(x_i)}_{\text{via Bayes' rule}} + \nabla_{x_i} \log p(x_i|x_{i-1}) + \underbrace{\nabla_{x_i} \log p(x_{i-1})}_{0} \\
&= \nabla_{x_i} \log p(x_i|x_{i+1}) + \underbrace{\nabla_{x_i} \log p(x_{i+1})}_{0} - \nabla_{x_i} \log p(x_i) + \nabla_{x_i} \log p(x_i, x_{i-1}) \\
&= \nabla_{x_i} \log p(x_i, x_{i+1}) - \nabla_{x_i} \log p(x_i) + \nabla_{x_i} \log p(x_i, x_{i-1}).
\end{aligned}
\tag{15}
$$

**Local score.** From (15), we notice that we could learn a pairwise score network such that $\tilde{\mathbf{s}}_\theta(t, x_{i-1}(t), x_i(t)) \approx \nabla_{x_i(t)} \log p(x_{i-1}(t), x_i(t))$, and $\tilde{\mathbf{s}}_\theta(t, x_i(t)) \approx \nabla_{x_i(t)} \log p(x_i(t))$, trained by sampling tuples $x_{i-1}, x_i, x_{i+1} \sim p_0$ and noising them. Note that this requires 3 calls to the local score network instead of 1 as in App. B.1.

**For $k > 1$.**

$$
\begin{aligned}
\nabla_{x_i} \log p(x_{1:L}) &= \nabla_{x_i} \log p(x_{i-k}, \ldots, x_{i-1}, x_i, x_{i+1}, \ldots, x_{i+k}) \\
&= \nabla_{x_i} \log p(x_i|x_{i-k:i-1}) + \sum_{j=i+1}^{i+k} \nabla_{x_i} \log p(x_j|x_{j-k:j-1}) \\
&= \nabla_{x_i} \log p(x_{i-k:i}) + \sum_{j=i+1}^{i+k} \nabla_{x_i} \log p(x_{j-k:j}) - \sum_{m=i}^{i+k-1} \nabla_{x_i} \log p(x_{m-k+1:m}) \\
&= \sum_{j=i}^{i+k} \nabla_{x_i} \log p(x_{j-k:j}) - \sum_{m=i}^{i+k-1} \nabla_{x_i} \log p(x_{m-k+1:m})
\end{aligned}
\tag{16}
$$

From (16), we obtained a $2k+1$ window size model, while having the scores with window size of $k$ and $k+1$. This would require roughly twice more forward passes during AAO sampling, but will double the effective window size.

**Full score.** The first, intermediary and last elements of the score are then respectively given by

$$
\begin{aligned}
\nabla_{x_1(t)} \log p(x_{1:L}(t)) &= \nabla_{x_1(t)} \log p(x_1(t)) + \nabla_{x_1(t)} \log p(x_2(t)|x_1(t)) \\
&= \nabla_{x_1(t)} \log p(x_1(t), x_2(t)) \\
&\triangleq \tilde{\mathbf{s}}_\theta(t, x_1(t), x_2(t))[1].
\end{aligned}
$$

$$
\begin{aligned}
\nabla_{x_i(t)} \log p(x_{1:L}(t)) &= \mathbf{s}_\theta(t, x_{i-1}(t), x_i(t), x_{i+1}(t)) \\
&\triangleq \tilde{\mathbf{s}}_\theta(t, x_i(t), x_{i+1}(t))[1] + \tilde{\mathbf{s}}_\theta(t, x_{i-1}(t), x_i(t))[2] - \tilde{\mathbf{s}}_\theta(t, x_i(t)).
\end{aligned}
$$

$$
\begin{aligned}
\nabla_{x_L(t)} \log p(x_{1:L}(t)) &= \nabla_{x_L(t)} \log p(x_L(t)|x_{L-1}(t)) \\
&= \nabla_{x_L(t)} \log p(x_L(t), x_{L-1}(t)) \\
&= \tilde{\mathbf{s}}_\theta(t, x_{L-1}(t), x_L(t))[2].
\end{aligned}
$$

**Joint sampling.** The full score over $x_{1:L}(t)$ could therefore be reconstructed from these, enabling joint sampling.

**AR.** Additionally, autoregressive forecasting could be achieved with guidance by iterating over pairs $(x_{i-1}, x_i)$, thus using a local score of window size $k+1$, in contrast to App. B.1 which requires an inflated window of size $2k+1$.

## C  More related work

In Sec. 4, we presented relevant works that focus on machine learning models and more specifically diffusion models for PDE modelling. Our work builds upon different conditional generation protocols used for conditional sampling in diffusion models. Sec. 2 presents the techniques we considered for our models. In the following, we summarise all the available methods for building a conditional diffusion model.

**Conditional diffusion models.**  The development of conditional generation methods for diffusion models is an active area of research, whereby a principled approach is to frame this as the problem of simulating a conditional denoising process which converges to the conditional of interest when fully denoised [68, 65, 12]. One approach is to leverage Bayes's rule, and combine a pretrained score network, with an estimate of the conditional over observation given noised data—referred to as a *guidance* term [13, 48, 66, 65, 27]. Several refinements over the estimation of this guidance have been suggested [60, 19]. Another line of work has focused on 'inpainting' tasks—problems where the observations lie in a subspace of the diffused space—and suggested 'freezing' the conditioning data when sampling [68, 72, 46, 47]. Lastly, another popular approach, sometimes referred to as *classifier-free guidance*, is to directly learn the conditional score *amortised* over the observation of interest [26, 79, 71]. This approach requires having access at training time to pairs of data and observation.

**Machine learning models for PDEs.**  One class of ML-based methods that can learn to approximately solve PDEs given some initial state assumes a fixed finite-dimensional spatial discretisation (i.e. mesh), typically a regular grid, and, leveraging convolutional neural networks, learns to map initial values of the grid to future states [20, 83, 5]. These are data-driven methods, which are trained on trajectories generated by conventional solvers, typically at fine resolution but then downscaled.

Alternatively, mesh-free approaches learn infinite-dimensional operators via neural networks, coined *neural operators* [2, 36, 45, 51, 49, 6]. These methods are able to map parameters along with an initial state to a set of neural network parameters that can then be used for simulation at a different discretisation. Neural operators typically achieve this by leveraging the Fourier transform and working in the spectral domain  [38, 21, 52]. Fourier neural operators (FNOs) can be used as the backbone of the diffusion model, but U-Net-based [59] models have been shown to outperform them [22].

**Classical solvers for PDEs.**  Conventional numerical methods, e.g. finite element methods [43], are commonly used to solve time-dependent PDEs in practice [1]. In contrast to ML-based methods, numerical methods have the advantage of providing theoretical guarantees of convergence and bounds on error, which depend on time and space discretisation. However, many real-world problems, e.g. fluid and weather modelling, are described by complex time-dependent PDEs, and thus to obtain accurate and reliable solutions, the space and time discretisations have to be very small [23]. As such, in this case, the conventional methods require significant computational and time resources. To tackle this issue, several recent works [33, 69] attempted to speed up numerical solvers using automatic differentiation and hardware accelerators (GPU/TPU).

While conventional numerical methods have been intensively studied and applied in forecasting, they are not suitable for solving inverse problems out-of-the-box. To overcome this, several works [31, 41] propose numerical approaches specifically developed for certain PDEs, while  Arridge et al. [3] provides an overview on data-driven, regularisation-based solutions to inverse problems. Moreover, unlike diffusion models, numerical solvers do not offer a probabilistic treatment, which is crucial in certain applications of PDE modelling. For example, in weather and climate modelling, being able to predict a range of probable weather scenarios is key in decision-making, such as issuing public hazards warnings and planning renewable energy use [54].

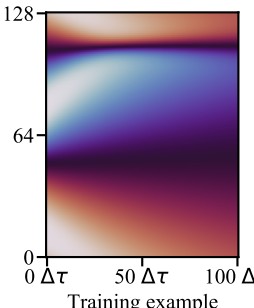 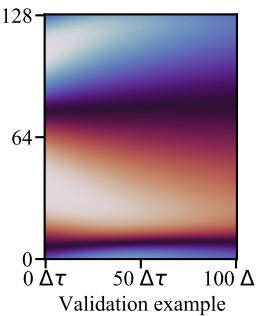 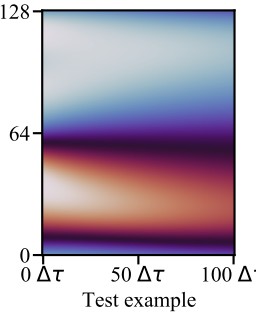

Training example          Validation example          Test example

Figure 4: True test trajectories from solving the Burgers' equation following the setup of [77]. In this case, both training, validation, and test trajectories have the same length. We used $\Delta\tau = 0.01s$, so the trajectory contains states from time $\tau = 0s$ to $\tau = 1s$. The spatial dimension is discretized in 128 evenly distributed states

# D Data generation and implementation details

## D.1 Burgers' equation

Ground truth trajectories for the Burgers' equation [8]

$$\frac{\partial u}{\partial \tau} + u\frac{\partial u}{\partial z} = \nu\frac{\partial^2 u}{\partial z^2},$$

are obtained from the open-source dataset made available by Li et al. [40], where they consider $(x, \tau) \in (0, 1) \times (0, 1]$. The dataset consists of 1200 trajectories of 101 timesteps, i.e. $\Delta\tau = 0.01$, and where the spatial discretisation used is 128. 800 trajectories were used as training set, 200 as validation set and the remaining 200 as test set. These trajectories were generated following the setup described in [77] using the code available in their public repository. Initial conditions are sampled from a Gaussian random field defined as $\mathcal{N}(0, 625^2(-\Delta + 5^2 I)^{-4})$ (although be aware that in the paper they state to be using $\mathcal{N}(0, 25^2(-\Delta + 5^2 I)^{-4})$, which is different from what the code suggests) and then they integrate the Burgers' equation using spectral methods up to $\tau = 1$ using a viscosity of 0.01. Specifically, they solve the equations using a spectral Fourier discretisation and a fourth-order stiff time-stepping scheme (ETDRK4) [14] with step-size $10^{-4}$ using the MATLAB Chebfun package [16]. Examples of training, validation, and test trajectories are shown in Fig. 4.

## D.2 1D Kuramoto-Sivashinsky

We generate the ground truth trajectories for the 1D Kuramoto-Sivashinsky equation

$$\frac{\partial u}{\partial \tau} + u\frac{\partial u}{\partial z} + \frac{\partial^2 u}{\partial z^2} + \nu\frac{\partial^4 u}{\partial z^4} = 0,$$

following the setup of Lippe et al. [42], which is based on the setup defined by Brandstetter et al. [7]. In contrast to Lippe et al. [42], we generate the data by keeping $\Delta x$ and $\Delta\tau$ fixed, i.e. using a constant time and spatial discretisation[2]. The public repository of Brandstetter et al. [7] can be used to generate our dataset by using the same arguments as the one defined in Lippe et al. [42, App D1.].

Brandstetter et al. [7] solves the KS-equation with periodic boundaries using the method of lines, with spatial derivatives computed using the pseudo-spectral method. Initial conditions are sampled from a distribution defined over truncated Fourier series, i.e. $u_0(x) = \sum_{k=1}^{K} A_k \sin(2\pi l_k x/L + \phi_k)$, where $A_k, l_k, \phi_{kk}$ are random coefficients representing the amplitude of the different sin waves, the phase shift, and the space-dependent frequency. The viscosity parameter is set to $\nu = 1$. Data is initially generated with `float64` precision and then converted to `float32` for training the different models. We generated 1024 trajectories of $140\Delta\tau$, where $\Delta\tau = 0.2s$, as training set, while the validation and test set both contain 128 trajectories of length $640\Delta\tau$. The spatial domain is discretised into 256 evenly spaced points. Trajectories used to train and evaluate the models can be seen in Fig. 5.

---

[2]This just requires changing lines 166-171 of the `generate_data.py` file in the public repository of Brandstetter et al. [7].

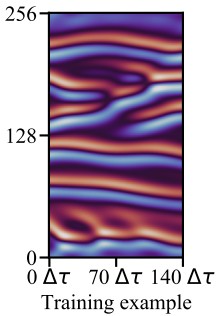
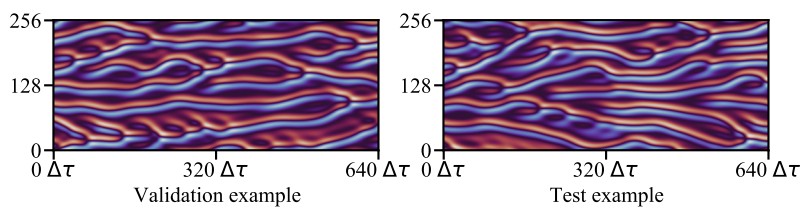

Training example    Validation example    Test example

Figure 5: Examples from the Kuramoto-Sivashinsky dataset. Training trajectories are shorter than those used to evaluate the models. Training trajectories have $140$ time steps generated every $\Delta\tau = 0.2s$, i.e. training trajectories contain examples from $0s$ to $28s$. Validation and test trajectories, instead, show the evolution of the equation for $640$ steps, i.e. from $0s$ to $128s$. The spatial dimension is discretized in 256 evenly distributed states.

### D.3  2D Kolmogorov-Flow equation

We use the setup described by [60] and their public repository to generate ground truth trajectories for the Kolmogorov-flow equation. Their implementation relies on the `jax-cfd` library [34] for solving the Navier-Stokes equation. As we have seen in Section 5, the Kolmogorov equation is defined as follows:

$$\frac{\partial \mathbf{u}}{\partial t} + \mathbf{u} \cdot \nabla \mathbf{u} - \nu \Delta^2 \mathbf{u} + \frac{1}{\rho}\nabla p - f = 0 \tag{17}$$

$$\nabla \cdot \mathbf{u} = 0$$

where $\nu > 0$ is the viscosity also defined as $\frac{1}{Re}$ with $Re$ being the Reynolds number, $\rho$ is the fluid density, $p$ the pressure field, and $f$ is an external forcing term. For the data generation they follow Kochkov et al. [34] setup, which used $Re = 1000$, or $\nu = 0.001$ respectively, a constant density $\rho = 1$ and $f = \sin(4y)\hat{\mathbf{x}} - 0.1\mathbf{u}$ where the first term can be seen as a forcing term and the second one as a dragging term. They solve the equation on a $256 \times 256$ domain grid defined on $[0, 2\pi]^2$ with periodic boundaries conditions but then they coarsen the generated states into a $64 \times 64$ resolution. While other works in the literature directly model the vorticity, i.e. the curl of the velocity field, they directly model the velocity field $\mathbf{u}$. They simulate Eq. (17) starting from different initial states for $128$ steps and they keep only the last 64 states. The training set we used contains 819 trajectories, the validation set contains 102 trajectories, and we test all the different models on 50 test trajectories. We show in Fig. 6 some example states from the training, validation, and test sets, corresponding to a time step $\in \{\Delta\tau, 8\Delta\tau, 16\Delta\tau, 24\Delta\tau, 32\Delta\tau, 40\Delta\tau, 48\Delta\tau, 56\Delta\tau, 64\Delta\tau\}$.

### D.4  Implementation details

Our implementation is built on PyTorch [50] and the codebase will soon be made publicly available. For generating samples we simulate the probabilistic ordinary differential equation (ODE) associated with the backward or denoising process which admits the same marginals as $\overleftarrow{x}(t)$, referred to as the *probability flow*

$$\mathrm{d}\bar{x}(t) = \left\{ -\tfrac{1}{2}\bar{x}(t) - \tfrac{1}{2}\nabla \log p_t(\bar{x}(t)) \right\} \mathrm{d}t.$$

using the DPM solver [44] in 128 evenly spaced discretisation steps. As a final step, we return the posterior mean over the noise-free data via Tweedie's formula. Unless specified otherwise, we set the guidance schedule to $r_t^2 = \gamma\sigma_t^2/\mu_t^2$ [19, 60], tune $\gamma$ via grid-search and set $\sigma_y = 0.01$.

In the following, we present all the hyperparameters used for the U-Net that parameterises the score function. For the forecasting task, we studied two different architectures for KS and Kolmogorov—one considered in [60] (SDA) and another one in [42] (PDE-Refiner). Tab. 3 contains the architecture and training parameters for the SDA architecture, while Tab. 4 contains the parameters used for the PDE-Refiner architecture. For KS, the SDA and PDE-Refiner architectures have very similar capacity in terms of the number of trainable parameters. The only difference is in the way they encode

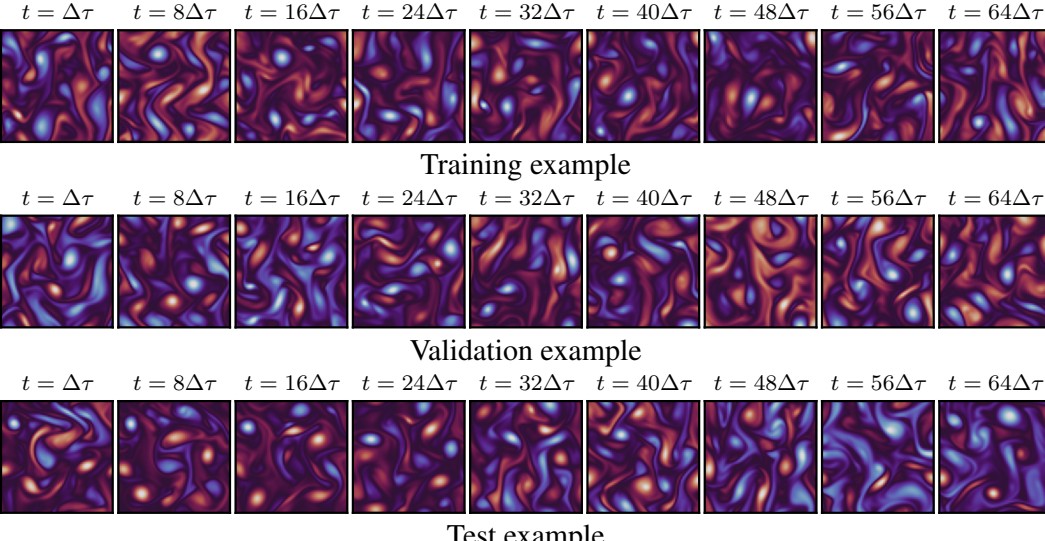

Figure 6: Examples from the Kolmogorov dataset. Training, validation, and test trajectories all have 64 time steps. For each example, we only show subset of the states, corresponding to $t \in \{\Delta\tau, 8\Delta\tau, 16\Delta\tau, 24\Delta\tau, 32\Delta\tau, 40\Delta\tau, 48\Delta\tau, 56\Delta\tau, 64\Delta\tau\}$

the diffusion time $t$: SDA encodes it as a separate channel that is concatenated to the input, while PDE-Refiner encodes $t$ into the parameters of the LayerNorm. For Kolmogorov, the architectures considered were different in terms of the number of residual blocks per level and channels per level, resulting in the PDE-Refiner architecture having more capacity than the SDA architecture.

The results in Figs. 1 and 2 correspond to the best-performing architecture for each model, but we show the full set of results in Tab. 5. The results for the data assimilation experiments are all obtained using the SDA architecture.

**Baselines implementation details.** In the forecasting experiment of Sec. 5, we tested the different score-based models against other ML-based models, including the state-of-the-art PDE-Refiner [42] and an MSE-trained U-Net and Fourier Neural Operator (FNO) [39]. For the implementation of the baselines, we used the following open-source code, all using the PDE-Refiner architecture from Tab. 4. In addition, for the MSE-trained U-Net, we also implemented a model using the SDA architecture, with results shown in Tab. 5.

**Hardware.** All models and the experiments we present in this paper were run on a single GPU. We used a GPU cluster with a mix of RTX 2080 Ti, RTX A5000, and Titan X GPUs. The majority of the GPUs we used have 11GB of memory. A direct map between experiments and a specific GPU was not tracked, therefore we cannot report these details. The results in Fig. 3 are computed on a RTX 2080 Ti GPU with 11GB of memory.

## E  All-at-once (AAO) joint sampling and study of solvers in forecasting

One approach for generating long rollouts conditioned on some observations using score-based generative modelling is to directly train a joint posterior score $s_\theta(t, x_{1:L}(t)|y)$ to approximate $\nabla_{x_{1:L}(t)} \log p(x_{1:L}(t)|y)$. The posterior can be composed from a prior score and a likelihood term, just as in the main paper. However, this approach cannot be used for generating long rollouts (large $L$), as the score network becomes impractically large. Moreover, depending on the dimensionality of each state, with longer rollouts the probability of running into memory issues increases significantly. To deal with this issue, in this section we investigate the approach proposed by Rozet and Louppe [60], whereby the prior score $(s_\theta(t, x_{1:L}(t)))$ is approximated with a series of local scores $(s_\theta(t, x_{i-k:i+k}(t)))$, which are easier to learn. The size of these local scores is in general significantly

Table 3: Hyperparameters used for the SDA U-Net architecture inspired by [60] in the three datasets considered in the paper. Blue refers to specific information about the joint model and red refers to the universal amortised model.

|  | BURGERS | KS | KOLMOGOROV |
|---|---|---|---|
| Residual blocks per level | (3,3,3) | (3,3,3,3) | (3,3,3) |
| Channels per level | (64,128,256) | (64,128,256,1024) | (96, 192, 384) |
| Kernel size | 3 | 3 | 3 |
| Padding | circular | circular | circular |
| Activation | SiLU | SiLU | SiLU |
| Normalization | LayerNorm | LayerNorm | LayerNorm |
| Optimizer | AdamW | AdamW | AdamW |
| Weight decay | $1e\text{-}3$ | $1e\text{-}3$ | $1e\text{-}3$ |
| Learning rate | $2e\text{-}4$ | $2e\text{-}4$ | $2e\text{-}4$ |
| Scheduler | linear | linear | linear |
| Epochs | 1024\|1024 | 4k\|15k | 10k\|10k |
| Batch size | 32 | 32 | 32 |

Table 4: Hyperparameters used for the PDE-Refiner U-Net architecture inspired by [42] in the forecasting experiment for the KS and Kolmogorov datasets. Blue refers to specific information about the joint model and red refers to the universal amortised model.

|  | KS | KOLMOGOROV |
|---|---|---|
| Residual blocks per level | (3,3,3,3) | (3,3,3,3) |
| Channels per level | (64,128,256,1024) | (128, 128, 256, 1024) |
| Kernel size | 3 | 3 |
| Padding | circular | circular |
| Activation | GELU | GELU |
| Normalization | GroupNorm | GroupNorm |
| Optimizer | AdamW | AdamW |
| Weight decay | $1e\text{-}3$ | $1e\text{-}3$ |
| Learning rate | $2e\text{-}4$ | $2e\text{-}4$ |
| Scheduler | linear | linear |
| Epochs | 2k\|15k | 700\|10k |
| Batch size | 128 | 128 |

smaller than the length of the generated trajectory and will be denoted as the window of the model $W$.

At sampling time, the entire trajectory is generated all-at-once, with the ability to condition on initial and/or intermediary states. We study two sampling strategies. The first one is based on the exponential integrator (EI) discretization scheme [82], while the second one uses the DPM++ solver [44]. For each strategy, the predictor steps can be followed by a few corrector steps (Langevin Monte Carlo) [68, 47]. For the results below, we consider the forecasting scenario for the Burgers' dataset. We forecast trajectories of 101 states, conditioned on a noisy version of the initial 6 states (with standard deviation $\sigma_y = 0.01$). The results are computed based on 30 test trajectories, each with different initial conditions. The sampling procedure uses 128 diffusion steps and a varying number of corrector steps.

**Influence of solver and time scheduling.** We show the results using two solvers: EI [82] and DPM++ [44]. For each, we study two time scheduling settings, which determine the spacing between the time steps during the sampling process

- Linear/uniform spacing: $t_i = t_N - (t_N - t_0)\frac{i}{N}$

- Quadratic: $t_i = \left(\frac{N-i}{N}t_0^{1/\kappa} + \frac{i}{N}t_N^{1/\kappa}\right)^{\kappa}$

where $N$ indicates the number of diffusion steps, $t_N$ and $t_0$ are chosen to be 1 and $10^{-3}$ and $\kappa$ is chosen to be 2.

Fig. 7 shows the RMSD between the generated samples and the true trajectories, averaged across the trajectory length. It indicates that for AAO sampling with 0 corrections, there is little difference between the two time schedules, and that DPM++ slightly outperforms EI in terms of RMSD. However, the differences are not significant when taking into account the error bars. As the number of corrections is increased, the difference in performance between the two solvers gets negligible. However, it does seem that with a large number of corrections (25), the quadratic time schedule leads to lower RMSD. Nevertheless, given that, in general, only $3-5$ corrector steps are used, we argue that the choice of solver and time schedule has a small impact on overall performance.

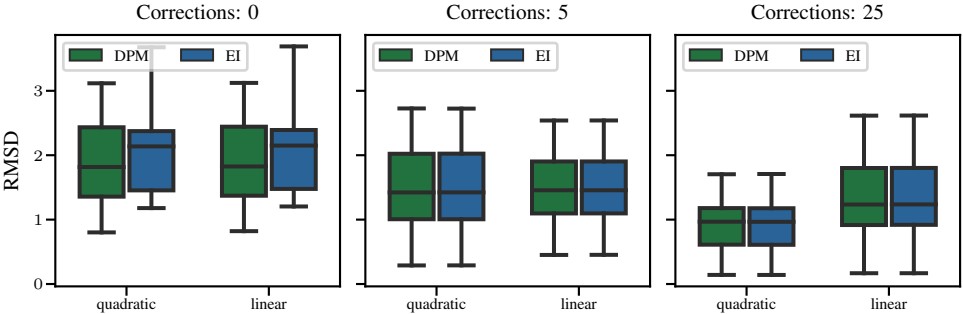

Figure 7: RMSD for Burgers' for the two solvers (DPM++ and EI), and the two time schedules (quadratic and linear) used. The results are shown for $0$ (left), $5$ (middle), and $25$ (right) corrector steps. When using no/only a few corrector steps, the RMSD is not highly impacted by the choice of solver and time schedule. However, at $25$ corrector steps, the quadratic time schedule seems to lead to lower RMSD.

**Influence of corrector steps and window size.**   Figs. 8 and 9 illustrate the effect of the window size and number of corrector steps on the mean squared error (MSE) and the correlation coefficient (as compared to the ground truth samples). Based on the findings from above, here we used the DPM++ solver with quadratic time discretisation.

The corrector steps seem to be crucial to prevent the error accumulation, with performance generally improving with increasing corrections for both models (window 5 and 9). However, this comes at a significantly higher computational cost, as each correction requires one extra function evaluation. Unsurprisingly, the bigger model (window 9) is better at capturing time dependencies between states, but it comes with increased memory requirements.

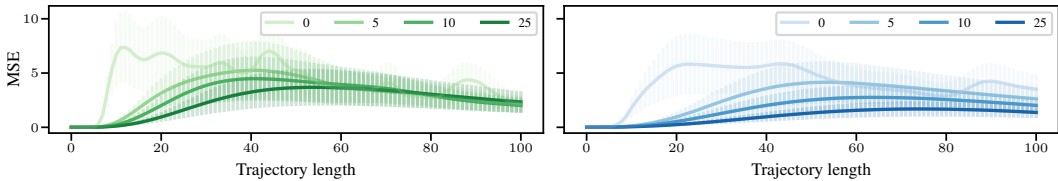

Figure 8: Evolution of the MSE for Burgers' along the trajectory length for a window of 5 (left) and 9 (right). The error bars indicate $\pm 3$ standard errors. Each line corresponds to a different number of corrector steps. Unsurprisingly, the window 9 model performs better than the window 5 model (when using the same number of corrector steps). Performing corrector steps, alongside predictor steps, is crucial for preventing errors from accumulating.

Finally, the findings from above are also confirmed in Fig. 10, which illustrates that the predictive performance increases with the window size and the number of corrector steps. Both of these come at an increased computational cost; increasing the window size too much might lead to memory issues, while increasing the number of corrections has a negative impact on the sampling time.

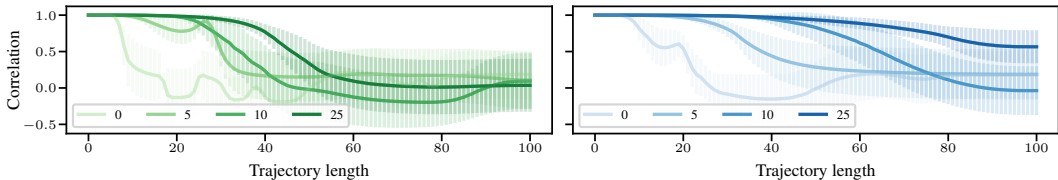

Figure 9: Evolution of the Pearson correlation coefficient for Burgers' along the trajectory length for a window of 5 (left) and 9 (right). The error bars indicate $\pm\, 3$ standard errors. The findings are entirely consistent with the ones suggested by the MSE results.

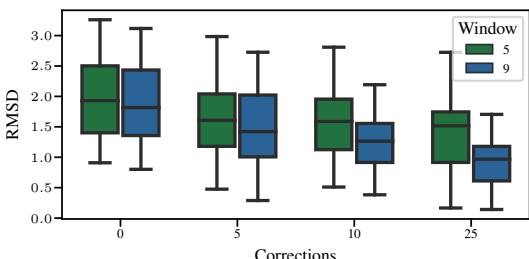

Figure 10: RMSD for Burgers' as a function of corrector steps for the two studied models (window 5 and window 9). Increasing both the window size and the number of corrections positively impacts the predictive performance.

## F    Additional experimental results on forecasting

This section shows further experimental results in the context of PDE forecasting. We show results on the Burgers', KS, and Kolmogorov datasets. We study the difference between all-at-once (AAO) and autoregressive (AR) sampling in forecasting, the influence of the guidance strength, and of the conditioning scenario in AR sampling for the joint and amortised models. Moreover, we also provide a comparison with some widely used benchmarks in forecasting: climatology and persistence. Finally, we provide results for two U-Net architecture choices for the diffusion-based models and the MSE-trained U-Net baseline.

### F.1    AAO vs AR

Once the joint model $s_\theta(t, x_{1:L}(t)) \approx \nabla_{x_{1:L}(t)} \log p(x_{1:L}(t))$ is trained, there are multiple ways in which it can be queried

- All-at-once (AAO): sampling the entire trajectory in one go (potentially conditioned on some initial and/or intermediary states);
- Autoregressively (AR): sampling a few states at a time, conditioned on a few previously generated states (as well as potentially on some initial and/or intermediary states).

Let us denote the length of the generated trajectory with $L$, the number of predictor steps used during the sampling process with $p$, and the number of corrector steps with $c$. Moreover, in the case of the AR procedure, we have two extra hyperparameters: the predictive horizon $P$ (the number of states generated at each iteration), and the number of conditioned states $C$. Finally, let the window of the model be $W$. Each sampling procedure has its advantages and disadvantages.

1. Sampling time: the AAO procedure (with no corrections) is faster than the AR one due to its non-recursive nature. We express the computational cost associated with each procedure in terms of the number of function evaluations (NFEs) (i.e. number of calls of the neural network) needed to generate the samples.
    - **AAO**: NFE = $(1 + c) \times p$
    - **AR**: NFE = $(1 + c) \times p \times (1 + \frac{L-W}{P})$
2. Memory cost: the AAO procedure is more memory-intensive than the AR one, as the entire generated trajectory length needs to fit in the memory when batching is not performed. If

batching is employed, then the NFEs (and hence, sampling time) increases. In the limit that only one local score evaluation fits in memory, the AAO and AR procedures require the same NFEs.

3. Redundancy of generated states: in the AR procedure, at each iteration the model re-generates the states it conditioned upon, leading to computational redundancy.

We show that the AAO procedure is not well-suited for forecasting tasks, by performing experiments on the Burgers' and KS datasets.

**Burgers'.** As mentioned above, the AR method comes at an increased computational cost at sampling time, but generates samples of higher quality and manages to maintain temporal coherency even during long rollouts. However, as illustrated in App. E, the quality of the AAO procedure can be improved by using multiple corrector steps. In particular, to obtain the same NFE for AAO and AR sampling (assuming the same number of predictor steps $p$ and no batching for AAO), we can use $c = \frac{L-W}{P}$ corrector steps.

Fig. 11 shows the results for a trajectory with $L = 101$, using a model with $W = 5$. The results are expressed in terms of the RMSD, calculated based on 30 test trajectories. For AAO sampling we set $c = 48$, while for AR sampling we set $c = 0$. In the AR sampling, we generated 2 states at a time conditioned on the 3 previously generated states. In both cases we conditioned on a noisy version of the initial 3 true states (with standard deviation $\sigma_y = 0.01$). For all experiments, we used the EI discretization scheme, with a quadratic time schedule. The AR procedure is clearly superior to the

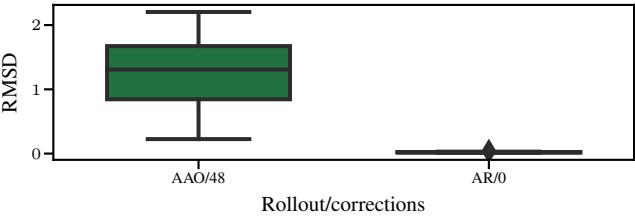

Figure 11: RMSD for Burgers' for AAO and AR generation with a window of 5. The number of corrections for AAO generation was chosen to match the computational budget of the AR sampling technique, yet the AR technique is clearly superior.

AAO, even with 48 corrector steps. This can be easily observed by comparing the generated samples to the ground truth as shown in Fig. 12.

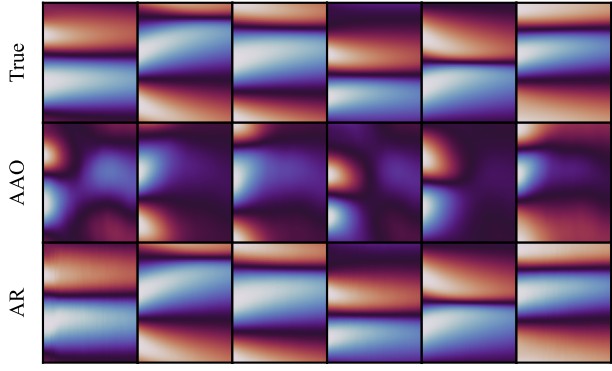

Figure 12: Comparison between the true samples (top), AAO samples (middle), and AR samples (bottom) for Burgers'. The AAO samples are only able to capture the initial states correctly, while the AR-generated samples maintain very good correlation with the ground truth even at long rollouts (101 time steps).

For a window of 9, we used the following number of corrector steps for each conditioning scenario:

- $P = 2, C = 7 : c = \frac{101-9}{2} = 46$
- $P = 3, C = 6 : c = \frac{101-9}{3} = 31$
- $P = 4, C = 5 : c = \frac{101-9}{4} = 23$
- $P = 5, C = 4 : c = \frac{101-9}{5} = 19$

Moreover, the number of states we initially condition upon is equal to $C$ (i.e., when comparing AAO with 46 corrections to AR with $P = 2, C = 7$ (2 | 7), we condition on the first 7 initial noised up states).

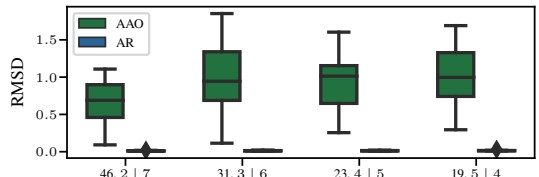

Figure 13: RMSD for Burgers' for AAO and AR generation with a window of 9 and with a varying computational budget. In the AAO case the computational budget is dictated by the number of corrector steps (indicated on the x-axis through the number before the comma). In the AR case, the budget is determined by the number of states generated at each iteration ($P$). This is also indicated on the x-axis in the format $P \mid C$ (e.g., 2 | 7 implies we generate two states at a time and condition on 7). AR is clearly superior to AAO for forecasting.

Once again, the AR-generated trajectories achieve significantly better metrics, which also reflects in the quality of the generated samples. Fig. 14 shows examples for AAO: 46/AR: 2 | 7.

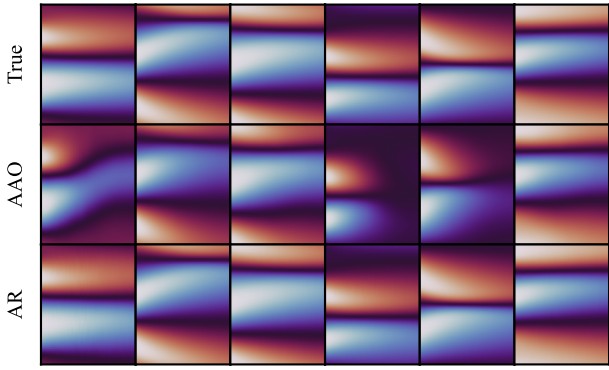

Figure 14: Comparison for Burgers' between the true samples (top), AAO samples with 46 corrections (middle), and AR 2 | 7 samples (bottom). With 46 corrector steps, some AAO trajectories manage to maintain good correlation with some of the easier test trajectories. However, the quality of the AR samples is clearly superior to the AAO samples, managing to model all test trajectories well.

**KS.** We show similar findings on the KS dataset, where we present the performance of AAO and AR sampling on 128 test trajectories of length $100\Delta t$ (corresponding to 20 seconds) in terms of MSE along the trajectory length. For AAO we employ a varying number of corrections (0, 1, 3, and 5) and for each setting we show the results for the best $\gamma \in \{0.03, 0.05, 0.1, 0.5\}$. Setting $\gamma$ to lower values led to unstable results. For AR we set $\gamma = 0.1$ and show results for 8 | 1. We condition on the first 8 true states and show the results for the remaining 92 states.

Fig. 15a clearly shows that the AAO sampling has poor forecasting performance as compared to AR sampling. Employing corrections has a positive impact close to the initial states, but they no longer seem to help for longer rollouts. Even if we employ 5 Langevin corrector steps, the AAO metrics are significantly worse than the AR metrics.

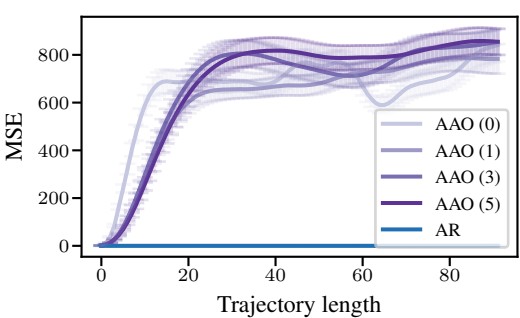

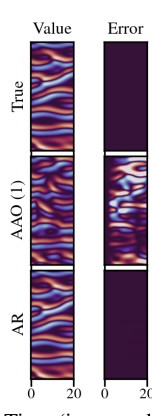

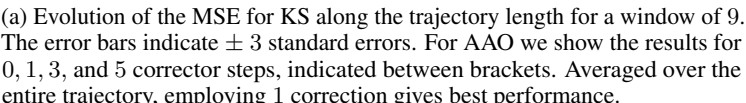

(a) Evolution of the MSE for KS along the trajectory length for a window of 9. The error bars indicate $\pm 3$ standard errors. For AAO we show the results for $0, 1, 3$, and $5$ corrector steps, indicated between brackets. Averaged over the entire trajectory, employing 1 correction gives best performance.

(b) Example trajectories from the window 9 model.

Figure 15: AAO vs AR comparison for the KS dataset.

Fig. 15b shows an example trajectory of length 100, alongside the predictions from AAO with 1 correction and AR sampling (left column). The right column shows the difference between the predictions and the ground truth. The AR procedure leads to results that are very similar to the ground truth for the first 100 states, whereas the predictions from AAO sampling quickly diverge from the ground truth.

**Kolmogorov.** We also confirm the previous findings in the Kolmogorov dataset, with experimental evidence shown in Fig. 16. We compare the performance between AAO and AR sampling for a model with a window size of 5 in terms of MSE along the trajectory length. We use 50 test trajectories of length 64. For AAO we show the results for 0 and 1 corrections, and show the results for the best setting for the guidance strength between $\gamma \in \{0.03, 0.05, 0.1, 0.3\}$, corresponding to $\gamma = 0.05$ in both cases. For AR we show results for $1 \mid 4$ and $\gamma = 0.1$. We condition on the first 4 states and show the results for the remaining 60 states.

We plot in Fig. 16b the vorticity associated with the true, AAO-predicted, and AR-predicted states at eight different time points. The AAO-predicted states diverge from the ground truth quickly, while the AR-predicted states stay close to the ground truth for more than half of the trajectory length.

### F.2 Influence of guidance strength in forecasting

**KS.** We studied three different settings for the parameter that controls the guidance strength $\gamma \in \{0.01, 0.03, 0.1\}$ for the KS dataset for joint AR sampling. We found $\gamma = 0.01$ to be unstable in the case of forecasting, so in Fig. 17 we show the results for $\gamma \in \{0.03, 0.1\}$. We consider a variety of window sizes $W$ and conditioning scenarios $P \mid C$, with $P + C = W$, where $P$ represents the number of states generated at each rollout iteration, and $C$ the number of states we condition upon. As shown in Fig. 17, $\gamma = 0.1$ is the better choice for low / middle-range values of $P$, and decreasing it to $\gamma = 0.03$ can lead to unstable results. In contrast, $\gamma = 0.03$ gives enhanced performance for longer predictive horizons $P$, as compared to $\gamma = 0.1$.

**Kolmogorov.** We studied five different settings for the guidance strength $\gamma$ in the Kolmogorov dataset $\gamma \in \{0.01, 0.03, 0.05, 0.1, 0.2, 0.3\}$. The lowest value 0.01 gave unstable results, so we are not showing it in Fig. 18. The findings are similar to those for the KS dataset, indicating that the guidance strength needs to be tuned depending on the conditioning scenario. For $P = 3$ and $P = 4$, the best setting of $\gamma$ is 0.03. However, for smaller $P$, $\gamma = 0.03$ gives unstable results, and the best settings are $\gamma = 0.05$ for $P = 2$, and $\gamma = 0.1$ for $P = 1$.

### F.3 Forecasting performance of different conditioning scenarios $P \mid C$

The autoregressive sampling method offers some flexibility in terms of the number of states generated at each iteration ($P$). From a computational budget perspective, the higher $P$, the fewer iterations

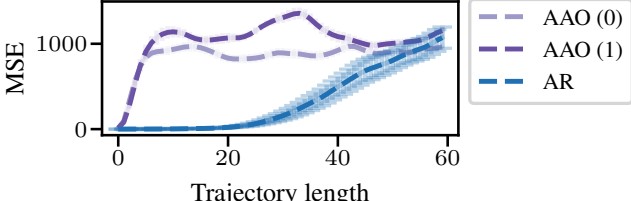

(a) Evolution of the MSE for Kolmogorov along the trajectory length for a model with window 5. The error bars indicate $\pm\,3$ standard errors. For AAO we show the results for 0, and 1 corrector steps, indicated between brackets. Averaged over the entire trajectory, employing corrections does not improve performance.

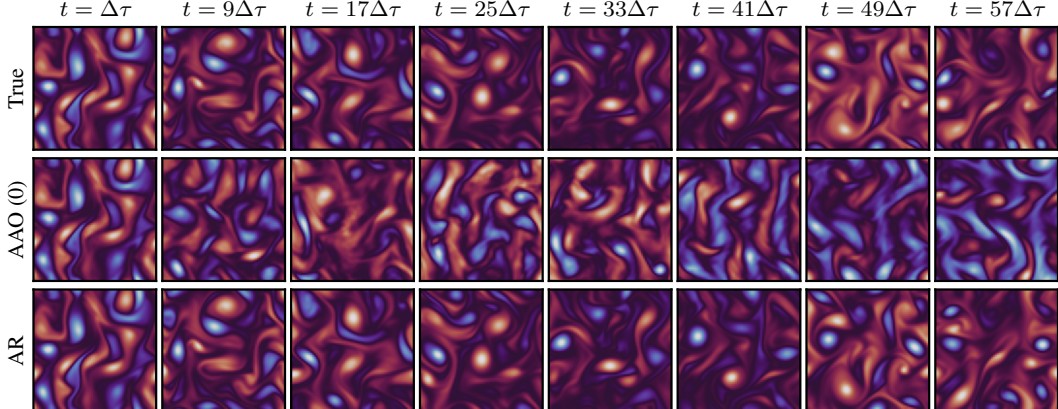

(b) The vorticity of the predicted states from the window 5 model. The top row shows the true vorticity, the middle one shows the vorticity of the AAO-predicted samples, and the bottom row, the vorticity of the AR-predicted states. We indicate by $t$ the time index of the state (i.e. $t = \Delta\tau$ means first forecast state).

Figure 16: AAO vs AR comparison for the Kolmogorov dataset.

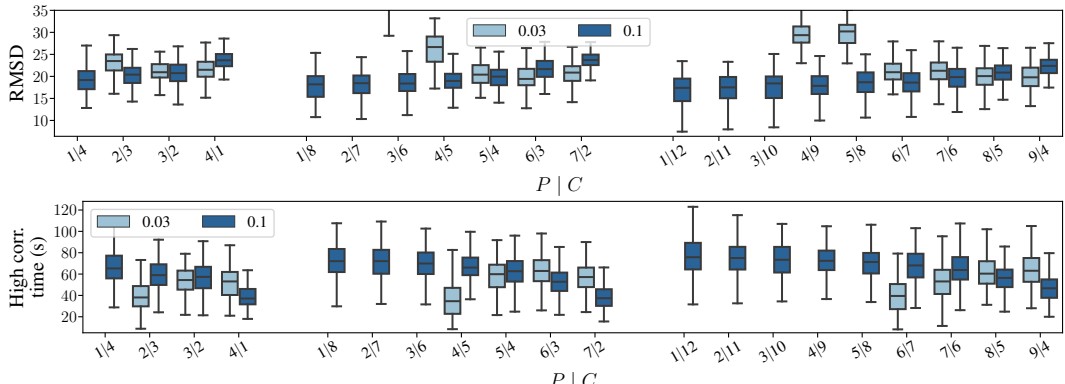

Figure 17: Forecasting performance of the joint model in the KS dataset for $\gamma \in \{0.03, 0.1\}$ for varying window sizes and $P \mid C$ conditioning scenarios. The top row shows the RMSD, while the bottom one shows the high correlation time. Where we do not show results for $\gamma = 0.03$ it is because they were unstable.

need to be performed. However, this might come at the cost of a decrease in the quality of the generated samples. In this section, we investigate this trade-off for the Burgers' and KS datasets. The results for the Kolmogorov dataset are shown in the main paper in Fig. 1.

We present results for a window of 9 in the Burgers' dataset and study several window sizes $W$ for the KS dataset. We also vary the predictive horizons $P$, and the number of conditioning frames $C = W - P$. The trajectories are conditioned upon the first $W - P$ true states.

The results are presented in terms of RMSD, which is computed based on 200 test trajectories for the Burgers dataset, and on 128 test trajectories for the KS dataset. The number of diffusion steps used is 128. For all experiments, we used the DPM++ solver, with a linear time discretisation scheme. At the

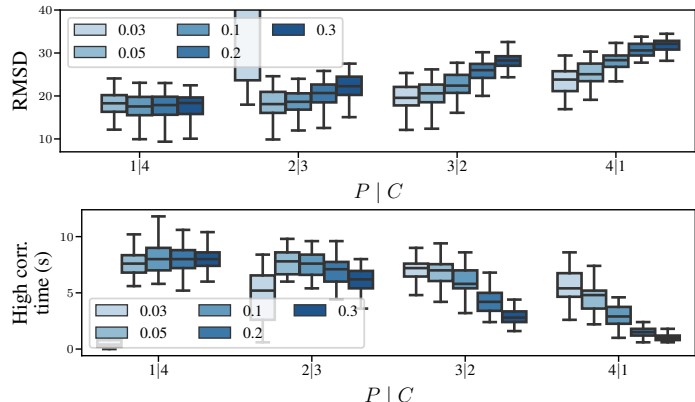

Figure 18: Forecasting performance of the joint model in the Kolmogorov dataset for $\gamma \in \{0.03, 0.05, 0.1, 0.2, 0.3\}$ for varying window sizes and $P \mid C$ conditioning scenarios. The top row shows the RMSD, while the bottom one shows the high correlation time.

end, we also performed an additional denoising step, and adopted the dynamic thresholding method for the Burgers' dataset. The guidance strength is fixed at $\gamma = 0.1$ for the Burgers' dataset, while for KS we show the best setting out of $\gamma \in \{0.03, 0.1\}$.

**Burgers'.** Fig. 19 shows that the performance is not very sensitive to $P$ for low to medium values of $P$, although the model seems to perform better with lower $P$ (and consequently, higher number of conditioning frames $C$). However, the performance quickly deteriorates for very high values of $P$ (i.e., $P = 8$).

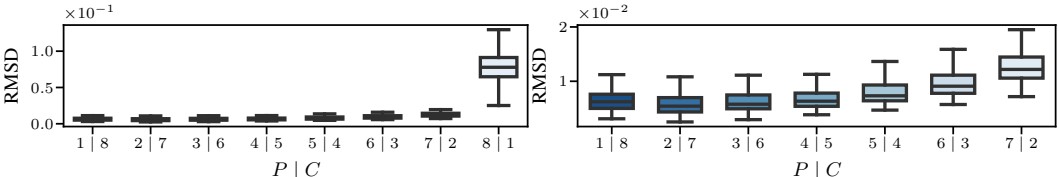

Figure 19: Burgers' dataset. RMSD for different values of $P$. The right plot is a zoomed in version of the left one, where we excluded the $8 \mid 1$ setting. Outliers have been removed.

**KS.** We repeat the same experiment for the KS dataset, but study a wider variety of window sizes $W$ and conditioning scenarios $P \mid C$, where $P + C = W$. Moreover, as mentioned in G.1, for each setting we test two values of the guidance parameter $\gamma$. In Fig. 20 we only show the results corresponding to the best guidance strength. We also show results for the amortised model introduced in Sec. 3.2 in Fig. 21, where a different model was trained for each $P \mid C$ scenario. Unlike the joint model, which was trained for 4000 epochs, the amortised model was only trained for 1024 epochs, since after that it started overfitting.

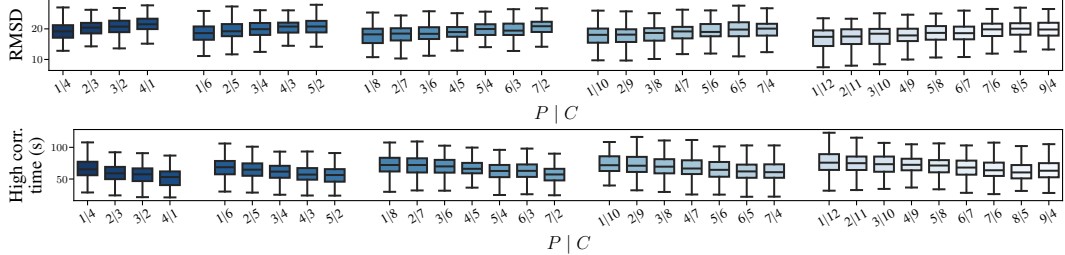

Figure 20: RMSD (top) and high correlation time (bottom) for the joint model on the KS dataset for varying window sizes $W$ and $P \mid C$ conditioning scenarios (where $P + C = W$). The model performs best for low values of $P$ ($P = 1$), and high values of the number of conditioning frames $C$.

**Joint model.** As observed in Fig. 20, the best conditioning scenario for the joint model for each window corresponds to $P = 1$ (i.e. a low number of predicted frames). We observe an increase in performance by increasing the window size, with the best performing setting corresponding to a model with window 13 (1 | 12). In this case the model achieves 77.3s high correlation time, as opposed to 73.2s for the window 9 model presented in the main paper. We believe further gains in performance could be achieved by increasing the window size further, but this would also come with higher memory requirements.

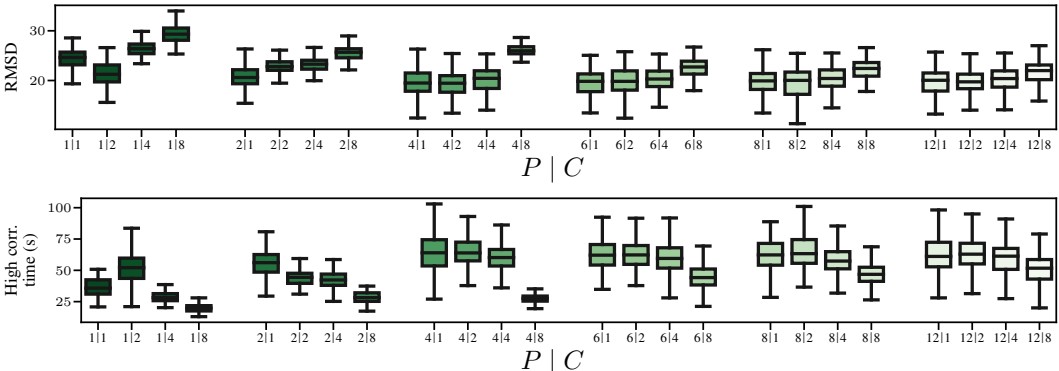

Figure 21: RMSD (top) and high correlation time (bottom) for the amortised model in the KS dataset with varying $P$ and $C$. The model performs best when generating relatively large blocks ($P \geq 4$) and conditioning on a lower number of frames ($C \leq 4$).

**Amortised model.** In contrast to the joint model, Fig. 21 shows that the amortised model underperforms for small $P = 1$, with high correlation time of less than 50s. We hypothesise that the reason for this might be the combination of the following: i) generating larger number of frames at the same time requires less steps for the same trajectory length, so there is less error accumulation; and ii) for larger $P$ the model is trained with a better learning task compared to $P = 1$, as larger $P$ provides more information to the model. For $P \geq 4$, the model performs similarly for any $P$, being slightly more sensitive to larger $C$ when $P$ is lower. Regardless of $P$, the performance of the model deteriorates significantly with larger $C = 8$, similar to the behaviour shown in Lippe et al. [42].

**Universal amortised model.** As indicated in Sec. 3.2, a more flexible alternative to the amortised model is the universal amortised model. Due to our novel training procedure, the latter does not need re-training for each $P \mid C$ conditioning scenario. Moreover, as shown in Fig. 1 and Fig. 22, the performance of the universal amortised model is stable for all $P \mid C$ scenarios, and, unlike other models from the literature [42], is not negatively affected by longer previous history.

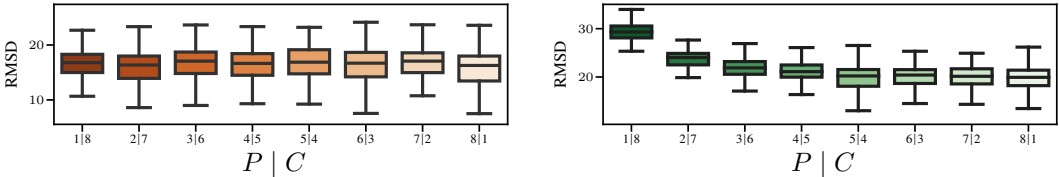

Figure 22: RMSD for universal amortised model (left) and amortised model (right) in the KS dataset with varying $P$ and $C$. Universal amortised model is more stable as well as outperforms the amortised model across all combinations of $P$ and $C$.

## F.4   Comparison with climatology and persistence

We also compare the performance of our models (joint AR and universal amortised) to two common forecasting benchmarks—*climatology*, determined by computing the average value over the entire training set, and *persistence*, obtained by repeating the last observed state for the entire trajectory. Rather than looking at RMSD or high correlation time, we look at a per time step criterion—MSE.

This allows us to find out at which point predicting just the mean value is preferred to the predictions of the models (i.e. when climatology gives lower MSE than the models).

Fig. 23 shows that the joint and universal amortised models outperform persistence throughout the trajectory length. For KS, they significantly outperform climatology until approximately 400 time steps, corresponding to about 80s. As observed in the main paper through the high correlation time, the amortised model seems to be better at long-term prediction than the joint model. For Kolmogorov, they outperform climatology for more than 40 time steps, corresponding to 8s.

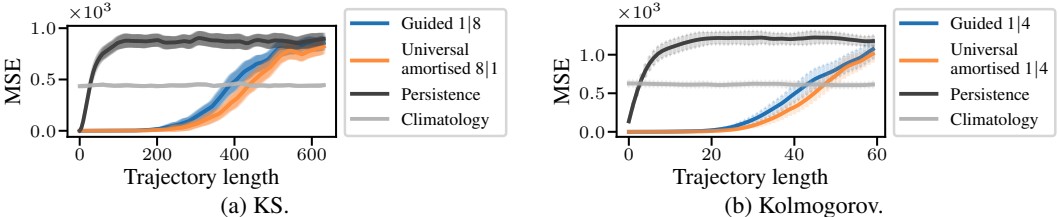

(a) KS.

(b) Kolmogorov.

Figure 23: MSE along the trajectory length for KS (left) and Kolmogorov (right) for 1) the joint AR model (blue), 2) the universal amortised model (orange), and the two forecasting benchmarks considered—3) persistence (black) and 4) climatology (grey). The error bars indicate $\pm 3$ standard errors. Our models outperform climatology for about two thirds of the trajectory length.

## F.5  Example samples

We show in Fig. 24 some example predicted trajectories using the joint $1 \mid 8$ model and the universal amortised $8 \mid 1$ model for the KS dataset, alongside the ground truth trajectories. We also show the error between the predictions and the true trajectories.

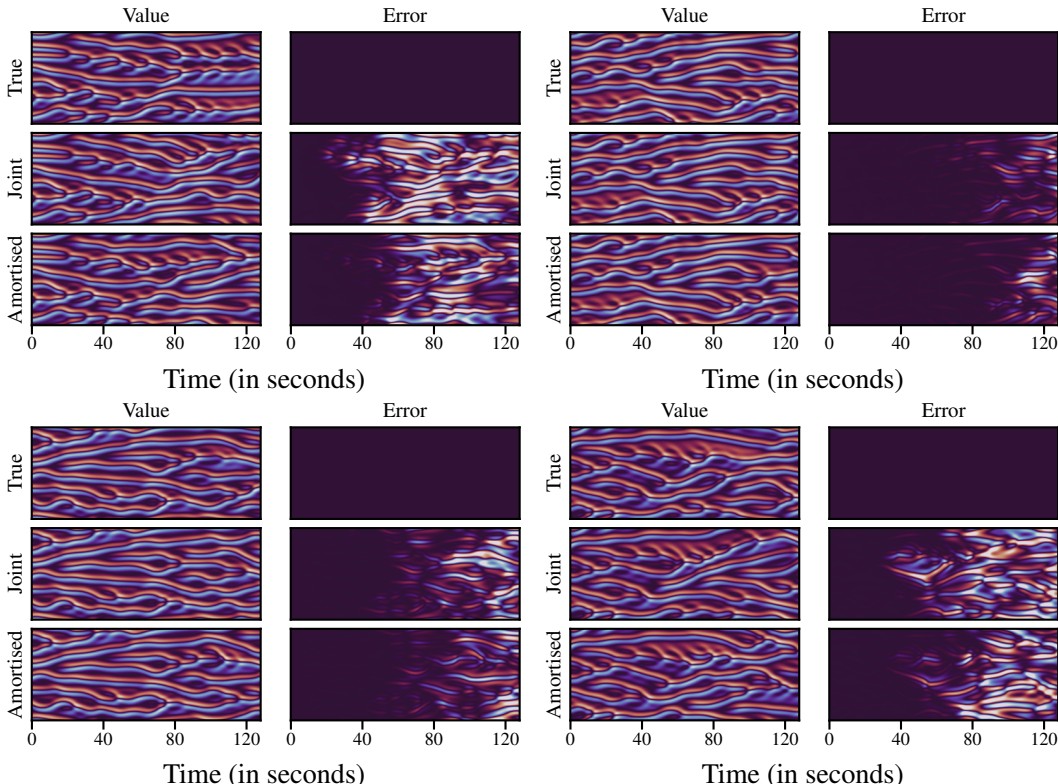

Figure 24: Example true trajectories, and model predictions for the joint $1 \mid 8$ (middle) and universal amortised $8 \mid 1$ (bottom) model on the KS dataset. The top left/right example corresponds to one of the trajectories with the highest/lowest RMSD.

For Kolmogorov, we show in Fig. 25 examples of the vorticity at different time steps $t$ corresponding to the true states (top), to the states predicted by the joint $1 \mid 4$ model (middle), and to the states predicted by the universal amortised $1 \mid 4$ model (bottom).

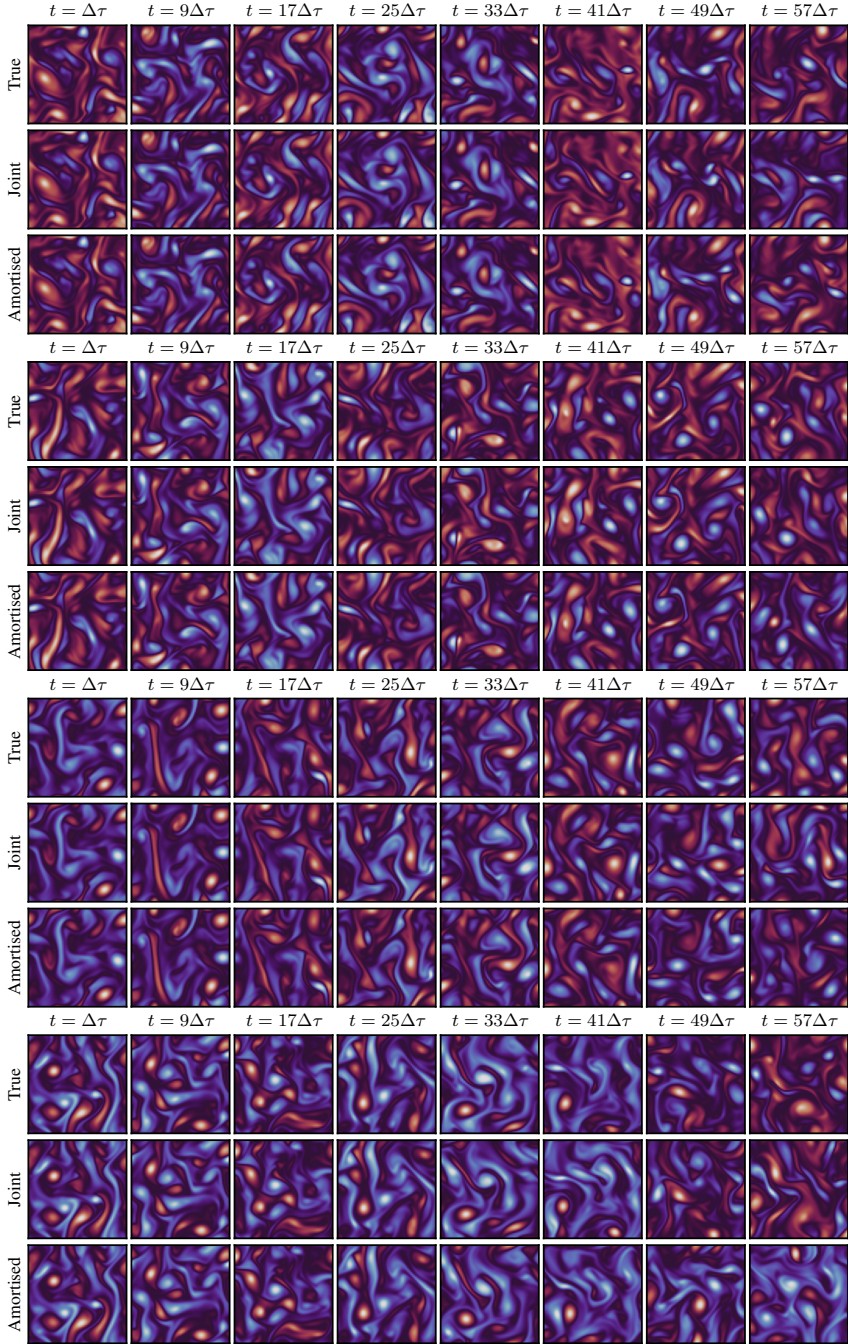

Figure 25: Example vorticity fields at different time steps and for four different initial states. Each plot consists of three different rows corresponding to the true trajectories (top), to the trajectories predicted by the joint $1 \mid 4$ model (middle), and to those predicted by the universal amortised $1 \mid 4$ model (bottom).

### F.6 Results for different architectures

For the diffusion models and the MSE-trained U-Net baseline outlined in Sec. 5, we study two different U-Net architectures and report the best results in Fig. 2. One follows [60] (SDA), while the other one follows [42] (PDE-Refiner). More details about the exact hyperparameters and training details can be found in App. D.4.

As illustrated in Tab. 5, we obtain better results with the SDA architecture for the joint AR model, while the PDE-Refiner architecture yields better performance for the amortised models. For the MSE-trained U-Net baseline, the SDA architecture gives better results for the KS dataset, but worse for Kolmogorov.

Table 5: The best high correlation time (mean $\pm$ 3 standard errors over test trajectories) achieved for two different architectures—SDA, inspired by [60], and PDE-Refiner, inspired by [42]. For the diffusion-based models, the conditioning scenario $P \mid C$ that corresponds to the best performance is indicated between brackets.

| | KS | | KOLMOGOROV | |
| --- | --- | --- | --- | --- |
| | SDA [60] | PDE-Refiner [42] | SDA [60] | PDE-Refiner [42] |
| JOINT AR | $73.2 \pm 4.3$s $(1 \mid 8)$ | $70.9 \pm 4.2$s $(1 \mid 8)$ | $8.2 \pm 0.6$s $(1 \mid 4)$ | $6.8 \pm 0.5$s $(1 \mid 4)$ |
| UNIV. AMORTISED | $82.0 \pm 4.4$s $(8 \mid 1)$ | $88.7 \pm 4.4$s $(7 \mid 2)$ | $8.9 \pm 0.6$s $(1 \mid 4)$ | $10.0 \pm 0.5$s $(1 \mid 4)$ |
| AMORTISED | $63.1 \pm 3.8$s $(7 \mid 2)$ | $78.6 \pm 4.4$s $(8 \mid 1)$ | $10.0 \pm 0.5$s $(1 \mid 4)$ | $10.6 \pm 0.5$s $(2 \mid 3)$ |
| MSE U-NET | $91.9 \pm 4.9$s | $82.5 \pm 4.8$s | $7.7 \pm 0.6$s | $10.0 \pm 0.3$s |

## G Additional experimental results on offline data assimilation

In this section, we provide more details about the offline DA setup. For KS, for each sparsity level we uniformly sample the indices associated with the observed variables in space-time. For Kolmogorov, we uniformly sample indices in space at each time step. For both datasets, we assume to always observe some full initial states. The number of such states for AR sampling depends on the chosen conditioning scenario and is equal to $C$. For AAO we condition on $W - 1$ initial states, where $W$ stands for the window size.

Furthermore, this section provides further experimental results for the offline data assimilation task for the KS and Kolmogorov datasets. We did not explore this for the Burgers' dataset since we decided to focus on PDEs with more challenging dynamics. We investigate the influence of the guidance strength, and the performance of the AR models with varying conditioning scenarios. We also perform a comparison between a joint model with Markov order of $k$ that is queried autoregressively and a $2k$-Markov order model queried all-at-once. This allows us to gain further insight into the differences between the two sampling schemes. Finally, we provide a quantitative and qualitative comparison to a simple interpolation baseline.

### G.1 Influence of the guidance strength for the joint model

We study several settings of the guidance parameter $\gamma$ for 1) the joint model sampled AAO, and 2) the joint model sampled autoregressively. We show the results in terms of RMSD evaluated on 50 test trajectories for varying levels of sparsity, as in the main paper.

**Joint AAO.** For the KS dataset, we consider a window 9 model, with $\gamma \in \{0.01, 0.03, 0.05, 0.1, 0.5\}$. For Kolmogorov, we show the results for a window 5 model and $\gamma \in \{0.03, 0.05, 0.1\}$. We query them AAO, and employ 0 and 1 corrections (AAO (0) and AAO (1)).

For KS, using $\gamma = 0.01$ results in unstable rollouts, therefore we only report the results for the other four values of $\gamma$. For the sparse observation regimes, we find that the best results are achieved by employing the lowest $\gamma$ value that still leads to stable results ($\gamma = 0.03$). For the denser regimes (i.e. proportion observed $\geq 10^{-1}$ for 0 corrections and $\geq 10^{-1.5}$ for 1 correction), the best setting corresponds to $\gamma = 0.05$.

Fig. 27 shows the results on the Kolmogorov dataset. In contrast to the KS dataset, we found that the best $\gamma$ value is consistent among all observation regimes, and corresponds to $\gamma = 0.05$ for both AAO

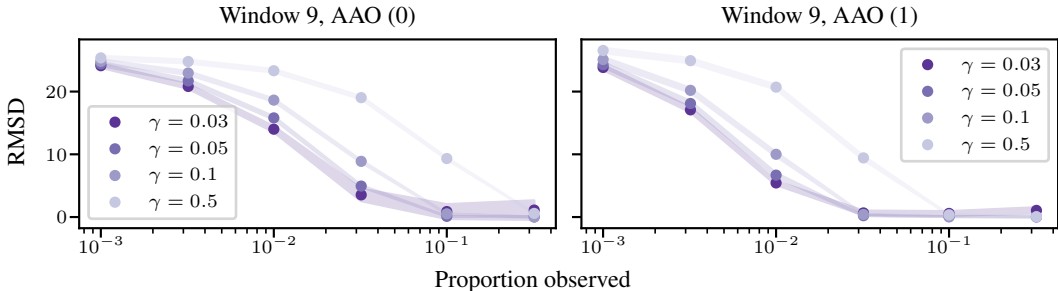

Figure 26: RMSD on the KS dataset for the joint model queried AAO with 0 (AAO (0) - left) and 1 (AAO (1) - right) corrections. We show results for a window of size 9. Confidence intervals indicate $\pm\,3$ standard errors.

(0) and AAO (1). In this case, decreasing the $\gamma$ value (i.e. increasing the guidance strength) too much does not lead to improved performance and leads to a much higher standard error, as compared to the other studied settings.

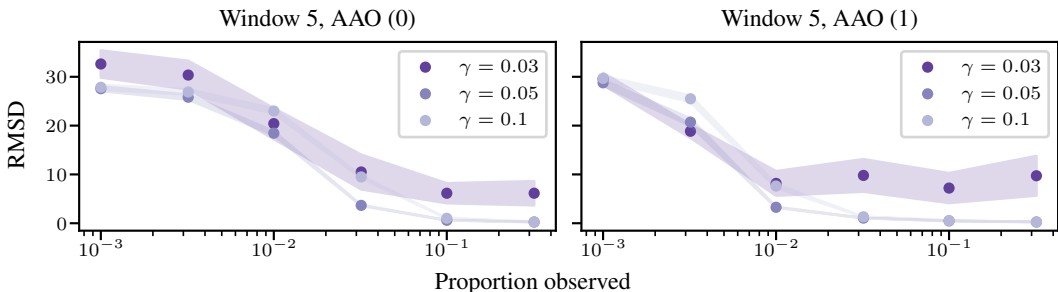

Figure 27: RMSD on the Kolmogorov dataset for the joint model queried AAO with 0 (AAO (0) - left) and 1 (AAO (1) - right) corrections. We show results for a window of size 5. Confidence intervals indicate $\pm\,3$ standard errors.

**Joint AR.** For the joint AR scheme not only can we vary the guidance strength, but we can also change the conditioning scenario $P \mid C$. In Fig. 28 we show the RMSD for the KS dataset for two values of $\gamma \in \{0.03, 0.1\}$ for a window 9 model. Each plot shows the results for a different sparsity level.

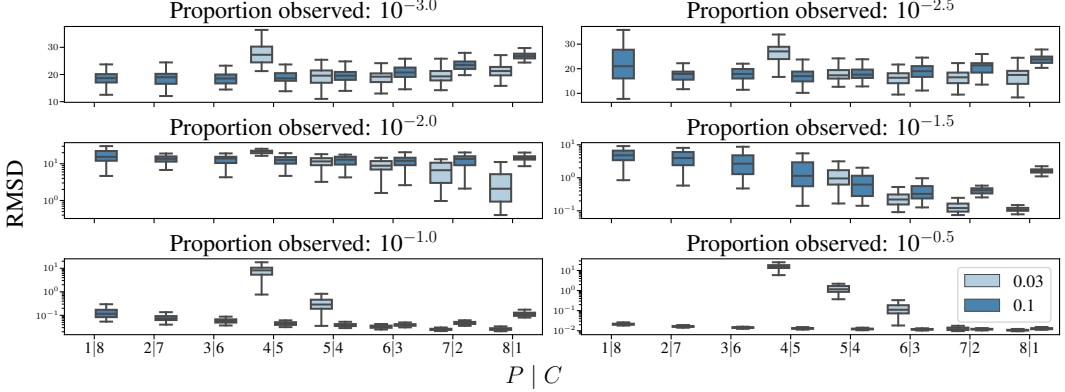

Figure 28: RMSD for the KS dataset on the offline DA task for varying conditioning scenarios and $\gamma = 0.03$ (light blue) and $\gamma = 0.1$ (dark blue). For each sparsity level (proportion observed), we consider all the $P \mid C$ scenarios for a model with window 9. When results are not presented for $\gamma = 0.03$, it is because they were unstable. Error bars indicate $\pm\,3$ standard errors. Note the logarithmic y-scale for the bottom two rows of plots.

Fig. 28 shows that for the lower values of $P$, i.e. settings where we predict only a few frames at a time, the results become unstable for low values of $\gamma$. However, especially in the sparser regimes, decreasing $\gamma$ to 0.03 can lead to a significant improvement in performance for the larger values of $P$, i.e. settings with a higher predictive horizon. We also tried decreasing $\gamma$ further to 0.01 but the results were unstable for the majority of $P \mid C$ settings. The only scenario where $\gamma = 0.01$ achieves a better performance than the settings presented in Fig. 28 is for $8 \mid 1$ and proportion observed of $10^{-1.5}$, and this is the scenario considered in Fig. 3 for the above-mentioned proportion of observed data.

For Kolmogorov, we studied $\gamma \in \{0.03, 0.05, 0.1, 0.2, 0.3\}$ for each conditioning scenario of a window 5 model ($P \in \{1, 2, 3, 4\}$). We show the best $\gamma$ setting for each conditioning setting in Tab. 6. Similarly to the KS dataset, we find that lower $\gamma$ values give better results for larger $P$, while they are unstable for lower $P$.

Table 6: Best $\gamma$ corresponding to each conditioning scenario $P$ for different proportion of data observed on the Kolmogorov dataset for the joint model with window size 5.

| Proportion observed | best $P$ | best $\gamma$ |
|---|---|---|
| 0.32 | 4 | 0.03 |
| 0.1 | 4 | 0.03 |
| 0.032 | 4 | 0.03 |
| 0.01 | 4 | 0.03 |
| 0.0032 | 1 | 0.1 |
| 0.001 | 1 | 0.2 |

### G.1.1 Conditioning for DA in the universal amortised model

In our experiments, we found that to get better performance for the universal amortised model with reconstruction guidance, both $\gamma$ and $\sigma_y$ need to be tuned given a particular proportion of data observed. What is more, the universal amortised model is also more sensitive to the choice of the reconstruction guidance parameters than the joint one. Due to computation constraints and sensitivity of the universal amortised model to the choice of hyperparameters in reconstruction guidance, we limit our experiments to a model with window 9 for the KS dataset.

For each proportion of observed data we perform a hyperparameter sweep varying $\gamma \in \{0.1, 0.05, 0.01\}$ and $\sigma_y \in \{0.01, 0.005, 0.001\}$. Tab. 7 shows the value for the best predictive horizon $P$, $\gamma$ and $\sigma_y$ for each proportion of data observed for a model with window size of 9 on the KS dataset, and Tab. 8 shows the results for the Kolmogorov dataset for a model with window size 5. Empirically, we found that the more data are observed, the lower values of $\gamma$ and $\sigma_y$ are required for optimal performance. Lower values of $\gamma$ and $\sigma_y$ increase the strength of the guidance term, which indeed is expected to guide the process better when more data are available.

As described in the Sec. 3.2, for the universal amortised model we propose to use conditioning via architecture for past states and reconstruction guidance for sparse observations DA. However, there are a few other options that could be used for incorporating sparse observations in amortised models: i) resampling [46], where at each reverse diffusion time step more sampling iterations are introduced, which mix observed and unobserved variables, and ii) conditioning via architecture, which requires a separate model, taking not only previous states, but also sparse observations as input, to be trained.

Table 7: Best $\gamma$, $\sigma_y$ and $P$ for different proportions of data observed on the KS dataset for universal amortised model with window size 9.

| Proportion observed | best $P$ | best $\gamma$ | best $\sigma_y$ |
|---|---|---|---|
| 0.32 | 6 | 0.05 | 0.001 |
| 0.1 | 6 | 0.05 | 0.001 |
| 0.032 | 6 | 0.1 | 0.001 |
| 0.01 | 6 | 0.1 | 0.005 |
| 0.0032 | 6 | 0.1 | 0.01 |
| 0.001 | 7 | 0.1 | 0.01 |

Table 8: Best $\gamma$, $\sigma_y$ and $P$ for different proportions of data observed on the Kolmogorov dataset for universal amortised model with window size 5.

| Proportion observed | best $P$ | best $\gamma$ | best $\sigma_y$ |
|---|---|---|---|
| 0.32 | 1 | 0.05 | 0.001 |
| 0.1 | 1 | 0.05 | 0.001 |
| 0.032 | 2 | 0.05 | 0.001 |
| 0.01 | 1 | 0.05 | 0.001 |
| 0.0032 | 1 | 0.1 | 0.01 |
| 0.001 | 1 | 0.1 | 0.01 |

We have not investigated the resampling strategy, as in their work [15] showed that it has worse performance compared to reconstruction guidance in image and protein design applications. Moreover, in practice resampling requires around 10 mixing sampling steps, with one step accounting for a single call to the score network, for each reverse diffusion iteration. Therefore, this makes the conditional sampling with resampling approach much more compute-intense and less effective in terms of sampling time.

We have studied the second approach for conditioning, where all the conditioning is happening via the architecture. For this, we trained a separate model, which takes in fully observed previous states and some random data observations with masks as an input to the score network. During training we sample the number of fully observed states $C$ as well as random masks for sparse observations within the rest of the $P$ frames for each batch. As a result, the model is trained to fit the following conditional score: $\mathbf{s}_\theta(t, x_{i-C:i+P}(t)|x_{i-C:i}, x_o)$. Fig. 29a shows the performance of this model (green) for different proportions of observed data. We considered a larger proportion of observed data, as for the lower ones considered in the main text of the paper, the model was significantly underperforming. From the Fig. 29a, it can be clearly seen that the model with all the conditioning done through the architecture performs significantly worse compared to our proposed version of the universal amortised model with reconstruction guidance. More specifically, the level of the universal amortised model's performance with $10\%$ of data observed is reached by the other model only when more than $70\%$ of data is observed. We hypothesise that such poor model performance can be explained by the fact that at some point during the AR rollout, the input condition (previously generated $C$ states and partially observed sparse data within window of size $P$) that is provided to the model becomes out-of-distribution due to the mismatch between the previously generated states and the partially observed variable from the true trajectory. Fig. 29b shows inputs to the score network depending on the AR step on KS data. The model was trained with window 9 and uses 3 previously generated states for conditioning. The mismatch between previously generated states and observed trajectory data is visually noticeable at AR step 60, while becoming even more drastic with more AR steps. Therefore, there is a mismatch between input distribution used during training and sampling.

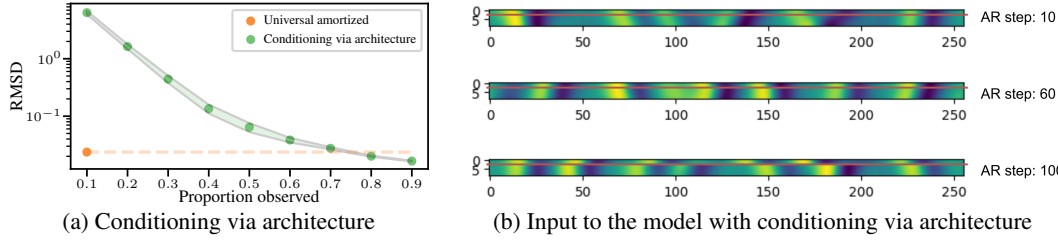

(a) Conditioning via architecture  (b) Input to the model with conditioning via architecture

Figure 29: Left: RMSD for the KS dataset on offline DA for the universal amortised model with all conditioning done via architecture. The model is compared against an identical universal amortised model that, in contrast, uses reconstruction guidance for conditioning on sparse observations. Right: The input that is used for model conditioning depending on the AR step (the longer the trajectory, the more AR steps are performed). The red line divides the previously generated $C$ states used for conditioning and the observations of the true trajectory within window $P$. The observations of the true trajectory are not fully observed by the model, but are illustrated without masks for more clarity. The vertical axis stands for the state within the window size used in the score network (9 in this case), while the horizontal one illustrates space discretisation of the PDE.

### G.2 Performance for varying window sizes

**Joint model.** We compare the performance of a joint model with window 5 ($k = 2$) with that of a model with window 9 ($k = 4$) on the KS dataset. For both models, we show results for AR sampling, as well as AAO sampling with 0, and 1 corrections (AAO (0) and AAO (1)) in Fig. 30. The window 5 model is represented with dashed lines, while the window 9 model with solid lines. For each proportion of data observed, we show the results corresponding to a value of $\gamma$ and $P \mid C$ for AR sampling that gave the best performance.

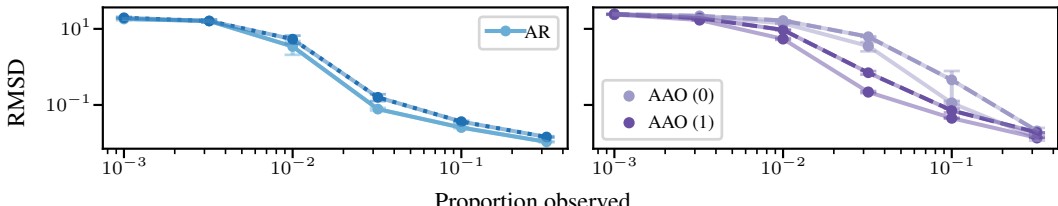

Figure 30: RMSD for the KS dataset on the offline DA task for two models, one with a window of size 5 (dashed lines), and one with a window size of 9 (solid lines). The left plot shows the results for AR sampling, while the right plot shows the results for AAO sampling with 0 and 1 corrections. The error bars indicate $\pm 3$ standard errors. Note the logarithmic y scale.

Fig. 30 shows that for both AAO and AR sampling the performance of the window 9 model is generally higher than that of the window 5 model, although for some sparsity settings the difference is not significant. The biggest improvement is seen for the middle range of sparsity (proportion of observed data between $10^{-2}$ and $10^{-1}$). We hypothesise that the reason why we do not see a significant improvement in performance between the two models in some settings is because a Markov order of $k = 2$ (corresponding to a window 5 model) is enough to deal with the task at hand, and increasing the Markov order offers limited additional useful information, leading to only marginal improvements.

### G.2.1 Comparison between AR ($k$) and AAO ($2k$)

As mentioned in Sec. 3.3, we hypothesise there are two reasons why the AR approach outperforms the AAO approach. The first reason has to do with modelling capacity—for a window of size $2k + 1$, the effective Markov order of the AR model is $2k$, while the AAO model has an effective Markov order of $k$. Secondly, in AAO sampling information needs to be propagated from the start of the sequence to its end, in order for the trajectory to be coherent. However, the score network only has a limited receptive field of $2k + 1$, limiting information propagation. The way to tackle this is through Langevin corrector steps, but they come at an additional computational cost. To empirically assess these claims in the context of DA, we compare a window 5 model ($k = 2$) queried autoregressively with a window 9 model ($k = 4$) sampled AAO. For the latter, we apply 0, 1, and 2 corrector steps (AAO (0), AAO (1), and AAO (2)).

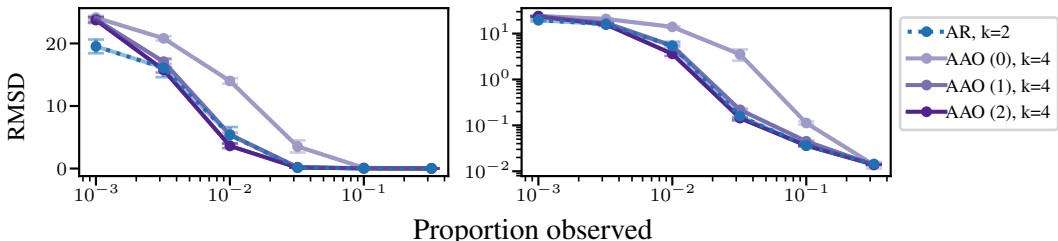

Figure 31: RMSD for the KS dataset on the offline DA task for two models, one with a window of size 5 queried autoregressively (dashed blue), and one with a window size of 9 and sampled AAO (solid purple). We show results for 0, 1, and 2 corrector steps, indicated with the $c$ in AAO ($c$). The error bars indicate $\pm 3$ standard errors. The right plot shows the same results as the left one, but on a logarithmic y scale.

The findings shown in Fig. 31 are consistent with our theoretical justification. In general, increasing the number of corrections (and hence, enhancing information propagation) increases the performance of AAO. Indeed, for the majority of sparsity levels (proportion observed $\geq 10^{-2.5}$), the performance of the AR $k = 2$ and AAO $k = 4$ with at least 1 correction become similar.

The Langevin corrector steps do not seem to make a significant difference for the two extremes considered—proportion observed of $10^{-3}$ and $10^{-0.5}$. In the first case, there are so few data observed, that the task becomes similar to forecasting. We have already shown that AAO is inefficient in that setting, regardless of whether corrections are applied or not (see Fig. 15). For a proportion observed of $10^{-0.5}$, the conditioning information is dense enough that corrections are not needed.

We can also corroborate the findings from App. F.1 regarding the performance of AAO vs. AR on forecasting with the findings from this section on DA to better understand the difference between AAO and AR sampling. The results on both tasks clearly show that a window 5 AR models ($k = 2$) does not lead to the same results as a window 9 AAO (0) model ($k = 4$), although they have the same effective Markov order. In the context of DA, the performance of the AR model can be recovered by applying at least one corrector step, which allows for more efficient information propagation. In contrast, in forecasting, applying corrector steps helps in the vicinity of the observed states, but makes little difference for long rollouts. The lack of observations (after the initial states) leads to inefficient information flow, which results in the AAO procedure failing to produce coherent trajectories. For AR, this is naturally tackled through the autoregressive fashion of the rollout.

### G.2.2 Performance for varying conditioning scenarios $P \mid C$

We study varying conditioning scenarios for the offline DA task. For the KS dataset we consider two joint models of different window sizes ($W = 5$ and $W = 9$) and show in Fig. 32 the performance for each $P \mid C$ scenario (indicated in the legend), where $P + C = W$. For each proportion of observed data and for each conditioning scenario $P \mid C$, we show the results corresponding to the best guidance strength $\gamma$.

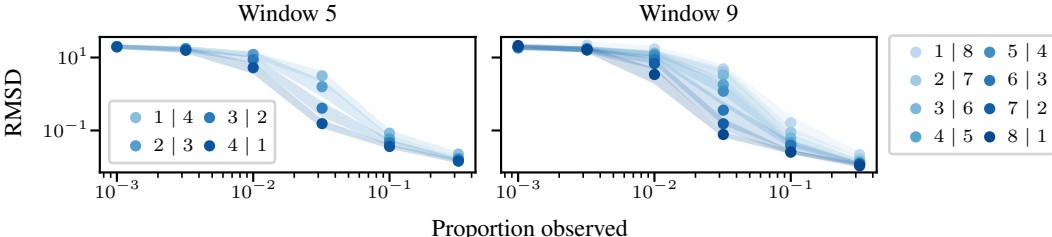

Figure 32: RMSD for the KS dataset on the offline DA task for two models, one with a window of size 5 (left), and one with a window size of 9 (right) for varying $P \mid C$ scenarios (indicated in the legend). Confidence intervals indicate $\pm 3$ standard errors. Note the logarithmic y scale.

We observe in Fig. 32 that for both window sizes the best performance is generally achieved for the larger values of $P$. This is probably due to the increasing amount of observations within the generated frames. However, for a proportion of observed data of $10^{-3}$, the observations are so sparse, that the task becomes closer to forecasting, where, as previously noted, a lower value of $P$ is preferred. In that setting, the best performance is achieved by the $1 \mid 4$ and $1 \mid 8$ scenarios, respectively, although there are only small differences between the different scenarios.

In the main paper, in Fig. 3, we show the figures corresponding to the best $P \mid C$ scenario for each proportion of observed data, with the exception of $10^{-3}$. There, we found little difference between the performance of $P \in \{1, 2, 3, 4\}$, and hence chose to show the scenario that offers the best trade-off between performance and computational complexity (i.e. $P = 4$)—an increase in RMSD of $0.25 (\approx 1.4\%)$ for an almost four-fold decrease in sampling time.

In Fig. 33a we show the performance of the amortised model with window 9 for different $P$ and $C$ for KS. The best performance is achieved by the $6 \mid 3$ model across most of the proportions of data observed. The universal amortised model is again more stable compared to the joint one when varying $P$ and $C$, however it struggles to get the same level of performance for intermediate values of observed data considered. Fig. 33b shows the same for the Kolmogorov dataset. In contrast to KS,

the best result is achieved by the model that has the longest history $(1 \mid 4)$. The performance of the model is again more stable for different $P \mid C$ scenarios compared to the Joint AR.

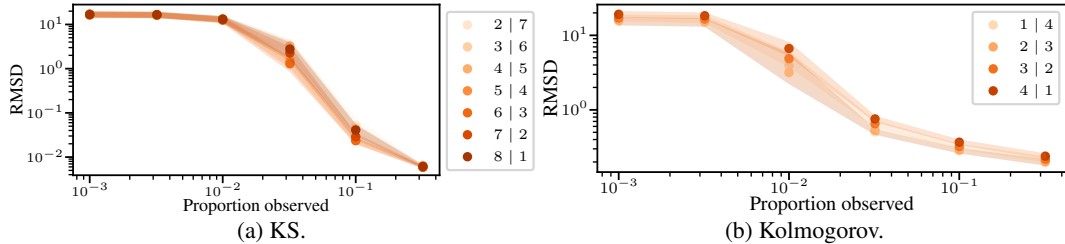

Figure 33: RMSD for the KS (left) and Kolmogorov (right) datasets on the offline DA task for the universal amortised model for varying $P \mid C$ scenarios (indicated in the legend). Confidence intervals indicate $\pm 3$ standard errors. Note the logarithmic y scale.

We performed the same analysis for the Kolmogorov dataset and show the results in Fig. 34. In this case we only study a model of window 5 for both the joint and universal amortised models. We observe similar findings as in the KS dataset for the joint model, with lower $P$ values being preferred for lower proportions of observed data, and higher $P$ when the amount of observations increases.

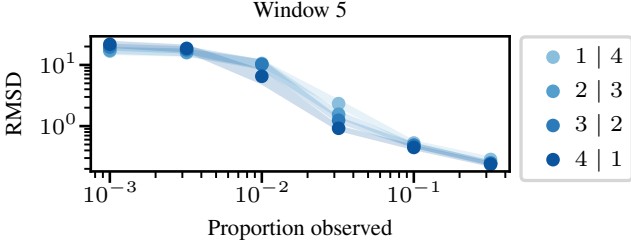

Figure 34: RMSD for the Kolmogorov dataset on the offline DA task for the joint model with window size 5 for varying $P \mid C$ scenarios (indicated in the legend). Confidence intervals indicate $\pm 3$ standard errors. Note the logarithmic y scale.

### G.3 Comparison to interpolation

In this section we perform a quantitative and qualitative comparison to a simple interpolation baseline. This was implemented using the interpolation functionality within the Scipy library [74]. For KS, we investigated three interpolation methods: linear, cubic, and nearest, and report the results for the method that gave the best results for each percentage of observed data. For Kolmogorov, we investigated linear and nearest interpolation, since the cubic method is not applicable to 3D data (2 spatial dimensions and 1 time dimension).

We present the results in Fig. 35. For the 1D KS dataset, all the methods studied in the main text (joint AR, AAO, and universal amortised) generally outperform the interpolation baseline for the sparse regimes (up to $10^{-1}$ proportion observed). The only exception is for universal amortised with $10^{-2}$ data observed, where the performance is on par. As more data is observed, the performance of all five methods becomes similar, indicating that the field can be easily reconstructed based on the observations alone. For the two sparsest regimes, linear interpolation gave the best results, while for the last four, we used cubic interpolation.

For Kolmogorov, there is a significant gap between the AR methods and the interpolation baseline, potentially indicating that interpolation is not as efficient when the number of data dimensions grows. The AAO methods also significantly outperform the interpolation baseline for all but the sparsest regimes. All results correspond to linear interpolation, since this gave the best performance.

For both datasets, we also provide a qualitative comparison between the joint AR predictions and the interpolation predictions, alongside the ground truth and the observed data. This is illustrated for two levels of sparsity for each dataset ($10^{-2}$ and $10^{-1}$). Fig. 36 shows the results for the KS dataset. For the smaller proportion of data observed, the interpolation baseline produces unphysical behaviour, possibly because it does not contain any prior knowledge about the dynamics of the dataset and

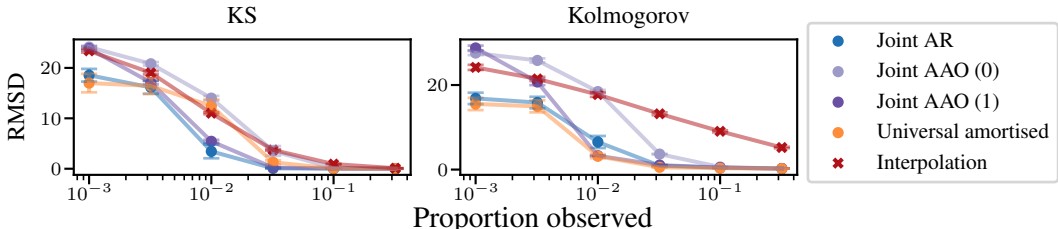

Figure 35: RMSD (mean $\pm 3$ standard errors) for the KS (left) and Kolmogorov (right) datasets on the offline DA task for varying sparsity levels. We show the results in Fig. 3, alongside an interpolation baseline (red). Note the logarithmic x scale.

the observations alone do not offer enough information for good trajectory reconstruction. In the case of the joint AR model, the learned dynamics from the prior score make the predictions more consistent with the dynamics describing the KS dataset. When the data is dense enough, even a simple interpolation baseline gives reasonable predictions (Fig. 36b).

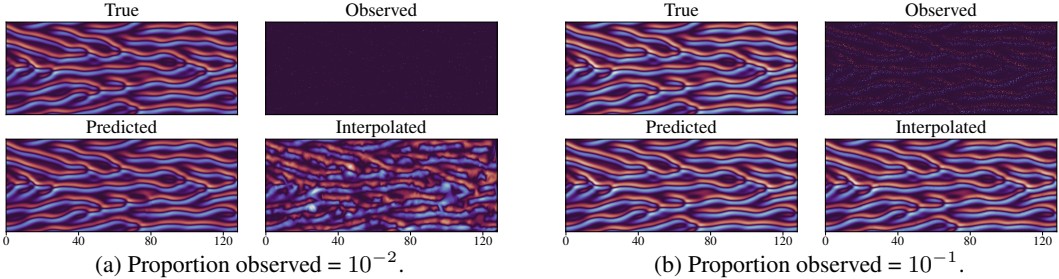

(a) Proportion observed = $10^{-2}$.

(b) Proportion observed = $10^{-1}$.

Figure 36: Qualitative comparison between the true, predicted (with the joint AR model), and interpolated trajectories, as well as the observed values for the KS dataset. The left plot corresponds to a proportion of observed data of $10^{-2}$, while the right corresponds to $10^{-1}$.

In contrast, for the Kolmogorov dataset, the interpolation results still show unphysical behaviour even for the higher percentage of data observed ($10^{-1}$). This is probably because of the higher dimensionality of the data, as well as because we are plotting the vorticity (curl of velocity field), rather than the predicted velocity fields, which involves differentiating the predictions. Overall, the visual quality of the predictions outputted by the methods studied in the main text are clearly superior to the interpolation baseline.

## H  Additional experimental results on online data assimilation

In this section we provide a more detailed explanation of the setup from the online DA task and provide results for several conditioning scenarios for the joint and amortised models.

The online DA scenario goes as described in Sec. 5.3. In Fig. 38 we also present a diagram, showing an example of which observations would be available at each DA step in the KS dataset. The darker dots indicate observations, while the lighter background indicates that no observations are available in that region at that DA step. The example corresponds to a proportion of data observed of $0.1$.

We study varying conditioning scenarios for the online DA task for the joint and universal amortised models. We do not show results for $P = 1$ due to the excessive computational time required for sampling, and only study the results for one window size for each dataset: window 9 for KS and window 5 for Kolmogorov.

**Joint model.**    For the window 9 joint model, we show the results on the KS dataset corresponding to the best guidance strength out of $\gamma \in \{0.03, 0.05, 0.1\}$ for AR sampling. For AAO sampling we studied $\gamma \in \{0.01, 0.03, 0.05, 0.1, 0.5\}$ with 0 and 1 corrections. We found that the results did not significantly vary with $\gamma$, with $\gamma = 0.01$ giving marginally better performance, and lower values giving unstable results. In Tab. 2 we show the results corresponding to $\gamma = 0.01$ for AAO with 0 (AAO (0)) and 1 (AAO (1)) corrections, and for $\gamma = 0.05$ and $4 \mid 5$ for AR.

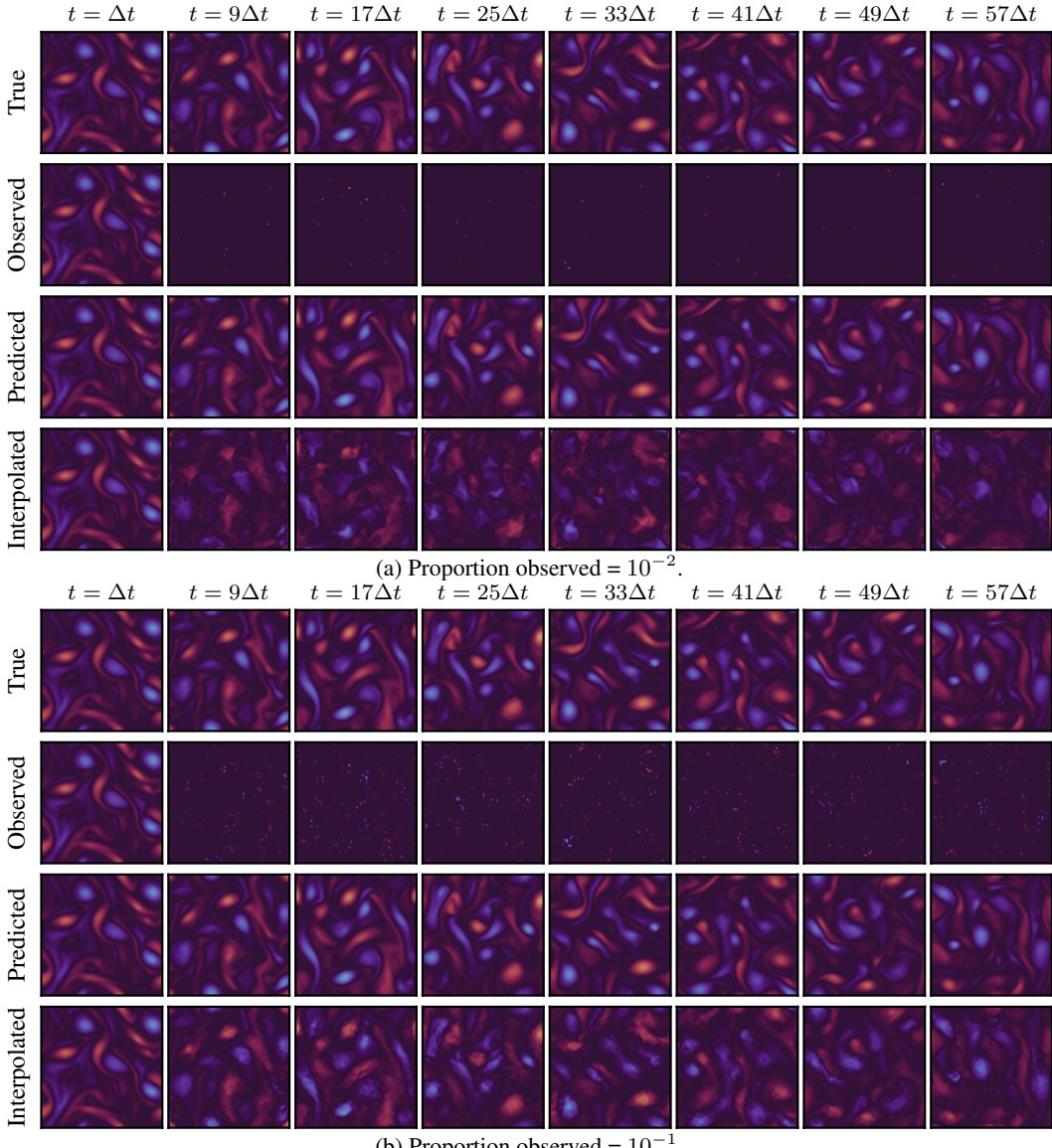

(a) Proportion observed = $10^{-2}$.

(b) Proportion observed = $10^{-1}$.

Figure 37: Qualitative comparison between the true, predicted (with the joint AR model), and interpolated trajectories, as well as the observed values for the Kolmogorov dataset. The top plot corresponds to a proportion of observed data of $10^{-2}$, while the bottom corresponds to $10^{-1}$.

Fig. 39a shows that the joint model has a similar performance for $P \in \{4, 5, 6\}$, with the lowest RMSD given by 6 | 3. Lower values of $P \in \{2, 3\}$ give poorer performance, just as in the offline DA task. Higher values of $P \in \{7, 8\}$ also perform worse, just as in the forecasting task. Thus, the fact that the middle values of $P$ perform best is probably due to the fact that the online DA task is a mix between DA and forecasting.

For the Kolmogorov dataset we studied $\gamma \in \{0.03, 0.05, 0.1, 0.3\}$ for a model with window 5 queried AAO. We employed 0 (AAO (0)) and 1 (AAO (1)) corrections. Similarly to the KS dataset, we found little difference between the $\gamma$ settings, and found that $\gamma = 0.03$ gave marginally better performance. For AR, we studied $\gamma \in \{0.03, 0.05, 0.1, 0.2\}$ for each possible value of $P$ ($P \in \{1, 2, 3, 4\}$). We found $\gamma = 0.03$ to be unstable for low values of $P \in \{1, 2\}$. The best setting was achieved for 2 | 3 and $\gamma = 0.05$. We show the results corresponding to the best $\gamma$ setting in Fig. 39b.

**Universal amortised model.** For the universal amortised model, we use the best $\gamma = 0.05$ and $\sigma_y = 0.001$ parameters found for the offline DA for the same proportion of observed data, due to

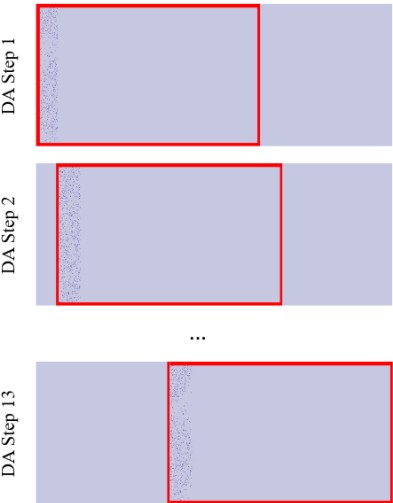

Figure 38: Setup for the DA online task for the KS dataset for a proportion of observed data of 0.1. Every 20 time steps, we get access to sparse observations from blocks of 20 states. At the first DA step we predict the first 400 states based on the sparse observations from the first 20 states. At the next DA step, we use the previously-generated forecast, as well as the new sparse observations coming from the next block (time step 20-40) to predict the next trajectory of length 400 (from time step 20 to 420). This repeats until the end of the sequence (640 time steps).

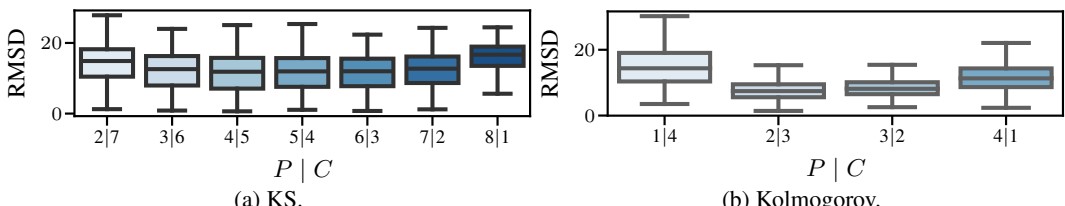

(a) KS.                                     (b) Kolmogorov.

Figure 39: RMSD for the joint model on the KS (left) and Kolmogorov (right) datasets for varying $P \mid C$ scenarios. For KS we use a window 9 model, whereas for Kolmogorov we use a window 5 model.

computational constraints. Fig. 40a shows the performance on the KS dataset for each $P \mid C$ scenario, where $P + C = W$. Fig. 40a indicates that the universal amortised model shows stable performance across different $P$ and $C$, with the setting of $5 \mid 4$ being slightly better than the other ones. Fig. 40b shows the performance on the Kolmogorov dataset for each $P \mid C$ scenario, where $P + C = W$. Fig. 40b suggests that conditioning the model on larger history results in better performance. Even though $1 \mid 4$ shows the best result, all other conditioning scenarios also have competitive performance compared to both AAO(1) and Joint AR models.

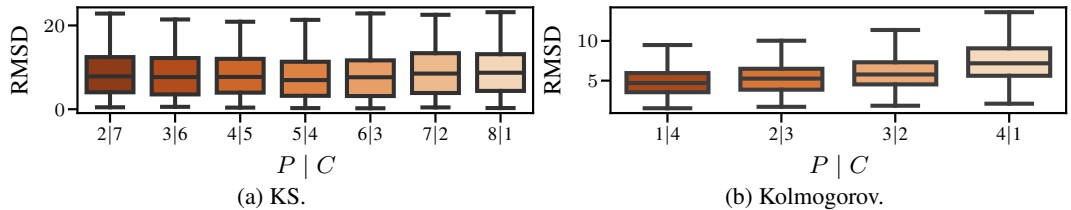

(a) KS.                                     (b) Kolmogorov.

Figure 40: RMSD for the universal amortised model on the KS (left) and Kolmogorov (right) datasets for varying $P \mid C$ scenarios. For KS we use a window 9 model, whereas for Kolmogorov we use a windw 5 model.

# I Frequency Analysis

Lippe et al. [42] showed that capturing high-frequencies in the spectrum is crucial for accurate long-term rollouts. Following their analysis we compute the frequency spectra of a one-step prediction for all the methods considered in Sec. 5 for both KS and Kolmogorov. In Fig. 41, we compare the spectrum of a one-step prediction at the beginning and end of the trajectory. In the case of KS, the spectra of the states generated by our models do not vary significantly depending on how far away they are from the initial state. In contrast, PDE-Refiner significantly overestimates the amplitude of high frequencies as the states depart from the initial conditions. However, we can see that at the beginning of the trajectory, PDE-Refiner can capture higher frequencies more accurately than all the considered methods.

For Kolmogorov the frequency spectrum is not as varied as the one from KS. Possible reasons include the coarser spatial discretisation used or the nature of the underlying PDE. We compute and plot the average frequency spectrum for the two channels (i.e. the two velocity components). From Fig. 42, we do not see much difference between the spectra of different models—all models are capable of capturing the ground truth spectrum well.

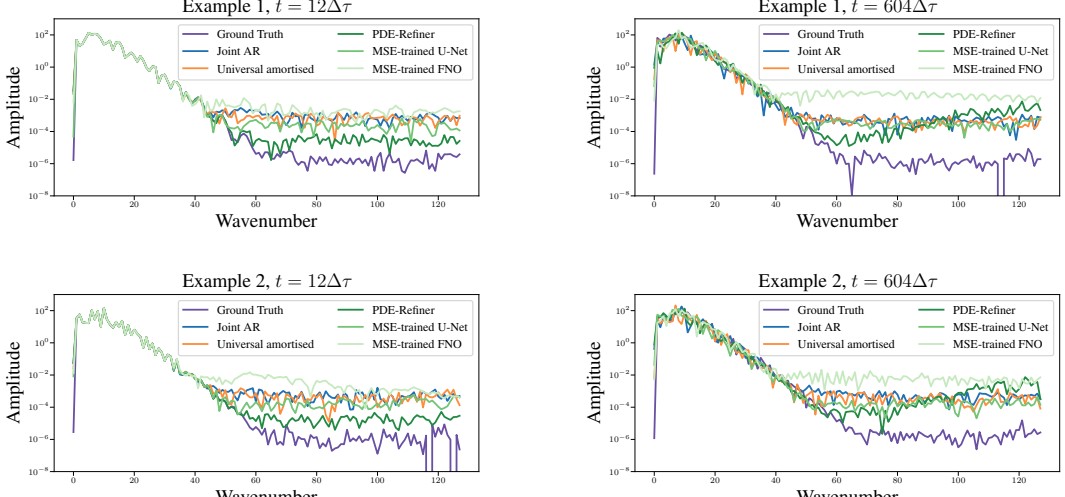

Figure 41: Analysis of the spectra of two KS trajectories for different methods (Joint AR, universal amortised, PDE-Refiner, MSE-trained U-Net, MSE-trained FNO). Each row corresponds to a different trajectory. The first column shows the spectra of states closer to the initial state, while the second corresponds to states further away.

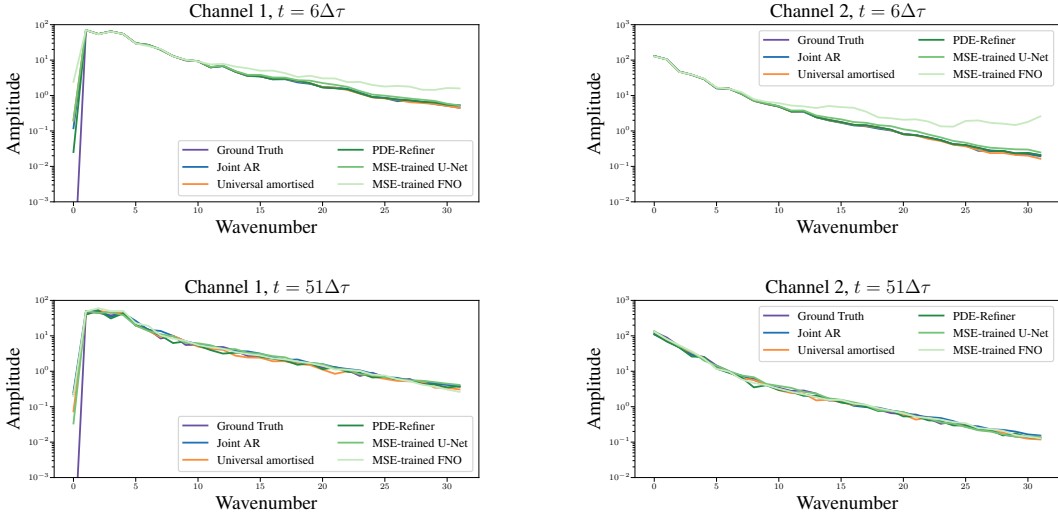

Figure 42: Analysis of the spectrum of a Kolmogorov trajectory for different methods (Joint AR, universal amortised, PDE-Refiner, MSE-trained U–Net, MSE-trained FNO). Each row corresponds to a different time step (left - closer to the initial state, right - closer to end of trajectory), and each column to a different channel.

## J Stability of long rollouts

To investigate the long-term behaviour of the different diffusion models, we generate a very long trajectory ($T = 2000\Delta\tau$, corresponding to $400$ seconds) for the KS equation. We expect the samples to diverge from the ground truth after a certain number of states, but the question is whether our models still generate physically plausible predictions. We illustrate in Fig. 43 an example of a long trajectory generated by the joint AR (middle) and universal amortised (bottom) models, showing that the generated states remain visually plausible throughout the trajectory. We also analyse the spectra of different one-step predictions along the trajectories in Fig. 44. We can see that both models manage to capture the low frequencies well, but overestimate the high-frequency components. This holds for all the different steps we considered—a state close to the initial (known) ones, a state in the middle of the long trajectory, and a state towards the end of the trajectory. This implies that the spectra do not change qualitatively along the trajectory length and hence, remain physically plausible throughout the trajectory.

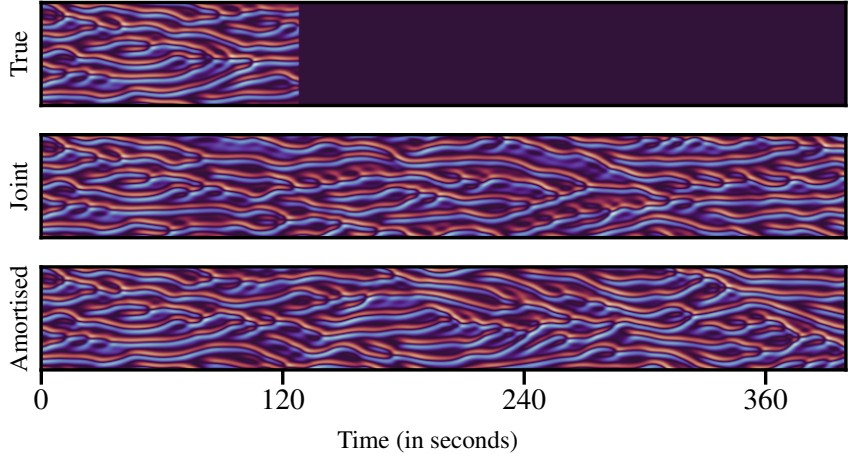

Figure 43: Example of a long trajectory ($2000\Delta\tau$ steps, corresponding to $400$s) generated by the joint AR (middle) and universal amortised (bottom) models. Note how, even if we are generating longer trajectories than what the models have been trained on, the states remain visually plausible.

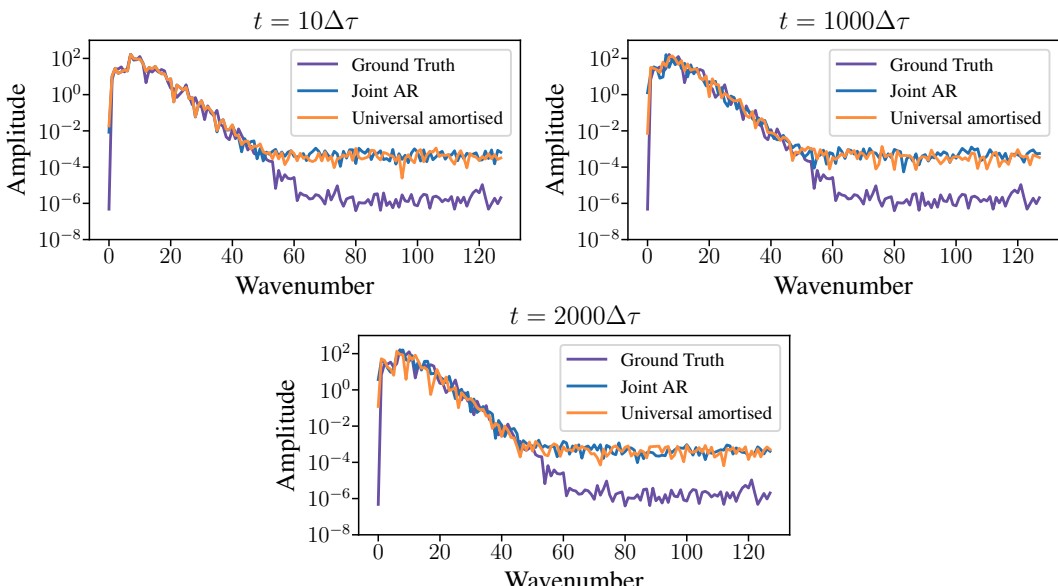

Figure 44: Frequency spectra of a state close to the initial state (top left), in the middle of the trajectory (top right), and towards the end of the trajectory (bottom). In each plot, we show the ground truth spectrum (blue), as well as the spectra of the trajectories generated by the joint AR (blue) and universal amortised models (orange).

# K  Algorithms

---

**Algorithm 1** Predictor-corrector (PC) sampling as implemented in [67, 60]. We will refer to this as the function `predictor-corrector-step(N,CS)`

---

**Input:** Discretisation steps $N$ for the reverse-time SDE, corrector steps $CS$
**Output:** Trajectory $x(0)$
Initialise $x(T) \sim p_T$
**for** $i = N - 1$ **to** $0$ **do**
    $x(t_i) \leftarrow$ Predictor $(x(t_{i+1}))$          ▷ Using Eq. (5) as score
    **for** $j = 1$ **to** $CS$ **do**
        $x(t_i) \leftarrow$ Corrector$(x(t_i))$          ▷ Using Eq. (5) as score
    **end for**
**end for**
**return** $x(0)$

---

**Algorithm 2** All-at-once (AAO) rollout as in [60]

---

**Input:** Trajectory length $L$, conditioning states $x_{1:C}$, discretisation steps $N$ for the reverse-time SDE, corrector steps $CS$
**Output:** Trajectory $x_{1:L}$
$x_{1:L}(0) \leftarrow$ `predictor-corrector-step(N,CS)`      ▷ Using Algorithm 2 from [60] to get an approximation of the score of the full trajectory and using guidance
**return** $x(0)$

---

**Algorithm 3** AR rollout with a joint model

**Input:** Trajectory length $L$, Markov window $M$, initial true conditioning states $x_{1:C}$, prediction states $P = M - C$, discretisation steps $N$ for the reverse-time SDE, corrector steps $CS$
**Output:** Trajectory $x_{1:L}$
Initialize $x_{\text{gen}} \leftarrow [x_{1:C}]$
**while** $\text{len}(x_{\text{gen}}) \leq L$ **do**
    Define conditioning states as $x_{\text{gen}}[-C:]$         $\triangleright$ Selecting last $C$ states from $x_{\text{gen}}$
    $x_{1:M}(0) \leftarrow \texttt{predictor-corrector-step(N,CS)}$         $\triangleright$ This has length $M$
    $x_{\text{gen}} \leftarrow x_{\text{gen}} + [x_{1:M}(0)[C:]]$         $\triangleright$ Selecting the last $P$ states
**end while**
$x_{1:L} = x_{\text{gen}}$
**return** $x_{1:L}$

---

**Algorithm 4** AR rollout with a universal or classic amortised model

**Input:** Trajectory length $L$, Markov window $M$, initial true conditioning states $x_{1:C}$, prediction states $P = M - C$, discretisation steps $N$ for the reverse-time SDE, corrector steps $CS$
**Output:** Trajectory $x_{1:L}$
Initialize $x_{\text{gen}} \leftarrow [x_{1:C}]$
**while** $\text{len}(x_{\text{gen}}) \leq L$ **do**
    Define conditioning states as $x_{\text{gen}}[-C:]$         $\triangleright$ Selecting last $C$ states from $x_{\text{gen}}$
    $x_{1:P}(0) \leftarrow \texttt{predictor-corrector-step(N,CS)}$         $\triangleright$ This has length $P$
    $x_{\text{gen}} \leftarrow x_{\text{gen}} + [x_{1:P}(0)]$
**end while**
$x_{1:L} = x_{\text{gen}}$
**return** $x_{1:L}$

