# OpenReview forum: "On conditional diffusion models for PDE simulations"
_NeurIPS.cc/2024/Conference — NeurIPS 2024 poster_

### Official Review · Reviewer_D3JV · 2024-07-11

**Soundness:** 3
**Presentation:** 3
**Contribution:** 3
**Rating:** 7
**Confidence:** 4

**Summary:**

The paper studies forecastiong temporal dynamics associated with forward PDE problems using diffusion models. Different strategies for approximating the score function of trajectories and how to parameterize them are proposed, ranging from sampling the whole trajectory all at once to sample it step by step autoregressively. In addition, the authors also propose to incorporate information from observation to refine and modulate the prediction process. The proposed methods are tested on two types of PDEs under different settings involving both forecasting and data assimilation.

**Strengths:**

1. The writing of the paper is pretty clear and self-contained.

2. Comprehensive empirical studies of different components and hyperparameters are carried out to support the claims.

3. The proposed universal amortized model, while being straightforward, is quite effective and flexible. It resolves several issues existed in common amortized models, including fixed context windows and performance degradation when given a longer context.

4. Applying reconstruction guidance to the diffusion process can be potentially useful for building hybrid forecasting system, e.g. combing data-driven forecasting model with some physics prior.

**Weaknesses:**

1. Many techniques in the proposed framework are not new. Using the local segment's score to approximate and decompose the score of global dynamics is introduced in Score-based Data Assimilation. Reconstruction guidance has been used in several previous works that applied diffusion model to inverse problems (e.g. Manifold Constrained Gradient).

2. The empirical performance of the proposed method does not outperform many data-driven forecasting models. Nonetheless, as authors have stated, the objective of the proposed framework is not only to do pure data-driven forecasting but do flexible and versatile sampling tasks such as online data assimilation. However, I think flexibility does not necessarily have to be orthogonal to the performance (see question section)

**Questions:**

1. In its current form, the best model variant of the proposed method still falls behind PDE-Refiner. Given PDE-Refiner is also closely related to denoising diffusion model (with velocity prediction, starting from last time step instead of pure noise), it seems that it is also possible to apply many of the proposed techniques to PDE-Refiner. More specifically, according to my understanding, PDE-Refiner uses the amortized architecture (P=1 and C=1) and it is autoregressive, while the paper has showcased the proposed universal amortized method seems superior (can predict more time steps at once, flexible time window and can use longer context). Can the authors provide more comments related to this point?

2. The major limitation of applying vanilla diffusion model to simulate PDE autoregressively is that it increases the NFE by several times (e.g. if the diffusion model needs k steps to sample one step then it will need kT NFE to sample T steps). The proposed method can alleviate this by doing joint sampling and predicting multiple future steps in one sampling loop, yet not too many study and comparison on NFE is presented in the main text. It will be clearer if computational cost statistics regarding the proposed method and existing methods like PDE-Refiner are also reported.

3. Compared to pure data-driven forecasting models like using UNet trained with MSE target, how much performance gain does online data assimilation bring? (the authors have conducted experiments on varying portions of observed data but it can be helpful to see a comparison against method that cannot do data assimilation)

**Limitations:**

The authors have discussed the limitation in the conclusion section.

---

> ### Author Rebuttal · Authors · 2024-08-06
>
> We thank the reviewer for the thoughtful consideration of our work. We have taken the time to address all the points raised in the review.
>
> **Novelty** - While we acknowledge that the techniques per-se are not new, we were the first to:
> 1) quantitatively evaluate the decomposition on physics data;
>
> 2) propose AR sampling instead of AAO sampling, which works much better in scarce observed data regimes and is key to performing forecasting;
>
> 3) use both reconstruction guidance and amortisation on different sets of variables simultaneously, which improved model performance and flexibility.
>
> **Flexibility vs performance** - We agree with the reviewer that flexibility does not necessarily have to be orthogonal to performance. One of the goals of the paper is to quantitatively evaluate the current status of diffusion models for PDE simulation compared to other deep learning approaches by highlighting pros and cons of each such approach on the considered tasks (forecasting, DA, and a mix thereof). Unsurprisingly, we found that in the case of pure forecasting, our methods lag behind SOTA methods that have specifically been designed for forecasting. However, we argue that even so, the forecasting performance of our methods is close to other commonly used ML-based benchmarks, while benefiting from a much more flexible conditioning framework.
>
> **Universal amortised training for PDE-Refiner** - Indeed, the “universal” amortized training and larger window size can be easily incorporated into PDE-Refiner. However, reconstruction guidance and data assimilation cannot be applied out-of-the-box, as PDE-Refiner first produces an estimate of the next state, and then refines this estimation using noise. This is in contrast to plain diffusion models that gradually build this estimation from pure noise, rather than from the neural operator prediction.
>
> **NFE** - We provide a more detailed analysis of the NFE below in the KS and Kolmogorov cases.
>
> For KS we simulate $640$ states. For the joint AR and amortised models, we use $128$ diffusion steps in the paper with a window $9$ model, and the conditioning scenario $P \mid C$ depends on the task. Overall, the computational cost is $(\frac{640-9}{P} + 1) \times 128$. In contrast, PDE-Refiner first produces the NO estimate, and then performs $3$ refining steps, leading to $4$ FE for each state. They always predict one state at a time, leading to NFE=$640 \times 4$, indeed leading to a lower computational cost than our approaches.
> However, subsequent analysis on the amortised model indicates that we can achieve similar performance with the universal amortised model by performing as few as 16 diffusion steps (please also see Fig. 7 in the attached PDF) (This is not true for the joint AR approach). This would lead to NFE = $(\frac{640-9}{P} + 1) \times 16$, which becomes comparable (and depending on $P$, even lower) than PDE-Refiner cost.
>
> For Kolmogorov, the same analysis holds, but we only need to simulate $64$ states, and found that for the universal amortised model we can also achieve good performance with $16$ diffusion steps.
>
> Thus, the universal amortised approach is comparable to PDE-Refiner in terms of computational time, while we acknowledge that the joint AR approach is more computationally intensive.
>
> **MSE-based U-Net for offline DA** - In the online data assimilation case, we assume that in the first step we do not observe the initial condition (lines 346-348), but only some sparse observations within the $s$ first time steps. Hence, plain autoregressive U-Net trained with MSE target cannot be directly applied to this task, as it expects to have fully observed initial state.
>
> However, we have included a comparison in the offline DA scenario, where we report the RMSD for the MSE-trained U-Net based on the initial conditions alone (please see Fig. 8 in attached PDF).
>
> We hope we have successfully addressed your concerns.

---

> > ### Comment · Reviewer_D3JV · 2024-08-10
> > **Reply to the rebuttal**
> >
> > I would like to thank the authors for the efforts and clarification, the majority of the concerns and questions I raised have been addressed and clarified. I think the proposed framework is very flexible and versatile for forecasting/inverse problems, despite the performance is not sota yet but I believe there is potential for a lot of improvements. The authors have also carried out comprehensive empirical study, which is informative will be practically beneficial to many data-driven PDE applications. I've raised my score to 7.

---

> > > ### Author Response · Authors · 2024-08-10
> > >
> > > We greatly appreciate your feedback and thank you for taking the time to go through our paper and rebuttal. There are indeed opportunities for improvement and we are excited to pursue them further.
> > >
> > > Thank you for increasing your score and for acknowledging the flexibility of our framework once again.

---

### Official Review · Reviewer_4n49 · 2024-07-12

**Soundness:** 3
**Presentation:** 3
**Contribution:** 3
**Rating:** 6
**Confidence:** 3

**Summary:**

The manuscript addresses the challenging problem of conditional diffusion modeling, towards accurate and efficient data assimilation in problems governed by PDE. The authors compare different diffusion modeling approaches in this setting, focusing on the auto-regressive setting, and also propose new approaches. The comparisons are done on relevant PDE benchmarks (chaotic, 1d and 2d spaces).

**Strengths:**

Using conditional diffusion models for data assimilation is an interesting approach, and the results on the given PDE systems are impressive. The paper is relatively easy to read, and contains interesting suggestions for improvements. I very much appreciated the brief introduction to score-based modeling, as I am not working day-to-day with diffusion models (more with PDEs). Scalability of the approaches are also addressed, briefly in the main part and more detailed in the appendix. In general, the appendix is also extensive and contains a lot of details and additional studies related to the main paper.

**Weaknesses:**

1) The PDEs considered as examples have turbulent / chaotic dynamics. There is not discussion about this, but it is important to mention: comparing trajectory data is not a good metric, because the PDE itself is causing sensitivity to the initial conditions, not the models. Rather, the PDE itself could be used as a metric (i.e., evaluate the PDE on the generated trajectory and check if it is zero); alternatively, Lyapunov exponents or density measures on the attractors can be used for comparison. The 1D Burgers equation does not have these problems. The "correlation time" defined by the authors is probably a way to mitigate the issues with chaotic/sensitive dynamics, but is not ideal, because the divergence from the true solution depends on the initial condition. Regardless of this issue with the metric, it seems the trained models are capable of generating trajectories close to the attractor (i.e. they "look reasonable"), which is already impressive.

2) There is no comparison to a classical scheme that solves the inverse problem. Especially looking at the long solution times (fig.3, between 25-100 minutes); I think it is possible to design a classical inverse problem solver that (a) starts with a random initial state of the PDE (b) solves the PDE with this guess, and then (c) updates the guess based on the error to the observations, and does all this iteratively in less than 25 minutes. It is ok to study how diffusion models perform in this setting, but it is not ok to not even mention this possibility of the classical approach in the related work. Many inverse problem solvers exist in this direction (also data-driven ones), a good overview is here:

Arridge, S., Maass, P., Öktem, O., Schönlieb, C.-B., 2019. Solving inverse problems using data-driven models. Acta Numerica 28, 1–174. https://doi.org/10.1017/S0962492919000059

It would be good to include a brief discussion of the benefits of the neural network / diffusion model approach compared to the more classical setting.

3) There is no theoretical justification (meaning: no rigorous proofs) of the suggested new architectures and training methods. The new approaches are motivated plausibly in the main paper, with additional computations done in the appendix, but without theory.


Minor:
1) l103: the statement $\mathcal{P}(u;\alpha)=\mathcal{P}(\partial_z u,\dots)$ is not correct. Does $\mathcal{P}$ only depend on $u$ or only (?) on its derivatives? Are PDEs  like $<v,\partial_z> u + u=0$ not included (because they include $u$ as well as $\partial_z u$)? Probably this is just an abuse of notation here.
2) l126: do the authors mean "...time dependency [...] $=x_{1:L}(t)$" instead of "$x_{1:L}(0)$"?

**Questions:**

1) l96: how can we divide by $\mu_t$ if it is a vector? Is it done component-wise?
2) It is probably my lack of experience with diffusion models, but how do we know the "true" probability density p to compare to (resp, $\nabla \log p$)? For example, in equation (3) and l.138, "the local score network ... is trained by minimizing the following DSM loss", which contains the expression $\nabla \log p$. This is probably not contained in the training data as a value for each x, right? How do we compute it? Equation 7 does not help, because it is also not clear to me how to compute the individual probability densities (for each i). In Equation 8 I can understand that the conditional is assumed to be Gaussian, which then leads to a simple expression, but it cannot be the case for the general density.

3) l305: "We assume to always observe some full initial states" this is very confusing to me. What is "some full"? Does it mean the full initial state is observed? This would invalidate the entire approach, because then there is no need for "data assimilation", the PDE can just be solved with a classical solver, from this given initial condition. Looking at appendix G, it seems that only a portion of the initial state is "always" given, which would be ok - then the classical approach is not really possible - but then the sentence should be rewritten to clarify this.

**Limitations:**

Some limitations are discussed, but not in the direction of a comparison to classical solvers (as I mentioned in the "weaknesses" section). It is not clear if the diffusion based modeling can be better than classical solution methods for the particular problems, and if not, which other problems it can be used on that are hard to solve with classical methods.

---

> ### Author Rebuttal · Authors · 2024-08-06
>
> We thank the reviewer for the positive consideration of our work and the useful feedback on it.
>
> **Turbulent dynamics + metrics** - Thanks for your suggestion, we will add a mention about the turbulent dynamics of the data. For a discussion about metrics and spectrum, please refer to common answer.
>
> **Classical solvers vs ML-based + runtime** - This is a great point and we will include a discussion in the updated manuscript. We believe ML-based approaches for PDE modelling are still fairly new and thorough comparisons between them and classical solvers will emerge. In the case of our paper we considered implementing such methods is out of the scope, since the focus is on ML-based solutions, with emphasis on diffusion models. But we agree that we should mention the possibility of implementing the classical approach.
>
> A motivating application of our work is ML-based approaches for weather modelling. For a more detailed description of the weather forecasting setup please see reply to Reviewer D4vs. Works in this field, such as Aardwark Weather [1] show that there is potential for ML-based methods to reduce the cost of medium-range weather forecasting (which is currently based on classical solvers) by orders of magnitude, while still achieving competitive performance. There have also been works proposing to tackle the forecasting component using diffusion-based models [2], where the probabilistic treatment is appealing - it allows one to predict a range of probable weather scenarios, which can be crucial for decision making, disaster prevention, etc.
>
> Regarding runtime, as pointed out in the common response, the universal amortised model can achieve a significant speed-up (8x faster) during sampling compared to what we report in Figure 3 in the main paper.
>
> We will incorporate a subsection on classical solvers in the related work section, mentioning their applicability in the tasks considered. We will also add a discussion on how ML-based methods compare to classical solvers.
>
> **Theoretical justification** - We acknowledge that classical methods benefit from a much more mathematically-rigorous framework. Unfortunately, guarantees for ML-based methods are rarely available (not only in the context of PDE modelling). Diffusion models have mathematical foundations (e.g. [3] or [4]), but it is still non-trivial to develop guarantees when taking into account the complex network dynamics within diffusion models. As such, this is probably more of a limitation of the field rather than of this paper in particular, and it is unclear whether such guarantees will ever be available for ML-based methods.
>
> **W1minor** - Thanks for pointing this out. The dependence on $u$ should be included as well.
>
> **W2minor** - L126: We omit putting (0) as time dependency and just use $x_{1:L}$. The initial Markov assumption is introduced on the sequence coming from the original data distribution.
>
> **$\mu_t$** - Yes, $\mu_t$ is a scalar value and then we divide the numerator component-wise.
>
> **Denoising score matching** - Indeed, computing $\nabla_x \log p(x)$ is not easy as we do not know $p(x)$. The workaround is to use the denoising score matching loss (Eq. 3), where we use $\nabla_x(t) \log p(x(t)|x(0))$ instead of $\nabla_x \log p(x)$. For a justification of why this is the case, see Appendix of [5]. Then, the quantity of interest becomes $p(x(t)|x(0))$, which is referred to as the noising kernel in diffusion models. This can be derived from the governing noising SDE. In particular, if the drift and the diffusion coefficient are affine transformation of $x(0)$, then $p(x(t)|x(0))$ is Gaussian. In our case, we are using an Ornstein-Uhlenbeck process (see Eq. 1) as noising process and therefore this satisfies the above conditions making $p(x(t)|x(0))$ Gaussian.
>
> **Offline DA setup** - The initial states are fully observed in the offline DA setting - we found this to be a convenient setting to analyse the performance of the model in a variety of settings, going from almost pure forecasting (the sparsest setting) to a very dense observation regime. We understand that in the context of classical solvers, where the treatment is deterministic, this just collapses to plain forecasting. However, sparse observations can potentially be helpful in reducing accumulating approximation errors (resulting from e.g. space/time discretisation). Nevertheless, this was not the focus of the experiment, but rather to see whether our approaches can “correct” their forecasting mistakes (i.e. make use of the observations to bring the predictions closer to ground truth), and alleviate error accumulation which is an inherent issue of autoregressive approaches.
>
> In the online DA experiment, the initial states were **not** fully observed, and thus, could not be tackled by a classical numerical solver. This is the task we believe best reflects the advantages of our approaches, and acted as the main motivation when developing them. The previous investigations on forecasting and offline DA were useful investigations to understand the workings of the models in a variety of conditioning scenarios, but we consider the online DA task to be the one with most relevance to upstream applications, such as in weather modeling.
>
> **References**
>
> [1] Vaughan, A. et al. (2024). Aardvark weather: end-to-end data-driven weather forecasting.
>
> [2] Price I. et al. Gencast: Diffusion-based ensemble forecasting for medium-range weather, 2024.
>
> [3] De Bortoli, V. Convergence of denoising diffusion models under the manifold hypothesis. TMLR (2022).
>
> [4] Benton, J. et al. (2024). Nearly d-linear convergence bounds for diffusion models via stochastic localization. ICLR
>
> [5] Vincent P. A connection between score matching and denoising autoencoders, 2010.

---

> > ### Comment · Reviewer_4n49 · 2024-08-10
> >
> > Thank you for the clarifications. I appreciate that you will update the manuscript with a discussion on classical solvers. It is vital to clarify this point in the manuscript: ML-based solvers for inverse problems involving PDE are still in their early phase. This should not keep anyone from conducting research, but the current ML approaches are outperformed by classical approaches in almost all settings, certainly (as the authors mention) related to theoretical understanding. Weather predictions may be a good application for ML these days, but it is not part of the current work. I will keep my score at 6.

---

> > > ### Author Response · Authors · 2024-08-10
> > >
> > > We would like to thank the reviewer once again for raising this important point and, as already mentioned, we will make sure to include it in the updated manuscript.
> > >
> > > We greatly appreciate the time taken to go through our paper and our rebuttal.

---

### Official Review · Reviewer_ofj7 · 2024-07-15

**Soundness:** 3
**Presentation:** 2
**Contribution:** 3
**Rating:** 6
**Confidence:** 3

**Summary:**

This paper studies the application of diffusion models (DMs) to PDE forecasting and data assimilation, focusing on different approaches to condition the DMs on initial conditions (forecasting) or sparse observations (data assimilation). The conditioning can either occur explicitly during training by conditioning the score network (amortised model), or at inference time via reconstruction guidance (joint model). The authors present an improved autoregressive sampling technique for the latter, which is shown to be important for forecasting tasks. The methods are evaluated on multiple PDE datasets, including on a hybrid forecasting+data assimilation problem setup.

**Strengths:**

- The studied models can tackle both forecasting and data assimilation jointly. Even if performance lags behind forecasting-only models, there might exist use-cases for this, and performance could be improved in future work.
- There are a lot of careful ablations and insightful comparisons of the studied conditional diffusion models variants.
- The universal amortised model is an interesting approach (taken from the ML for natural videos literature) that seems to perform promisingly.

**Weaknesses:**

- The forecasting evaluation could be enhanced with probabilistic metrics such as the CRPS and spread-skill ratios, by evaluating ensemble forecast of the diffusion models. It would also be beneficial to see evaluations of the spectra of the predictions.
- I think that it would be useful to include algorithms or pseudocode for how the different methods (joint AR, joint AAO, amortised etc.) are trained and how they function at inference time. This will make it easier to compare the different approaches in a more detailed and self-contained way.
- In fig. 1 for Kolmogorov: It would be nice to include some results for the amortised model, especially since you observe differences between the datasets.
- I might have missed an ablation for the window size W=P+C of the universal amortised model. This would be especially interesting to see on the Kolmogorov dataset (i.e. basically repeat Fig. 1 right but with W \in {3, 7, 9} etc.)

**Questions:**

- Is the MSE-trained U-Net using the same modern U-net architecture as your diffusion-based methods do?
- For the (non-universal) amortised models, you train one different model for each unique combination of P | C?  How do you explain that the amortised models trained for a specific P | C combination underperform the universal amortised model that might need to distribute capacity between all the P | C combinations that it was trained on?
- You could also perform the "universal" training of the amortised models (i.e. train on any combination of P|C) for non-diffusion models, right? That would be very interesting to see (e.g. it may or may not work for a MSE-trained model).

**Limitations:**

Yes.

---

> ### Author Rebuttal · Authors · 2024-08-06
>
> Thank you for your review and your comments, we aim to address them below.
>
>
> **Evaluation metrics** - Thank you for your suggestions, when we chose the evaluation metrics we followed [1], but we acknowledge that some other metrics can be used to assess the model. We provide some examples of energy spectra for the trajectories in the attached PDF (Fig. 3 and Fig. 9), showing that they are in agreement with the ground-truth spectra for low frequencies, and the differences mostly come from high frequency components, as also noted in [1].
>
> **Algorithm/Pseudocode** - Thank you for the suggestions, we agree that this would make it easier to compare the different methods and will include them in the updated manuscript should the paper be accepted.
>
> **Amortised Kolmogorov** -  Thank you for your suggestion. These models are still training at the moment because the cluster is currently very busy and they have been spending some time in the queue. We hope to be able to provide the results by the end of the discussion period. But this is definitely something we will include in the final version of the manuscript.
>
> **Ablation window size** - We launched models for window 3 and 7 for Kolmogorov for both the joint and amortised models. However, please note that because of the time constraints, their hyperparameters are not tuned and from the training curves, we can clearly see that the window 7 model would benefit from more training. We provide a few example results in terms of high correlation time below, as the other scenarios are still currently being queued on our cluster:
>
> - Amortised $1 \mid 2$ (window $3$) - $9.09 \pm 0.50$s
> - Amortised $1 \mid 4$ (window $5$) - $8.85 \pm 0.53$s - from paper
> - Amortised $1 \mid 6$ (window $7$) - $8.63 \pm 0.50$s
> - Amortised $3 \mid 4$ (window $7$) - $8.01 \pm 0.44$s
> - Joint $1 \mid 2$ (window $3$) - $7.63 \pm 0.41$s
> - Joint $1 \mid 4$ (window $3$) - $8.16 \pm 0.62$s - from paper
> - Joint $1 \mid 6$ (window $7$) - $7.73 \pm 0.50$s
>
> For the joint model, we can see that the performance is comparable between all window sizes, although the best one is achieved with the window $5$ model. We believe the window $7$ one does not outperform it because of convergence issues.
>
> For the universal amortised, we see a decreasing trend in performance with increasing window size. However, the performance of all models with $P=1$ is within error (and hence, comparable). We believe that this result might be influenced by hyperparameter tuning.
>
> **Architecture** - No, the MSE-trained U-Net and the modern U-Net architecture are not the same. We have experimented with several architectures and found that the architecture inspired by [2] (SDA) was the most stable one in the context of joint prediction. The MSE-trained U-Net is inspired by [1] and we found it better suited for amortised-style of models. As such, we optimised the architecture for each model and chose the optimal one, meaning that we used the SDA architecture for the joint and for the amortised models (for a one-to-one comparison), and the PDE-Refiner architecture for the baselines used in forecasting.
>
> We will include a more detailed description of the MSE-trained U-Net in the Appendix in the updated version of the manuscript, thanks again for pointing this out.
>
> **Non-universal vs. universal** - You are correct and we also thought this finding is interesting. We believe the reason why the universal amortised models outperforms the plain amortised models (where we need to train a model for each $P \mid C$ combination) is because the training task becomes more complex, which encourages generalisation. In effect, the universal amortised model also amortises over the predictive horizon $P$. We note, however, that this comes with the downside that in the KS case we had to train the universal amortised model for longer than the plain amortised models (which would start overfitting if they had been trained for the same amount of epochs as the universal amortised model).
>
> **Universal amortised for non-diffusion models** - This is indeed possible, as the framework we propose is not specific to diffusion-based models. We find your suggestion very interesting, but potentially not directly relevant to the topic of the paper, which aims to study diffusion-based models for PDE modelling.
>
> **References**
>
> [1] Lippe, P. et al. (2023). PDERefiner: Achieving Accurate Long Rollouts with Neural PDE Solvers. In Thirty-Seventh Conference on Neural Information Processing Systems.
>
> [2] Rozet, F. and Louppe, G. (2023). Score-based Data Assimilation. In Thirty-Seventh Conference on Neural Information Processing Systems.

---

> > ### Comment · Area_Chair_cgB4 · 2024-08-13
> >
> > Dear reviewer, please read the above rebuttal and evaluate whether it answers your concerns. If your evaluation remains unchanged, please at least acknowledge that you have read the author's response.

---

> > ### Comment · Reviewer_ofj7 · 2024-08-13
> >
> > I thank the authors for their rebuttal and responses. Based on the other reviews and the rebuttal, I tend towards keeping my score of 6. Some specific comments are below.
> >
> > I would encourage you to include the Amortised Kolmogorov results and algorithm pseudocode in your revised paper.
> >
> > > Ablation window size
> >
> > Your points regarding missing convergence or improper hyperparameter (HP) tuning would be great points to figure out for a revised draft too. It's minor of course (and not something that's absolutely needed for sure), but I do think that it would help readers and practitioners who may want to use your method on their own problems and datasets.
> >
> > > No, the MSE-trained U-Net and the modern U-Net architecture are not the same.
> >
> > I see. I would think that the most fair comparison would be using an MSE-trained modern UNet. Are you saying that you tried this but found that the MSE-trained U-Net inspired by [1] performs better than the modern UNet (with same HPs, optimization and similar parameter sizes)? If so, great. If not, very important to run this comparison.

---

> > > ### Author Response · Authors · 2024-08-14
> > >
> > > Thank you for taking the time to go through our rebuttal and responses, we greatly appreciate your feedback and will make sure to incorporate it in the final version of the manuscript.
> > >
> > > **Amortised Kolmogorov results** –  We thank the reviewer for this useful suggestion, and are currently working on getting the results for the final version.
> > >
> > > **Algorithm pseudocode** – We will make sure to include the pseudocode in the appendix of the updated manuscript.
> > >
> > > **Ablation window size** – We agree that further investigation of missing convergence and HP tuning might be insightful for readers and practitioners, and we will aim to add more information about this to the appendix of the updated manuscript.
> > >
> > > **MSE-trained U-Net and modern U-Net** –  We have investigated the performance of both architectures for guided and universal amortised models. From our experiments, the modern U-Net suggested in [2] has better performance for the guided model, while MSE-trained U-Net from [1] performs better or comparable to the modern U-Net in the amortised case. We have only used the MSE-trained U-Net for forecasting baselines, as the PDE-Refiner paper has already tuned the architecture for the tasks considered, so further tuning of architecture for baselines was beyond the scope of our work. We will make sure to add the discussion on architectures to the appendix of the updated version of the paper.

---

> > > > ### Comment · Reviewer_ofj7 · 2024-08-14
> > > >
> > > > Thanks! This all seems reasonable and I look forward to reading the revised paper.
> > > >
> > > > About the forecasting baseline. Thanks for the clarification. I understand it. Optimally, I'd still recommend trying a MSE-trained modern Unet as an alternative forecasting baseline. You don't need to do any tuning but you could simply use the same architecture that you used for the guided/amortised models. This experiment would eliminate architecture as a potential reason for differences in performances that are not intrinsic to the guided/amortised diffusion methods.

---

> > > > > ### Author Response · Authors · 2024-08-14
> > > > >
> > > > > Thanks for the clarification. Indeed, adding this additional baseline would be a good experiment to separate any architectural differences between the models and we will aim to run it and include the results in the appendix.
> > > > >
> > > > > Thank you once again for all your feedback!

---

### Official Review · Reviewer_iiGR · 2024-07-18

**Soundness:** 3
**Presentation:** 2
**Contribution:** 2
**Rating:** 7
**Confidence:** 3

**Summary:**

The authors perform an extensive study of generative diffusion models applied to the task of PDE forecasting and data assimilation (DA). Further, they introduce an autoregressive (AR) sampling strategy, and a universal amortised model based with variable context size, based on masking.

**Strengths:**

The authors have done a large number of tests with different configuration, leading to useful insights for the community that go beyond the architectures introduced in this paper. The AR sampling does not suffer from the poor memory scaling of AAO sampling. By leveraging masking during training (similar to how LLMs are trained), their universal amortised model is flexible and robust with respect to the size of the conditioning window.

**Weaknesses:**

AR sampling scales poorly in time as each step costs (1 + c)*p evals, that is, a diffusion sampling process, known to be expensive, has to be performed at every timestep. The inference times for both AR and AAO models are large compared to the baselines in the forecasting literature or to Rozet et al. for DA. The validation metrics presented could be clearer. The RMSD and MSE (appendix) presented are not normalized (in some sense) to account for the magnitude of the data, making the absolute number hard to interpret. Further, insightful metrics are missing, such as the energy spectrum of the predicted trajectories or long term trajectory stability. There are only a few comparison trajectories for the forecasting tests. While qualitative results are always subjective, it is known to be hard to judge the quality of a predicted trajectory just from quantitative summary statistics. Similarly, for the offline DA results, the partial observations are not shown.

**Questions:**

1. What are the training times? How do they relate to model performance, i.e. validation metrics?
2.  What is the long-term behaviour of the AR model? I.e. longer than seen during training.
3. How do the Fourier modes of the predicted trajectories look like? Given that PDE-Refiner is a baseline it would be interesting to see how it compares.
4. How well do the predicted trajectories match the physics? This is related to the question above (the energy spectrum, for example).

 -- End of questions --

A brief explanation of the review and grades given, put here for lack of a better place:

Soundness: 3 - extensive benchmarking
Presentation: 2 - appropriate but limited validation metrics (see Weaknesses)
Contribution: 2 - extensive benchmarking, approach novel in this context, lacking some interesting metrics

Rating 6 - A technically solid paper with extensive tests. It is held back by the validations done, the scalability issues, and inference times.

Rating after rebuttal 7 - Added validation metrics and spectral studies. Scalability, accuracy in forecasting task and inference times remain an open question for future work.

**Limitations:**

Some of the limitations are properly acknowledged. The time scaling of the AR sampling, and the lower performance on forecasting tasks compared to SOTA approaches. The fact that while flexible, the universal amortised model has a maximum conditioning size (as opposed to AR models) should be mentioned.

---

> ### Author Rebuttal · Authors · 2024-08-06
>
> We thank the reviewer for the time taken to review our paper and the positive consideration of it. We address the comments below.
>
> **Alternative metrics**
>
> Thanks for suggesting alternative metrics, such as the energy spectrum, as well as providing more qualitative examples. We agree that these are indeed helpful in understanding the quality of the generated PDE trajectories, Please find in the attached PDF:
> - Example energy spectra (Fig. 3 - KS and Fig. 9 - Kolmogorov).
> - More examples of trajectories from each of the evaluated models on forecasting (e.g. Fig. 1 - KS and Fig. 2 - Kolmogorov, besides the examples provided in Fig. 24 for KS and Fig. 25 for Kolmogorov in the Appendix)
> - Example of trajectories for DA offline, alongside the masks used and interpolation results, which we included as a baseline for offline DA (Fig. 5 - KS and Fig. 6 - Kolmgorov)
>
> We will make sure to include them in the final version of the manuscript.
>
> For the quantitative evaluation, we decided to use high correlation time for an easier comparison with the results provided in [1]. RMSD/MSE are metrics that we frequently encounter in the literature (e.g. [1], [2]). While we acknowledge they are not perfect in evaluating PDE dynamics, we believe the problem of finding the right metric is a research question on its own and was not the focus of this investigation.
>
> **Training times** - We found that for the KS dataset, the joint model requires less training (4k epochs) compared to the universal amortized model (15k) steps (please also see Table 3 p.21 for more training details). However, both models, universal amortised and joint, were trained for the same amount of epochs for Kolmogorov. All considered models take from a few days to a maximum of one week to train depending on the dedicated GPU hardware. Since we relied on a GPU cluster with different machines on it, it is challenging for us to report a more specific comparison about the training times.
>
> Standard diffusion validation loss (measured by how well the model predicts the added noise) is not comparable across joint and amortised models, as the amortised model predicts the conditional score, so validation loss is often smaller than the one for the joint model.
>
> **Long-term trajectory stability** - The KS dataset is already analysing this behavior, since training examples are shorter than validation and test (140 states / 28s for train vs. 640 states / 128s for validation and test). The evolution of MSE error can be found in Fig. 23, where the error starts to linearly increase after around 50 seconds.
>
> To analyse even longer-term behaviour (e.g. does the model still produce plausible states), we generated a trajectory of 2000 steps (400s), clearly showing that the model produces plausible predictions (see Fig. 4 on the attached PDF). Moreover, we computed the energy spectra of a states close to the initial condition, a state in the middle of the trajectory, and of the last state. We observe that the energy spectra of these three states do not show different behaviour, indicating that our model does not start generating physically unplausible states if we roll it out for very long times.
>
> **Fourier modes** - Please see Fig. 3 and Fig. 9 on the attached PDF for example energy spectra.
>
> **Physics match** - We believe the energy spectrum + long-term behaviour already address this question, but we are open to investigate other ways of assessing how well the generated trajectories match the physics.
>
> **References**
>
> [1] Lippe, P. et al. (2023). PDERefiner: Achieving Accurate Long Rollouts with Neural PDE Solvers. In Thirty-Seventh Conference on Neural Information Processing Systems.
>
> [2] Georg Kohl et al. Turbulent flow simulation using autoregressive conditional diffusion models. 2023.

---

> ### Comment · Reviewer_iiGR · 2024-08-09
> **Answer to rebuttal**
>
> Thank you for addressing my concerns regarding the metrics, and for providing them in the PDF. I think that their addition to the appendix of the paper strengthens it overall.
>
> The weakness regarding inference cost persists (I read the rebuttal to D3JV). I see the analysis and discussion regarding PDE-Refiner, but that just means that both methods are expensive. However, I acknowledge that this is an initial exploration and therefore cannot be expected to solve all problems.
>
> I see from the spectral analysis that PDE-Refiner has a more accurate spectra, and that even the U-Net and FNO perform similarly or better than the proposed method. I agree with D3JV that the denoising process could and should be used to achieve effects similar to PDE-Refiner. While I also agree with you in your rebuttal to D3JV when you point out that the non-forecasting tasks are not trivial to incorporate into PDE-Refiner's framework, I think it is a necessary improvement to the proposed method, as it can at least amortize some of the inference cost.
>
> All in all, you have addressed some of my concerns, and I acknowledge that those that remain unresolved can (and should) be the subject of future work. I am happy to raise my score from 6 to 7 and recommend that this paper be accepted.

---

> > ### Author Response · Authors · 2024-08-09
> >
> > We greatly appreciate you taking the time to go through our rebuttal. We will make sure to add the extra material we provided in the appendix.
> >
> > **Inference cost** - We agree that the computational cost is one of the main weaknesses of diffusion models (including, thus, PDE-Refiner), but as you said, improving inference costs is a research question on its own. This was not necessarily our concern in this paper, but we agree that it would be an interesting research question to explore.
> >
> > **Spectral analysis** - Indeed, in the KS case, PDE-Refiner and U-Net seem to have more accurate spectra in the context of forecasting. But as you pointed out, adjusting PDE-Refiner to be able to successfully tackle non-forecasting tasks is non-trivial—these tasks are of significant interest to us, as one of the main motivations of the study is to develop a highly flexible (in terms of conditioning) diffusion-based model for PDE modelling. Nevertheless, we agree that integrating some of our techniques into PDE-Refiner would be an interesting research direction to explore.
> >
> > We are happy we managed to address some of your concerns and got the chance to discuss some promising future research directions. Thank you for increasing your score and recommending the paper to be accepted.

---

### Official Review · Reviewer_D4vs · 2024-07-21

**Soundness:** 3
**Presentation:** 3
**Contribution:** 3
**Rating:** 7
**Confidence:** 2

**Summary:**

- The paper tackles of the problem of ML-based PDE modelling. Specifically two sub-problems: (a) *forecasting*: to generate rollouts given initial observations; and (b) *data assimilation*: to refine a trajectory given partial and noisy observations.
- The proposed approach extends a score-based diffusion model solution (notably, related to Rozet & Louppe NeurIPS '23, and Lippe et al. NeurIPS '23) that jointly tackles the two sub-problems.
- First, by proposing a *joint model* that allows for auto-regressive sampling of the trajectory (in contrast to "all-at-once" of Rozet & Louppe). Second, to consider *amortization* using a conditional diffusion model that no longer requires multiple trained models for choices of prediction and correction steps.
- The approach is evaluated on standard benchmarks 1D Kuramoto-Sivashinsky ("KS") and 2D Kolmogorov flow ("Kolmogorov") and compared with standard baselines (e.g., PDERefiner of Lippe et al.). For forecasting, the results are on-par or behind baselines. For data assimilation, the approach outperforms Rozet & Louppe.

**Strengths:**

1. The paper proposes a common framework to tackle multiple PDE simulation related tasks of forecasting, offline and online data assimilation. This appears to constrast existing work that solely focus on one.
2. The paper is overall well-written. I especially appreciated the related works discussions, which also highlight concurrent works.
3. The evaluation is reasonable, and considers multiple relevant baselines (including very recent ones).
4. The approach (although involves more computation) has some practical merits e.g., using "universal amortized" strategy to rely on a single trained model.

**Weaknesses:**

## Major concerns

I have no major concerns with the paper.

## Minor concerns

**1. Results - Forecasting**
- While a common framework for multiple PDE simulation tasks is appealing, I wonder the specific contributions for the forecasting problem given that many baselines outperform the proposed approach (and if I understand right, with lower computation cost).
- Although, the paper claims (p8, L299) the approach is appealing for flexibility, I am unclear on scenarios that require this at the price of empirical gains.

**2. Slightly unclear - experimental settings/results on offline DA vs. AAO**
- Firstly, I have to preface that the AAO paper (Rozet & Louppe, 2023) doesn't appear to have an extensive quantitative analysis.
- However, I am curious of the difference of the experimental setup between this and AAO paper: is the focus here sparse-observations, whereas AAO reports dense observations?
- Moreover, in the dense observation setting, I see that errors (Fig. 3) are comparable. As a result, I wonder whether the results are in favour of the proposed approach only in the sparse observation regime (where the errors, although better than baselines, are still significantly large).

**Questions:**

1. **Forecasting results**: can the authors comment on gains of the proposed approach for forecasting?
2. **Experimental settings vs. AAO**: can the authors comment on how the setup/evaluation regime differs from that of AAO?

**Limitations:**

Yes, this is a dedicated and balanced discussion on this.

---

> ### Author Rebuttal · Authors · 2024-08-03
>
> We would like to thank the reviewer for their positive consideration. We address the points raised in the Weaknesses and Questions sections below.
>
> **Forecasting results**
>
> It is true that in the plain forecasting task, our proposed models do not achieve SOTA performance, as we also highlight in the limitations section of our work, with the main reason being that we choose to trade off forecasting performance with conditioning flexibility. Nevertheless, the paper brings two main contributions in the context of forecasting:
>
> 1. With the proposed autoregressive (AR) sampling approach, we show that diffusion-based models can achieve results of comparable quality with other benchmarks (e.g., MSE-trained U-Net, MSE-trained FNO), while preserving flexibility. This is in contrast to the all-at-once (AAO) sampling approach proposed by [1] which predicts trajectories that diverge from the ground state early on.
>
> 2. We propose a new training strategy for amortised models (which can also be applied to non-diffusion based models), which achieves stable performance for a wide range of conditioning scenarios. This is in contrast to other amortised approaches from the literature which report a decrease in performance with increasing history length [2].
>
> **Scenarios where flexibility is important**
>
> A setting where flexibility is important is the mix of forecasting and data assimilation (DA) considered in the paper, especially in the online DA case. More specifically, not only do we want to produce forecasts based on some estimated initial states, but we also gather some observations in an online manner that can help the model correct its previous forecasts to better match the observed data.
>
> One important application of the above setting is *weather prediction*. The goal is to produce a forecast for a certain horizon (e.g. days to weeks for medium-range prediction). At the same time, we continuously acquire new observations from a variety of sources, such as weather stations, radiosondes, wind profilers, and satellites [3]. Thus, we would like to update our forecasts whenever these new observations arrive to better reflect reality. This is currently done in two stages, where the forecasting system first generates the forecast, and then the data assimilation system incorporates the observations. However, they are both very costly systems [4]. Most of the current ML-based approaches for weather modeling only tackle the forecasting component (e.g. PanguWeather [5], GraphCast [6]). A more preferable approach would be to perform both tasks at once, being able to flexibly condition on both initial states, as well as any incoming observations. And this is exactly what we are proposing in this paper, with the proposed online DA task being the one that most closely resembles this scenario. However, we do acknowledge that it is still in a toy-ish setting and a natural future direction involves investigating the potential of the technique to scale.
>
> Finally, another setting where flexibility is important is where the nature of the conditioning information might change over time, implying that during training one does not have access to examples of the conditioning information they might encounter at test time. In our case, we address this by conditioning the models using reconstruction guidance. In terms of practical applications, we can imagine an example in the weather modelling setting. For example, a new sensor might be added to the sources of information. Due to the flexibility of our models, we would not have to re-train the model to account for this new instrument.
>
> **Experimental settings vs. AAO**
>
> Indeed, in contrast to our work, [1] does not provide any quantitative results (on the Kolmogorov flow).
>
> Their paper performed the following experiments:
> - Observing the velocity field every four steps, coarsened to a resolution 8 $\times$ 8 and perturbed by Gaussian noise ($\Sigma_y = 0.1 \mathbf{I}$).
> - Observing a regularly sampled sparse velocity field at every time step, with factor $n$ ($n$ implies 1 observation for each $n \times n$ region). They use $n$=2,4,8,16. $n$=16 implies 0.3% observed data.
>
> They did not use the 1D KS dataset, only 2D Kolmogorov flow data. Moreover, in our work, we sample observations at random locations, and at each time step. The fraction of observed variables varies between 0.1% to 30%.
>
> In our experiments, we show that the proposed method is able to perform at least on par with AAO across different settings, and it could, thus, be useful if we expect the fraction of observed variables to vary.
>
> **References**
>
> [1] Rozet, F. and Louppe, G. (2023). Score-based Data Assimilation. In Thirty-Seventh Conference on Neural Information Processing Systems.
>
> [2] Lippe, P. et al. (2023). PDERefiner: Achieving Accurate Long Rollouts with Neural PDE Solvers. In Thirty-Seventh Conference on Neural Information Processing Systems.
>
> [3] Keeley, S. (2022). Observations. https://www.ecmwf.int/en/research/dataassimilation/observations.
>
> [4] Bauer, P. et al. (2020). The ECMWF Scalability Programme: Progress and Plans.
>
> [5] Bi, K. et al. (2023). Accurate medium-range global weather forecasting with 3D neural networks. Nature, 619(7970):533–538.
>
> [6] Lam, R. et al. (2023). GraphCast: Learning skillful medium-range global weather forecasting.

---

> > ### Comment · Reviewer_D4vs · 2024-08-13
> >
> > I thank the authors for their detailed response. After having read the other reviews and rebuttals, I will keep my score of leaning towards acceptance.

---

> > > ### Author Response · Authors · 2024-08-14
> > >
> > > We greatly appreciate you taking the time to go through our rebuttal and the other reviews. Thank you for recommending the paper to be accepted.

---

### Author Rebuttal · Authors · 2024-08-06

We would like to thank the reviewers for the time taken to review our paper, the overall positive consideration of our work, and their feedback and useful suggestions. We are content that we managed to get the main conclusions of our investigation across, providing an “insightful comparison of the studied conditional diffusion models variants”. We are also happy that the universal amortised approach we introduced was found to be “interesting”, “effective”, and “flexible”, with promising performance, especially on tasks that involve a mix between forecasting and data assimilation. Such an example is the online DA scenario, which was inspired from related real-life applications from weather modeling. In the following, we provide a general comment addressing the common concerns of the reviews. We are happy to clarify further during the discussion period if some concerns still remain.

**Performance metrics**

Several reviewers suggested alternative performance metrics, besides the ones used in the paper (high correlation time, RMSD, and per time-step MSE in the appendix). We chose these metrics following other works from the field ([1], [2]). However, we do acknowledge that they have their weaknesses, and the usefulness of complementing the quantitative metrics with some qualitative ones to better understand the behaviour of our models. As such, we provide in the attached PDF:

- In addition to Fig. 24 and Fig. 25 from Appendix F.5, we provide additional examples of forecasting trajectories for qualitative evaluation of our models (Fig. 1 - KS and Fig. 2 - Kolmogorov). For offline DA, we provide additional examples where we show the generated trajectories (joint AR), alongside the masks used, and an interpolation baseline for $10^{-2}$ and $10^{-1}$ proportion observed (Fig. 5 - KS and Fig. 6 - Kolmogorov).

- Energy spectra of predicted trajectories, to assess how they compare to the ground truth and how they compare to the spectra of other relevant models (e.g. PDE-Refiner). In Fig. 3 and Fig. 9 we show the spectra at the beginning of the trajectory and at the end of the trajectory. We can see that, in the KS case (Fig. 3), the spectra of the states generated by our models do not vary significantly depending on how far away they are from the initial state. In contrast, PDE-Refiner significantly overestimates the amplitude of high frequencies as the states depart from the initial conditions. For Kolmogorov (Fig. 9), we observe that all methods (except for MSE-trained FNO) generate similar spectra, that resemble the ground truth spectrum well. We do not observe much difference between states close and far away from the initial state.

- Example of very long KS trajectories (including energy spectra), to investigate the long-term behaviour of our models (Fig. 4). In particular, we expect the samples to diverge from the ground truth after a certain number of states, but the question is whether our models still generate physically plausible predictions. Fig. 4 shows that this is indeed the case. By investigating the energy spectra for different states along the trajectory, we do not observe much difference depending on how far away the states are from the initial one---all spectra indicate that the models capture the low frequencies correctly, but overestimate the high frequency components. Overall, these additional qualitative studies show that, even for longer-term predictions, the models still produce physically realistic states.

- As suggested by reviewer D3JV, we provide a modified version of Figure 3 (top) from the main paper, showing the RMSD for a plain forecasting baseline (MSE-trained U-Net). This aims to show the benefit of the additional observations in the offline DA setting, illustrating that the models can improve performance with relatively few observations (proportion observed higher than $\approx 4\times 10^{-3}$).

**Computational time**

Another common concern was regarding computational time of our approaches.

The joint AR approach does come with an increased computational cost as opposed to other techniques from the literature, but we believe that the rapid advances in diffusion model sampling could bring this cost down. However, the universal amortised model has a similar computational cost to SOTA methods (PDE-Refiner). This is not apparent in the paper and as such we included an extra figure in the PDF (see Fig. 7) that shows that while in the main text we used 128 diffusion steps, the amortised model can achieve similar performance with as few as 16 diffusion steps. We will include this finding in the updated version of the manuscript. For a more detailed analysis of computational cost, please also see response to reviewer D3JV.

We hope we addressed most of your questions and concerns and are looking forward to further engaging with you in the discussion period.

[1] Lippe, P., Veeling, B. S., Perdikaris, P., Turner, R. E., and Brandstetter, J. (2023). PDERefiner: Achieving Accurate Long Rollouts with Neural PDE Solvers. In Thirty-Seventh Conference on Neural Information Processing Systems.

[2] Georg Kohl, Li-Wei Chen, and Nils Thuerey. Turbulent flow simulation using autoregressive conditional diffusion models. 2023.

---

### Comment · Area_Chair_cgB4 · 2024-08-08

Dear authors and reviewers,

The authors-reviewers discussion period has now started.

@Reviewers: Please read the authors' response, ask any further questions you may have or at least acknowledge that you have read the response. Consider updating your review and your score when appropriate. Please try to limit borderline cases (scores 4 or 5) to a minimum. Ponder whether the community would benefit from the paper being published, in which case you should lean towards accepting it. If you believe the paper is not ready in its current form or won't be ready after the minor revisions proposed by the authors, then lean towards rejection.

@Authors: Please keep your answers as clear and concise as possible.

The AC

---

### Decision · Program_Chairs · 2024-09-25

**Decision:**

Accept (poster)

**Comment:**

The reviewers unanimously recommend acceptance (7-7-6-6-7). The author-reviewer discussion has been constructive and has led to a number of improvements to the paper. The authors are asked to implement the changes discussed with the reviewers in the final version of the paper.